# Riemannian Low-rank Adaptation for Federated Fine-tuning of Foundation Models

## Abstract

Rank-adaptive low-rank adaptation (LoRA), a parameter-efficient fine-tuning (PEFT) technology, has achieved state-of-the-art performance in fine-tuning foundation models (FM). Directly transplanting the rank-adaptive LoRA methods from centralized learning to federated learning raises two critical issues: client drift and rank drift. This paper presents a Riemannian LoRA algorithm with adaptive rank for federated fine-tuning of foundation models (FFT-FM), RAFFT, which solves the client-drift and rank-drift issues, and significantly improves the computational cost. First, by utilizing Riemannian Procrustes analysis, we propose a Riemannian parameter matching method to avoid the client-drift issue for ensuring the effectiveness of FFT-FM with rank-adaptive LoRA, and to reduce the cost of matrix decomposition by transforming the singular value decomposition (SVD) of high-dimensional full parameter matrices into the SVD of low-dimensional $r \times r$ matrices, where $r$ is the rank parameter in the LoRA. We theoretically derive the equivalence between our RAFFT algorithm with rank-adaptive LoRA for the FFT-FM and the standard FFT-FM on the full parameter matrices based on FedAvg and verify the bounded error introduced by approximation. Second, by leveraging Riemannian manifold theory, we develop a Riemannian gradient descent (RGD) method to guarantee the local full parameter matrices on clients in the form of low-rank ones with fixed rank optimized by the server in each FFT-FM round, for alleviating the rank-drift issue to speed up the convergence of RAFFT. We theoretically demonstrate that the RGD optimization on the Riemannian manifold ensures the rank invariance during the local update process and the RGD optimization can converge in the FFT-FM context.

## 1 Introduction

Parameter-efficient fine-tuning (PEFT) for fine-tuning of foundation models (FT-FM) has attracted active research in recent years, such as adapter tuning (Houlsby et al., 2019; Lin et al., 2020; Pfeiffer et al., 2021; Rücklé et al., 2021; He et al., 2022), prefix tuning (Li & Liang, 2021), P-Tuning (Liu et al., 2021b), P-Tuning V2 (Liu et al., 2021a), low-rank adaptation (LoRA) (Hu et al., 2022), and prompt tuning (Lester et al., 2021; Li & Liang, 2021). These methods freeze the backbone parameters (i.e., original weights of pre-trained FMs) and adjust only a small portion of parameters (i.e., adapter weights during the FT phase), to improve the efficiency. Among the above PEFT approaches, LoRA has achieved the state-of-the-art FT performance (Wu et al., 2024). The idea of the LoRA method can be illustrated as follows: $\mathbf{W} = \mathbf{W}_0 + \Delta\mathbf{W} = \mathbf{W}_0 + \mathbf{B}\mathbf{A}$, where $\mathbf{W} \in \mathbb{R}^{m \times n}$ is the complete parameter of a FM. $\mathbf{W}_0$ and $\Delta\mathbf{W}$ are the backbone (pre-trained FM parameters) and adapter parameters respectively. A low-rank decomposition $\Delta\mathbf{W} = \mathbf{B}\mathbf{A}$ where $\mathbf{B} \in \mathbb{R}^{m \times r}$, $\mathbf{A} \in \mathbb{R}^{r \times n}$, and rank $r << \min(m, n)$. During the FT phase, only $\mathbf{B}$ and $\mathbf{A}$ are trained while freezing $\mathbf{W}_0$ on the clients. Notice that in the LoRA-style methods, rank $r$ is often user-defined and keeps unchanged during the entire FT process.

In order to further enhance the performance of the LoRA-style methods, various rank-adaptive LoRA methods have been recently proposed to dynamically allocate the ranks among parameter matrices based on their importance: assigning more/less trainable parameters with higher/lower ranks to the critical/insignificant parameter matrices for better model performance/efficiency. These methods can be broadly classified into two categories: (1) SVD-based approaches in centralized fine-tuning (CFT) dynamically adjust the ranks by truncating singular values in the form of SVD (Zhang et al.,

2023b; Ding et al., 2023; Zhang et al., 2023a); (2) Rank sampling-based algorithms in both CFT and FFT search for the best rank by sampling a range of ranks and sorting the representations learnt by the model at different ranks during training (Valipour et al., 2023; Rajabzadeh et al., 2024; Xu et al., 2024). They need to fine-tune the model multiple times at different ranks, raising non-trivial cost. Several pioneering rank-adaptive LoRA algorithms for FFT-FM iteratively aggregate local full parameter metrics into a global one and perform the SVD on the latter to find the optimal rank (Chai et al., 2022; Wu et al., 2021; Niu et al., 2023). However, iterative SVD on the high-dimensional full parameter matrices is prohibitive.

A critical challenge of transplanting the SVD-based rank-adaptive LoRA methods from the CFT to the FFT is the client-drift issue (Sun et al., 2024; Wang et al., 2024). To address this, they attempt to resolve the problem by either freezing one of the parameter matrices or using aggregation methods based on matrix stacking and multiplication. However, we addresses a more challenging problem by considering both low-rank parameter matrices $\mathbf{U}^k, \mathbf{V}^k$ and $\mathbf{\Sigma}^k$ to determine the rank. For a detailed analysis, please refer to Appendix 7.8. Here, we introduce an illustrative example with two clients to better explain this issue. An implicit assumption ensuring the global convergence of FFT-FM based on FedAvg McMahan et al. (2017) is $\Delta \mathbf{W} = \frac{1}{2}(\Delta \mathbf{W}^1 + \Delta \mathbf{W}^2)$, where $\Delta \mathbf{W}$ is the global model parameter after the model aggregation on server, $\Delta \mathbf{W}^1$ and $\Delta \mathbf{W}^2$ are the local model parameter on clients 1 and 2. Let the SVDs be $\Delta \mathbf{W}^1 = \mathbf{U}^1 \mathbf{\Sigma}^1 \mathbf{V}^1$ and $\Delta \mathbf{W}^2 = \mathbf{U}^2 \mathbf{\Sigma}^2 \mathbf{V}^2$. After using FedAvg to aggregate the low-rank parameter matrices, the server produces

$$\underbrace{\frac{1}{2}\left(\mathbf{U}^1 + \mathbf{U}^2\right) \times \frac{1}{2}\left(\mathbf{\Sigma}^1 + \mathbf{\Sigma}^2\right) \times \frac{1}{2}\left(\mathbf{V}^1 + \mathbf{V}^2\right)}_{\text{Parameter aggregation with SVD-based rank-adaptive LoRA + FedAvg}} \neq \underbrace{\frac{1}{2}\left(\mathbf{U}^1 \mathbf{\Sigma}^1 \mathbf{V}^1 + \mathbf{U}^2 \mathbf{\Sigma}^2 \mathbf{V}^2\right) = \frac{1}{2}(\Delta \mathbf{W}^1 + \Delta \mathbf{W}^2) = \Delta \mathbf{W}}_{\text{Ideal parameter aggregation with FedAvg on full parameter matrices}} \quad (1)$$

where the left side denotes the parameter aggregation of FFT-FM with LoRA, while the right side represents the standard parameter aggregation of FFT-FM. Therefore, the client drift issue is defined as the discrepancy between independently aggregating client's low-rank matrices and aggregating the ideal parameter matrix. As shown in Figure 1, the difference between two sides may become more significant when (1) the number of local update steps between aggregations is large and (2) the local datasets are different across clients. In this case, FFT-FM with LoRA may fail to converge and result in poor model performance.

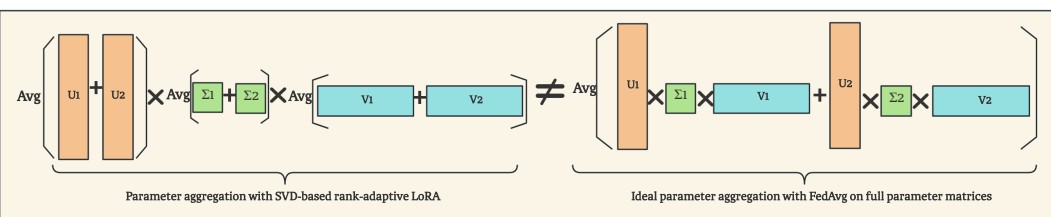

Figure 1: Client Drift in FFT-FM with Rank-Adaptive LoRA

Another significant challenge in FFT-FM with rank-adaptive LoRA is the rank-drift issue, caused by data heterogeneity (McMahan et al., 2017; Smith et al., 2017; Chen et al., 2018; Khodak et al., 2019; Sattler et al., 2019; Liu et al., 2019a; Hsieh et al., 2020), a common problem in federated learning. Non-Independent Identically Distributed (Non-IID) data across clients result in substantial differences between local models and the global model. During training, local models may oscillate across rounds, leading to unstable convergence and suboptimal performance. Similarly, the optimal ranks of local models vary significantly due to data heterogeneity, causing rank-drift. This oscillation further slows convergence and impairs performance. For a detailed analysis, please refer to Appendix 7.8.

To address the client-drift issue, by utilizing Riemannian Procrustes analysis, due to the fact that all local parameter matrices lie on the Riemannian manifold, we propose a Riemannian parameter matching method to match the local parameter matrices $\mathbf{U}^k$ and $\mathbf{V}^k$ on other clients with pivots $\mathbf{U}^1$ and $\mathbf{V}^1$ on client 1, in terms of their lengths and directions. To maintain the consistency between low-rank parameter metrics before and after the Riemannian parameter matching, i.e., $\Delta \mathbf{W}^k = \mathbf{U}^k \mathbf{\Sigma}^k \mathbf{V}^k = \tilde{\mathbf{U}}^k \tilde{\mathbf{\Sigma}}^k \tilde{\mathbf{V}}^k$, for ensuring the effectiveness of FFT-FM with rank-adaptive LoRA, we derive a modified diagonal matrix $\tilde{\mathbf{\Sigma}}^k$ for the other clients by performing the SVD on low-dimensional $r \times r$ matrices $(\mathbf{U}^1)^T \mathbf{U}^k$ and $(\mathbf{V}^1)^T \mathbf{V}^k$. Based on the global diagonal matrix $\frac{1}{K}\left(\mathbf{\Sigma}^1 + \sum_{k=2}^K \tilde{\mathbf{\Sigma}}^k\right)$, it is easy to find the optimal rank of the global parameter matrix, with aggre-

gation on only the local low-rank matrices $\mathbf{U}^k$, $\tilde{\mathbf{\Sigma}}^k$, and $\mathbf{V}^k$ (See Eq.(14)). We theoretically derive the equivalence between our RAFFT algorithm with rank-adaptive LoRA for the FFT-FM and the standard FFT-FM on the full parameter matrices based on FedAvg and verify the bounded error introduced by approximation.

For alleviating the rank-drift issue, we propose to extend stochastic gradient descent (SGD) algorithms on a Riemannian manifold and develop a Riemannian gradient descent (RGD) method to guarantee the local full parameter matrices $\Delta\mathbf{W}^k$ on the clients in the form of low-rank ones $\mathbf{U}^k$, $\mathbf{\Sigma}^k$, and $\mathbf{V}^k$ with fixed rank optimized by the server in each FFT-FM round, to speed up the convergence of RAFFT: $\left((\mathbf{U}^k)^{(t+1)}, (\mathbf{\Sigma}^k)^{(t+1)}, (\mathbf{V}^k)^{(t+1)}\right) = f\left(-\eta_t\left(\left(\Gamma_{\mathbf{U}^k}L^k\right)^{(t)}, \left(\Gamma_{\mathbf{\Sigma}^k}L^k\right)^{(t)}, \left(\Gamma_{\mathbf{V}^k}L^k\right)^{(t)}\right)\right)$, where $\eta_t$ is a learning rate at round $t$, $\Delta\mathbf{W}^k$ is a point on a Riemannian manifold $M$ consisting of all matrices with a fixed rank $r$ in $\mathbb{R}^{m\times n}$, $L^k(\Delta\mathbf{W}^k)$ is the loss function $L^k$ at $\Delta\mathbf{W}^k$ in Euclidean space, $\Gamma_{\mathbf{V}^k}L^k$ represent the components of the Riemannian gradient at $\Delta\mathbf{W}^k$, and $f$ is a retraction $f : T_M \to M$ mapping the tangent space at $\Delta\mathbf{W}^k$ to the manifold. We theoretically demonstrate that the RGD optimization on the Riemannian manifold ensures the rank invariance during the local update process and the RGD optimization can converge in the FFT-FM context.

To our best knowledge, this work is the first to offer a rank-adaptive low-rank adaptation solution for FFT-FM with Riemannian theory. Our RAFFT method exhibits three compelling advantages: (1) The proposed Riemannian parameter matching technique based on Riemannian Procrustes analysis is able to solve the client-drift issue when transplanting the centralized LoRA methods to the federated setting. Existing rank-adaptive LoRA methods in the CFT can be plugged into our framework for FFT-FM. Moreover, the model aggregation of FFT-FM is conducted on only local low-rank parameter matrices, without the need of aggregating the global full parameter matrices; (2) Transforming the SVD of high-dimensional full parameter matrices into the SVD of low-dimensional $r\times r$ matrices can significantly improve the efficiency of FFT-FM with low-rank adaption. In addition, this design avoids the expensive cost of training FFT multiple times at different ranks by rank sampling-based algorithms; and (3) The combination of Riemannian gradient descent and Riemannian manifold theory guarantees the local parameter matrices with fixed rank optimized by the server in each FFT-FM round, for alleviating the rank-drift issue to speed up the convergence of RAFFT.

Empirical evaluation on real datasets demonstrates the superior performance of our RAFFT model against several state-of-the-art federated prompt tuning, federated LoRA, and federated rank-adaptive LoRA methods. More experiments, implementation details, and hyperparameter setting are presented in Appendices 7.5-7.6.

## 2 BACKGROUND

### 2.1 RANK-ADAPTIVE LOW-RANK ADAPTATION

AdaLoRA Zhang et al. (2023b) is a parameter-efficient fine-tuning (PEFT) method in centralized setting that parameterizes full parameter metrics as $\Delta\mathbf{W} = \mathbf{U}\mathbf{\Sigma}\mathbf{V}$ and adaptively allocates the rank $r$ for low-rank parameter metrics $\mathbf{U}$, $\mathbf{\Sigma}$, and $\mathbf{V}$ according to their importance in the form of singular value decomposition (SVD) of $\Delta\mathbf{W}$.

$$\mathbf{W} = \mathbf{W}_0 + \Delta\mathbf{W} = \mathbf{W}_0 + \mathbf{U}\mathbf{\Sigma}\mathbf{V} \tag{2}$$

where $\mathbf{W}, \mathbf{W}_0, \Delta\mathbf{W} \in \mathbb{R}^{m\times n}$ are the complete, backbone, and adapter parameter matrices respectively. $\mathbf{U} \in \mathbb{R}^{m\times r}$ and $\mathbf{V} \in \mathbb{R}^{r\times n}$ are matrices representing the left and right singular vectors of full parameter matrices $\Delta\mathbf{W}$ respectively. $\mathbf{\Sigma} \in \mathbb{R}^{r\times r}$ is a diagonal matrix containing the singular values $\{\mathbf{\Sigma}_{ii}\}_{1\leq i\leq r}$ with rank $r << \min(m,n)$. AdaLoRA is thus able to dynamically adjust the number of the singular values in $\mathbf{\Sigma}$ and assign more/less trainable parameters with higher/lower rank $r$ to the critical/insignificant parameter matrices for better model performance/efficiency.

### 2.2 FEDERATED FINE-TUNING

First, given a machine learning (ML) task (e.g., image classification), $K$ clients with their local training data $D = \{D_1, \cdots, D_K\}$, and a server, federated learning (FL) based on the FedAvg algorithm McMahan et al. (2017) aims to learn a global ML model on the server by optimizing the problem below.

$$\min_{\mathbf{W} \in \mathbb{R}^{m \times n}} \mathcal{L}(\mathbf{W}) = \sum_{k=1}^{K} \frac{n^k}{n} L^k(\mathbf{W}) \qquad \text{where } L^k(\mathbf{W}) = \frac{1}{n_k} \sum_{\{x_i, y_i\} \in D^k} l_i(\mathbf{W}) \qquad (3)$$

where $l_i(\mathbf{W}) = l(x_i, y_i; \mathbf{W})$ denotes the loss function of the prediction on data example $\{x_i, y_i\} \in D_k$ made with a global model parameter $\mathbf{W}$. $n^k = |D_k|$ denotes the size of local dataset $D_k$. $n$ is the size of total training data $D$, i.e., $n = n^1 + \cdots + n^K$. In the FL, the global model parameter $\mathbf{W}$ is iteratively updated with the aggregation of all local model parameters $\mathbf{W}^1, \cdot, \mathbf{W}^K$ on $K$ clients in each round, i.e., $\mathbf{W} = \sum_{k=1}^{K} \frac{n^k}{n} \mathbf{W}^k$.

In the context of federated fine-tuning (FFT), in Eq.(2), the backbone parameter matrices $\mathbf{W}_0$ are frozen and only the adapter parameter matrices $\Delta\mathbf{W}$ are adjusted in each round by updating their low-rank versions $\mathbf{U}$, $\boldsymbol{\Sigma}$, and $\mathbf{V}$ respectively.

$$\min_{\Delta\mathbf{W} \in \mathbb{R}^{m \times n}} \mathcal{L}(\Delta\mathbf{W}) = \sum_{k=1}^{K} \frac{n^k}{n} L^k(\Delta\mathbf{W}) \qquad \text{where } L^k(\Delta\mathbf{W}) = \frac{1}{n_k} \sum_{\{x_i, y_i\} \in D^k} l_i(\Delta\mathbf{W}) \qquad (4)$$

# 3 RIEMANNIAN PARAMETER MATCHING FOR ELIMINATING CLIENT DRIFT ON SERVER

As discussed in Section 1, directly using FedAvg McMahan et al. (2017) to aggregate the clients' local matrices $\mathbf{U}^k, \boldsymbol{\Sigma}^k, \mathbf{V}^k$ on the server side leads to client-drift issues, as shown in Eq.(1), which negatively impact the convergence and performance of federated learning. Among three categories of rank-adaptive low-rank approximation techniques, SVD-based methods in centralized learning Zhang et al. (2023b); Ding et al. (2023); Zhang et al. (2023a) encounter the dilemma of the client drift issue (Sun et al., 2024), rank sampling-based algorithms Valipour et al. (2023); Rajabzadeh et al. (2024); Xu et al. (2024) and federated rank-adaptive LoRA algorithms Chai et al. (2022); Wu et al. (2021); Niu et al. (2023) raise extremely expensive cost. To address these challenges, we propose a Riemannian parameter matching method to match the local parameter matrices $\mathbf{U}^k$ and $\mathbf{V}^k$ from other clients with the pivot matrices $\mathbf{U}^1$ and $\mathbf{V}^1$ on client 1, i.e., $\tilde{\mathbf{U}}^k \approx \mathbf{U}^1$ and $\tilde{\mathbf{V}}^k \approx \mathbf{V}^1$, preserving both their lengths and directions. Correspondingly, we adjust the diagonal matrix $\boldsymbol{\Sigma}^k$ to ensure consistency in the low-rank parameter measurements before and after Riemannian parameter matching, i.e., $\mathbf{U}^k \boldsymbol{\Sigma}^k \mathbf{V}^k = \tilde{\mathbf{U}}^k \tilde{\boldsymbol{\Sigma}}^k \tilde{\mathbf{V}}^k \approx \mathbf{U}^1 \tilde{\boldsymbol{\Sigma}}^k \mathbf{V}^1$. This approach allows the server to directly aggregate the aligned local low-rank matrices $\mathbf{U}^k$, $\tilde{\boldsymbol{\Sigma}}^k$, and $\mathbf{V}^k$ while avoiding client-drift issues. Moreover, the global diagonal matrix enables the server to easily identify the optimal rank for the global parameter matrix, enhancing both efficiency and performance.

Since both $\mathbf{U}^k$ and $\mathbf{V}^k$ are orthogonal matrices, they belong to the Riemann manifold, which is defined as the set of standard orthogonal matrices $S = \{\mathbf{U}^k \in \mathbb{R}^{m \times r} : (\mathbf{U}^K)^T(\mathbf{U}^K) = I\}$ (Chakraborty & Vemuri, 2019; Atiyah & Todd, 1960). Therefore, we can performing the Riemannian parameter matching to ensure that the length and direction of the parameter matrices between clients are as aligned as possible. We adopt the Procrustes distance metric proposed in Kendall et al. (2003) to efficiently compute the parameter matching on the manifold. The procrustes representation $S_p$ on the Riemann manifold $S$ can be defined as follows: an $m \times r$ matrix $\mathbf{U}^k$ on $S_{r,m}$ is identified with the equivalence class of matrices $\mathbf{U}^k \mathbf{R}^k$ in $\mathbf{R}_{m,r}$, for $\mathbf{R} > 0$.

The squared Procrustes distance for each two matrices $\mathbf{U}^k$ and $\mathbf{U}^1$, considering the representation $S_p$ on the Riemann manifold $S$, is defined as the minimum squared Euclidean distance over all pairs of matrices in the respective equivalence classes.

Therefore, we aim to solve the following Riemann distance optimization problem.

$$\begin{aligned} d_{S_P}^2(\mathbf{U}^1, \mathbf{U}^k) &= \min_{\mathbf{R}^k > 0} \text{tr}\left((\mathbf{U}^1 - \mathbf{U}^k \mathbf{R}^k)^T(\mathbf{U}^1 - \mathbf{U}^k \mathbf{R}^k)\right) \\ &= \min_{\mathbf{R}^k > 0} \text{tr}\left((\mathbf{R}^k)^T \mathbf{R}^k - 2(\mathbf{U}^1)^T \mathbf{U}^k \mathbf{R}^k + I_r\right) \end{aligned} \qquad (5)$$

where $\mathbf{I_r}$ is an identity matrix. Matrix $\mathbf{R}^k$ is a $r \times r$ symmetric positive constraint that ensures the aggregated points remain on the Riemannian manifold.

Notice that $(\mathbf{U}^1)^T \mathbf{U}^k$ is a low-rank matrix with low dimensions $r \times r$. We perform the SVD on $(\mathbf{U}^1)^T \mathbf{U}^k$ to generate $(\mathbf{U}^1)^T \mathbf{U}^k = \mathbf{P}^k \boldsymbol{\Lambda}^k \mathbf{Q}^k$. According to the properties of the matrix trace and

the symmetry of the Frobenius inner product (Schneider, 2015), the optimization problem in Eq.(5) is further converted to another equivalent optimization problem below.

$$\min_{\mathbf{R}^k} \operatorname{tr}\left((\mathbf{R}^K)^T \mathbf{R}^k - 2(\mathbf{U}^1)^T \mathbf{U}^k \mathbf{R}^k\right) = \min_{\mathbf{R}^k} \|\mathbf{R}^k - \mathbf{P}^k \mathbf{\Lambda}^k \mathbf{Q}^k\|_F^2 = \min_{\mathbf{R}^k} \|(\mathbf{Q}^k)^T \mathbf{R}^k \mathbf{Q}^k - (\mathbf{Q}^k)^T \mathbf{P}^k \mathbf{\Lambda}^k\|_F^2 \tag{6}$$

We introduce two auxiliary variable $\mathbf{Y}^k$, $\mathbf{Z}^k$ and let $\mathbf{Y}^k = (\mathbf{Q}^k)^T \mathbf{R}^k \mathbf{Q}^k$, $\mathbf{Z}^k = (\mathbf{Q}^k)^T \mathbf{P}^k$, then we transform the Riemann Procrustes distance optimization problem into an equivalent optimization problem.

$$d_{S_P}^2(\mathbf{U}^1, \mathbf{U}^k) \iff \min_{\mathbf{R}^k} \|\mathbf{Y}^k - \mathbf{Z}^k \mathbf{\Lambda}^k\|_F^2 = \sum_{i=1}^r (\mathbf{Y}_{ii}^k - \mathbf{Z}_{ii}^k \mathbf{\Lambda}_i^k)^2 + \sum_{j>i}\left((\mathbf{Y}_{ij}^k - \mathbf{Z}_{ij}^k \mathbf{\Lambda}_i^k)^2 + (\mathbf{Y}_{ij}^k - \mathbf{Z}_{ji}^k \mathbf{\Lambda}_j^k)^2\right) \tag{7}$$

where $\mathbf{Y}_{ij}^k$, $\mathbf{Z}_{ij}^k$ and $\mathbf{Z}_{ji}^k$ represent the elements of the symmetric matrix $\mathbf{Y}^k$ and matrix $\mathbf{Z}^k$, respectively. Additionally $\mathbf{Y}_{ii}^k$, $\mathbf{\Lambda}_i^k$ and $\mathbf{\Lambda}_j^k$ denote the diagonal elements of the symmetric matrix $\mathbf{Y}^k$ and matrix $\mathbf{\Lambda}^k$.

Given the inherent symmetry of Y, the variables $\mathbf{Y}_{ij}^k$ (where j $\geq$ i ) are uncoupled, allowing us to minimize each term separately. Specifically, we minimize $(\mathbf{Y}_{ii}^k - \mathbf{Z}_{ii}^k \mathbf{\Lambda}_i^k)^2$ and $(\mathbf{Y}_{ij}^k - \mathbf{Z}_{ij}^k \mathbf{\Lambda}_i^k)^2 + (\mathbf{Y}_{ij}^k - \mathbf{Z}_{ji}^k \mathbf{\Lambda}_j^k)^2$ for $j > i$. Since Y is a symmetric matrix, it suffices to compute only the upper triangular part. Then we generate the solution of the optimal problem as follows.

$$\mathbf{Y}_{ij}^k = \begin{cases} \mathbf{Z}_{ii}^k \mathbf{\Lambda}_i^k, & i = j \leq \operatorname{rank}((\mathbf{U}^1)^T \mathbf{U}^k), \\ \frac{\mathbf{z}_{ij}^k \mathbf{\Lambda}_i^k + \mathbf{z}_{ji}^k \mathbf{\Lambda}_j^k}{2}, & j > i \text{ and } i \leq \operatorname{rank}((\mathbf{U}^1)^T \mathbf{U}^k), \\ 0, & \text{otherwise.} \end{cases} \tag{8}$$

Then we generate optimal $\mathbf{R}^k$ below.

$$\mathbf{R}^k = \mathbf{Q}^k \mathbf{Y}^k (\mathbf{Q}^k)^T \tag{9}$$

By following the same approach, we produce an optimal matrix $\mathbf{S}^k$ for another Riemannian parameter matching problem $\min_{\mathbf{S}^k} d_{S_P}^2(\mathbf{V}^1, \mathbf{V}^k)$ s.t. $(\mathbf{S}^k)^T = \mathbf{S}^k$.

Therefore, the low-rank parameter matrices are aligned together. $\tilde{\mathbf{U}}^k$ and $\tilde{\mathbf{V}}^k$ are the aligned parameter matrices for client $k$.

$$\tilde{\mathbf{U}}^k = \mathbf{U}^k \mathbf{R}^k \approx \mathbf{U}^1, \quad \tilde{\mathbf{V}}^k = \mathbf{S}^k \mathbf{V}^k \approx \mathbf{V}^1 \tag{10}$$

To maintain parameter consistency before and after performing the Riemannian parameter matching, as well as the effectiveness of FFT-FM with rank-adaptive LoRA, i.e., $\tilde{\mathbf{U}}^k \tilde{\mathbf{\Sigma}}^k \tilde{\mathbf{V}}^k = \mathbf{U}^k \mathbf{\Sigma}^k \mathbf{V}^k = \Delta \mathbf{W}^k$, we need to derive a modified version $\tilde{\mathbf{\Sigma}}^k$ of $\mathbf{\Sigma}^k$.

$$\min_{\tilde{\mathbf{\Sigma}}^k} \|\tilde{\mathbf{U}}^k \tilde{\mathbf{\Sigma}}^k \tilde{\mathbf{V}}^k - \mathbf{U}^k \mathbf{\Sigma}^k \mathbf{V}^k\|_F^2 \tag{11}$$

However, it is difficult to directly solve the above optimization problem. We convert it to another equivalent problem below.

$$\min_{\tilde{\mathbf{\Sigma}}^k} \|(\mathbf{U}^k)^T \left(\tilde{\mathbf{U}}^k \tilde{\mathbf{\Sigma}}^k \tilde{\mathbf{V}}^k - \mathbf{U}^k \mathbf{\Sigma}^k \mathbf{V}^k\right)(\mathbf{V}^k)^T\|_F^2 \tag{12}$$

Since $\mathbf{U}^k$ and $\mathbf{V}^k$ are orthogonal matrices, the problem can be further transformed. Notice that $\tilde{\mathbf{U}}^k = \mathbf{U}^k \mathbf{R}^k$, $\tilde{\mathbf{V}}^k = \mathbf{S}^k \mathbf{V}^k$. Additionally, since $\mathbf{R}^k$ and $\mathbf{S}^k$ are non-singular matrices, an invertible matrix must exist.

$$\min_{\tilde{\mathbf{\Sigma}}^k} \|\mathbf{R}^k \tilde{\mathbf{\Sigma}}^k \mathbf{S}^k - \mathbf{\Sigma}^k\|_F^2 = \min_{\tilde{\mathbf{\Sigma}}^k} \|\tilde{\mathbf{\Sigma}}^k - (\mathbf{R}^k)^{-1} \mathbf{\Sigma}^k (\mathbf{S}^k)^{-1}\|_F^2 \tag{13}$$

The optimal solution is obtained when $\tilde{\mathbf{\Sigma}}_{\mathbf{ii}}^k = (\mathbf{R}_{\mathbf{ii}}^k)^{-1} \mathbf{\Sigma}_{\mathbf{ii}}^k (\mathbf{S}_{\mathbf{ii}}^k)^{-1}$. Next, the server can directly perform the aggregation on local low-rank parameter matrices to generate the global ones without the client-drift issue.

$$\frac{1}{K}\left(\mathbf{U}^1 + \sum_{k=2}^K \mathbf{U}^k \mathbf{R}^k\right) \times \frac{1}{K}\left(\mathbf{\Sigma}^1 + \sum_{k=2}^K \tilde{\mathbf{\Sigma}}^k\right) \times \frac{1}{K}\left(\mathbf{V}^1 + \sum_{k=2}^K \mathbf{S}^K \mathbf{V}^K\right)$$
$$= \mathbf{U}^1 \times \frac{1}{K}\left(\mathbf{\Sigma}^1 + \sum_{k=2}^K \tilde{\mathbf{\Sigma}}^k\right) \times \mathbf{V}^1 \approx \frac{1}{K}\left(\sum_{k=1}^K \mathbf{U}^k \mathbf{\Sigma}^k \mathbf{V}^k\right) = \frac{1}{K}\sum_{k=1}^K \Delta \mathbf{W}^k \tag{14}$$

Thus, when searching for the optimal rank on the globally low-rank parameter matrix, there is no need to physically generate full parameter matrix $\Delta \mathbf{W}$ and execute iterative SVD on $\Delta \mathbf{W}$. However, approximation errors may arise during the calculation process. The following theorem derives the bounded error introduced by approximation in Eq.(14).

**Theorem 1.** *We assume that the Frobenius norm of $\mathbf{R}^k$ is bounded by a constant $H$, and its eigenvalues lie within the interval $[\lambda_{R_{\min}}, \lambda_{R_{\max}}]$. Similarly, the Frobenius norm of $\mathbf{S}^k$ is bounded by a constant $N$, with eigenvalues in the range $[\lambda_{S_{\min}}, \lambda_{S_{\max}}]$. Additionally, the matrix $\Delta W$ has rank $r$. Given these conditions, the total approximation error bound in Eq.(14) is derived as follows*

$$\|\mathbf{U}^1 \times \frac{1}{K}\left(\mathbf{\Sigma}^1 + \sum_{k=2}^{K}\tilde{\mathbf{\Sigma}}^k\right) \times \mathbf{V}^1 - \frac{1}{K}\left(\sum_{k=1}^{K}\mathbf{U}^k\mathbf{\Sigma}^k\mathbf{V}^k\right)\|_F^2$$

$$= \sum_{k=2}^{K}\left(2r(1-H-N) + (1+(\frac{\lambda_{R_{\max}}}{\lambda_{R_{\min}}})^2)H^2 + (1+(\frac{\lambda_{S_{\max}}}{\lambda_{S_{\min}}})^2)N^2\right) \tag{15}$$

*Please refer to Appendix 7.2 for detailed proof of Theorem 1.*

By following the same idea in existing efforts Falini (2022); Wu et al. (2022), this paper utilizes singular value contribution rate $\Theta(r)$ to determine the rank $\tilde{\mathbf{\Sigma}}^{k'}$ after each round of FFT-FM. $\Theta(r)$ is defined as the cumulative percentage of the first $r$ singular values to all the singular values. We retain the first $r$ singular values such that $\Theta(r)$ is bigger than a threshold value $\varphi$.

$$\Theta(r) = \frac{\sum_{i=1}^{r}\tilde{\mathbf{\Sigma}}_{ii}'}{\sum_{i=1}^{r_{max}}\tilde{\mathbf{\Sigma}}_{ii}'} \geq \varphi \tag{16}$$

where $r_{max}$ is a maximum acceptable rank.

## 4 RIEMANNIAN GRADIENT DESCENT FOR ALLEVIATING RANK DRIFT ON CLIENTS

The data heterogeneity issue in the FFT-FM may bring the rank-drift challenges. When the server dispatches the low-rank parameter matrix with the appropriate rank $r$ to the clients, the rank of its local full parameter matrix $\Delta\mathbf{W}^k = \mathbf{U}^k\mathbf{\Sigma}^k\mathbf{V}^k$ on each client may significantly change after local training epoch, and so does the appropriate rank of the global parameter matrix. This may slow down the convergence of FFT-FM and degrade the model performance too. We develop a Riemannian gradient descent (RGD) method to guarantee the local full parameter matrices $\Delta\mathbf{W}^k$ in the form of low-rank ones $\mathbf{U}^k, \mathbf{\Sigma}^k$, and $\mathbf{V}^k$ with fixed appropriate rank determined by the server in each iteration of FFT-FM, to speed up the convergence of RAFFT.

By following AdaLoRA (Zhang et al., 2023b), we parameterize full parameter metrics on client $k$ as $\Delta\mathbf{W}^k = \mathbf{U}^k\mathbf{\Sigma}^k\mathbf{V}^k$, in order to avoid the expensive cost of iterative SVD on the large-scale full parameter metrics. We introduce a regularizer into the loss function, defined as $R(\mathbf{U}^k, \mathbf{V}^k)$ to enforce the orthogonality of $\mathbf{U}^k$ and $\mathbf{V}^k$.

$$R(\mathbf{U}^k, \mathbf{V}^k) = \|(\mathbf{U}^k)^T\mathbf{U}^k - \mathbf{I}\|_F^2 + \|\mathbf{V}^k(\mathbf{V}^k)^T - \mathbf{I}\|_F^2 \tag{17}$$

We introduce a Riemannian manifold containing all fixed-rank matrices, i.e., the set $M$ of all matrices with a fixed rank $r$: $M = \{\Delta\mathbf{W} : \Delta\mathbf{W} \in \mathbb{R}^{m\times n}, \text{rank}(\Delta\mathbf{W}) = r\}$ (do Carmo, 2018; Vandereycken, 2014). This forms a smooth submanifold of $\mathbb{R}^{m\times n}$. This structure ensures that the updated parameter matrices $\mathbf{U}^k\mathbf{\Sigma}^k\mathbf{V}^k$ stay within the manifold, maintaining rank invariance throughout the local update process. In terms of low-rank parameter metrics, the local objective function of FFT-FM on client $k$ is defined below.

$$\min_{\Delta\mathbf{W}^k \in M} L^k(\Delta\mathbf{W}^k) = L^k(\mathbf{U}^k, \mathbf{\Sigma}^k, \mathbf{V}^k) = \frac{1}{n_k}\sum_{\{x_i,y_i\}\in D^k} l_i(\mathbf{U}^k, \mathbf{\Sigma}^k, \mathbf{V}^k) + R(\mathbf{U}^k, \mathbf{V}^k) \tag{18}$$

The tangent space $T_M$ at the point $\Delta\mathbf{W}$ on the manifold $M$ is defined as follows.

$$T_M = \left\{(\mathbf{U}^k\ \mathbf{U}_\perp^k)\begin{bmatrix}\mathbb{R}^{r\times r} & \mathbb{R}^{r\times(n-r)} \\ \mathbb{R}^{(m-r)\times r} & 0^{(m-r)\times(n-r)}\end{bmatrix}(\mathbf{V}^k\ \mathbf{V}_\perp^k)^T\right\}$$

$$= \left\{\mathbf{U}^k\mathbf{G}(\mathbf{V}^k)^T + \mathbf{U}_p^k(\mathbf{V}^k)^T + \mathbf{U}^k(\mathbf{V}_p^k)^T\right.$$

$$\left. : \mathbf{G} \in \mathbb{R}^{r\times r}, \mathbf{U}_p^k \in \mathbb{R}^{m\times r}, (\mathbf{U}_p^k)^T\mathbf{U}^k = 0, \mathbf{V}_p^k \in \mathbb{R}^{n\times r}, (\mathbf{V}_p^k)^T\mathbf{V}^k = 0\right\} \tag{19}$$

where $\mathbf{U}_\perp^k$ and $\mathbf{V}_\perp^k$ denote the orthogonal complements of $\mathbf{U}^k$ and $\mathbf{V}^k$ respectively. $\mathbf{G}, \mathbf{U}_p^k, \mathbf{V}_p^k$ represent the tangent vector at $\Delta W$. Please refer to Appendix 7.3 for details of tangent space.

When the data have a manifold structure, traditional gradient descent methods in Euclidean space, such as SGD, often result in updates that stray off the manifold. Thus, we propose to project the gradient $\nabla L^k(\Delta\mathbf{W}^k)$ in Euclidean space onto the tangent space $T_M$ at the point $\Delta\mathbf{W}$ on the manifold $M$ and compute the Riemannian gradient $\Gamma L^k(\Delta\mathbf{W}^k)$. This projection can be accomplished

through orthogonal projection $P_{T_M}$, ensuring that the optimization trajectory remains confined to the manifold Lee (2012) and is effectively directed towards minimizing the loss function.

$$\Gamma L^k(\Delta \mathbf{W}^k) = (\Gamma_{\mathbf{U}^k} L^k, \Gamma_{\mathbf{\Sigma}^k} L^k, \Gamma_{\mathbf{V}^k} L^k) = P \nabla L^k \in T_M \subseteq \mathbb{R}^{m \times n}$$
$$= \mathbf{U}^k((\mathbf{U}^k)^T \nabla L^k \mathbf{V}^k)(\mathbf{V}^k)^T + (\mathbf{I}_m - \mathbf{U}^k(\mathbf{U}^k)^T) \nabla L^k \mathbf{V}^k (\mathbf{V}^k)^T + \quad (20)$$
$$\mathbf{U}^k(\mathbf{U}^k)^T \nabla L^k (\mathbf{I}_n - (\mathbf{V}^k)(\mathbf{V}^k)^T)$$

where $\Gamma_{\mathbf{U}^k} L^k$, $\Gamma_{\mathbf{\Sigma}^k} L^k$, and $\Gamma_{\mathbf{V}^k} L^k$ represent the Riemannian gradients with respect to $\mathbf{U}^k$, $\mathbf{\Sigma}^k$, and $\mathbf{V}^k$ respectively. $\mathbf{I}_m$ and $\mathbf{I}_n$ are the $m/n$-dimensional identity matrices. Please refer to Appendix 7.3 for derivation details of Riemannian gradient.

Therefore, the gradients of $\mathbf{U}^k$, $\mathbf{\Sigma}^k$, and $\mathbf{V}^k$ in Riemannian space can be derived as follows.

$$\Gamma_{\mathbf{U}^k} L^k = (\mathbf{I}_m - \mathbf{U}^k(\mathbf{U}^k)^T) \nabla_{\mathbf{U}^k} L^k, \qquad \Gamma_{\mathbf{V}^k} L^k = (\mathbf{I}_n - \mathbf{V}^k(\mathbf{V}^k)^T) \nabla_{\mathbf{V}^k} L^k,$$
$$\Gamma_{\mathbf{\Sigma}^k} L^k = ((\mathbf{U}^k)^T \nabla_{\mathbf{U}^k} L^k - \nabla_U(L^k)^T \mathbf{U}^k) \mathbf{\Sigma}^k + \mathbf{\Sigma}^k(\mathbf{V}^k \nabla_{\mathbf{V}^k}(L^k)^T - \nabla_{\mathbf{V}^k} L^k(\mathbf{V}^k)^T) + \nabla_{\mathbf{\Sigma}^k} L^k \quad (21)$$

where $\nabla_{\mathbf{U}^k} L^k$, $\nabla_{\mathbf{V}^k} L^k$, and $\nabla_{\mathbf{\Sigma}^k} L^k$ denote the corresponding Euclidean gradients. They are the conventional derivatives in Euclidean space that are used to compute the gradient descent steps before projection onto the manifold.

We try to find the steepest descent direction $P \nabla L^k$ within the tangent space $T_M$. Consider the intrinsic geometry of the manifold, we take a gradient step in the direction opposite to $P \nabla L^k$ and apply the learning rate $\eta_t$ to reach a new point $\Delta \mathbf{W}^k$, which still lies within the tangent space $T_M$. The retraction maps the tangent vector corresponding to the point $\Delta \mathbf{W}^k$ back to the manifold, ensuring that the rank of the updated point $\Delta \mathbf{W}^k$ remains unchanged. Namely, our RAFFT approach ensures that the local updates of $\mathbf{U}^k$, $\mathbf{\Sigma}^k$, and $\mathbf{V}^k$ preserve a fixed-rank structure. Thus, the approach keeps the rank invariant during the stage of local updates by maintaining the model parameters moving towards the Riemannian manifold.

The following theorem demonstrates that the RGD optimization on the Riemannian manifold ensures the rank invariance during the local update process.

**Theorem 2.** *Let $\Delta \mathbf{W}^k$ be a point on a Riemannian manifold $M$ consisting of all matrices with a fixed rank $r$ in $\mathbb{R}^{m \times n}$. Suppose $\nabla L^k(\Delta \mathbf{W}^k)$ is the Euclidean gradient of the loss function $L^k$ at $\Delta \mathbf{W}^k$, and $\Gamma_{\mathbf{U}^k} L^k$, $\Gamma_{\mathbf{\Sigma}^k} L^k$, and $\Gamma_{\mathbf{V}^k} L^k$ represent the components of the Riemannian gradient at $\Delta \mathbf{W}^k$. The RGD optimization with a learning rate $\eta_t$ at round $t$ ($0 \le t \le C$) ensures that the local update*

$$\left((\mathbf{U}^k)^{(t+1)}, (\mathbf{\Sigma}^k)^{(t+1)}, (\mathbf{V}^k)^{(t+1)}\right) = f\left(-\eta_t \left(\left(\Gamma_{\mathbf{U}^k} L^k\right)^{(t)}, \left(\Gamma_{\mathbf{\Sigma}^k} L^k\right)^{(t)}, \left(\Gamma_{\mathbf{V}^k} L^k\right)^{(t)}\right)\right) \quad (22)$$

*preserves the rank $r$ of $\Delta \mathbf{W}^k$, maintaining the structure within the manifold $M$, where $f$ is a retraction $f : T_M \to M$ mapping the tangent space at $\Delta \mathbf{W}^k$ to the manifold.*

*Please refer to Appendix 7.3 for the derivation of Riemannian retraction function and the detailed proof of Theorem 2.*

We also conduct the convergence analysis of our RAFFT algorithm based on the RGD optimization on the Riemannian manifold.

**Theorem 3** (Nonconvex). *Suppose that optimization problem in Eq.(18) satisfies Assumption 1 and 2. We run Algorithm 1, where the selected clients perform gradient descent with a fixed stepsize, and we set $\eta_t \le \frac{1}{c_g}$. Then, the output of Algorithm 1 satisfies the following*

$$\min_{0 \le t \le C} \mathbb{E}\left[\| \left(\left(\Gamma_{\mathbf{U}^k} L^k\right)^{(t)}, \left(\Gamma_{\mathbf{\Sigma}^k} L^k\right)^{(t)}, \left(\Gamma_{\mathbf{V}^k} L^k\right)^{(t)}\right) \|^2\right] \le \frac{2c_g}{C} \left(L(\Delta \mathbf{W}^1) - L(\Delta \mathbf{W}^*)\right) \quad (23)$$

*where $\Delta \mathbf{W}^*$ denotes the point on the manifold $\mathcal{M}$ that minimizes the function $L(\Delta \mathbf{W}^*) = \arg\min_{\Delta \mathbf{W} \in \mathcal{M}} L(\Delta \mathbf{W})$.*

**Theorem 4** (Convex). *Suppose that optimization problem in Eq.(18) satisfies Assumptions 1, 2, and 3, where the local functions $L$ are geodesically convex in $W$ (see Definition3). We consider Algorithm 1, where the selected clients perform gradient descent with a fixed stepsize, and the stepsize is set such that $\eta_t \le \frac{1}{2c_g}$. Then, the output of Algorithm 1 satisfies the following*

$$\mathbb{E}\left[L(\Delta \mathbf{W}^{(C)}) - L(\Delta \mathbf{W}^*)\right] \le \frac{\zeta c_g d^2(\Delta \mathbf{W}, \Delta \mathbf{W}^*)}{(\zeta + C - 2)} = \frac{\zeta c_g d^2(\Delta(\mathbf{U_0}, \mathbf{\Sigma_0}, \mathbf{V_0}), \Delta(\mathbf{U^*}, \mathbf{\Sigma^*}, \mathbf{V^*}))}{(\zeta + C - 2)}$$
$$(24)$$

*where $\Delta \mathbf{W}^*$ denotes the point on the manifold $\mathcal{M}$ that minimizes the function $L(\Delta \mathbf{W}^*) = \arg\min_{\Delta \mathbf{W} \in \mathcal{M}} L(\Delta \mathbf{W})$ and $\zeta$ is defined in Assumptions 3*

*Please refer to Appendix 7.4 for detailed proof of Theorems 3 and 4*

By assembling different pieces together, we provide the pseudo code of our RAFFT algorithm in Algorithm 1 in Appendix 7.9.

## 5 EXPERIMENTAL EVALUATION

In this section, we present the experimental results over 13 baselines and 3 commonly-used tasks to demonstrate the advantages of our method. We consider an FL environment with 100 devices and a parameter server. In each epoch, we randomly sample 10 devices to perform the local update. We exploit three widely used NLP tasks including SST-2 (Socher et al., 2013), MRPC Dolan & Brockett (2005) and MPQA (Wiebe et al., 2005). We evaluated fix rank method and all other methods on RoBERTa-LARGE (Liu et al., 2019b), which consists of 24 layers of transformers followed by a large language model head and 355M pretrained parameters. To demonstrate the adaptability of our method, we carried out extra experiments with two additional decoder-based models,i.e., LLaMA 3B Touvron et al. (2023) model on MRPC, MPQA, SST-2 dataset, LLaMA 7B model Radford et al. (2019) on MRPC, MPQA, SST-2 dataset. At the same time, we conducted experiments on the image classification data sets CIFAR10 (Krizhevsky et al., 2010), CIFAR100 (Krizhevsky et al., 2010), and Tiny-IamgeNet Le & Yang (2015) of the ViT model. The backbones of these models are frozen for all methods. Please see Appendix 7.5 for details.

|  | alpha = 5 | | | alpha = 3 | | | alpha = 1 | | |
|---|---|---|---|---|---|---|---|---|---|
| Method | Accuracy | Loss | Time | Accuracy | Loss | Time | Accuracy | Loss | Time |
| Cntralized LoRA | 80.40 | 0.3606 | 5,712 | 70.30 | 0.8222 | 5,762 | 71.00 | 0.8047 | 5,287 |
| AdaLoRA | 77.75 | 0.4070 | 5,450 | 78.70 | 0.3736 | 5,464 | 77.75 | 0.4070 | 5,421 |
| P-tuning v2 | 87.25 | 0.2512 | 5,717 | 88.90 | 0.2510 | 5,820 | 87.85 | 0.2779 | 5,450 |
| FedPrompt | 87.20 | 0.2556 | 5,636 | 88.80 | 0.2632 | 5,648 | 88.30 | 0.2917 | 5,450 |
| FedPepTAO | 87.60 | 0.2460 | 5,639 | 89.65 | 0.2526 | 5,651 | 88.70 | 0.2651 | 5,442 |
| PromptFL | 87.35 | 0.2507 | 5,670 | 89.40 | 0.2507 | 5,708 | 86.55 | 0.3115 | 5,453 |
| PE_FL | 87.00 | 0.2570 | 5,682 | 88.65 | 0.2643 | 5,678 | 87.90 | 0.2954 | 5,447 |
| SLoRA | 88.75 | 0.2474 | 5,296 | 87.90 | 0.2451 | 5,318 | 86.40 | 0.2878 | 5,481 |
| HetLoRA | 83.75 | 0.2964 | 5,057 | 86.25 | 0.2611 | 5,056 | 85.95 | 0.2518 | 5,038 |
| FedLoRA | 78.10 | 0.4356 | 4,996 | 77.50 | 0.3702 | 5,033 | 76.80 | 0.4629 | 5,019 |
| FFA-LoRA | 87.05 | 0.2167 | 5,000 | 87.45 | 0.2176 | 5,318 | 86.80 | 0.2249 | 5,317 |
| Fedkseed | 87.81 | 0.2662 | 5,641 | 88.77 | 0.2698 | 5,675 | 88.74 | 0.3022 | 5,447 |
| FLoRA | 85.60 | 0.2756 | 5,306 | 88.34 | 0.2594 | 5,310 | 87.69 | 0.2656 | 5,210 |
| RAFFT | 92.00 | 0.2079 | 5,010 | 91.55 | 0.2293 | 4,980 | 90.90 | 0.1934 | 4,965 |
| RAFFT-RGD | 88.35 | 0.2365 | 4,952 | 89.10 | 0.2514 | 4,955 | 88.90 | 0.2349 | 4,961 |
| RAFFT-MR | 90.80 | 0.2602 | 5,021 | 90.60 | 0.2501 | 4,917 | 86.05 | 0.2718 | 4,886 |

Table 1: Performance comparison of different methods on LLaMA 7B+MPQA.

**Baselines.** We take 13 existing approaches as baselines, including 2 stand-alone version lora method,i.e., **LoRA** (Hu et al., 2022), **AdaLoRA** (Zhang et al., 2023b), which significantly reduce the number of trainable parameters by injecting low-rank decomposition matrices into each layer of a Transformer model. We adapt both methods to a federated learning setting for comparison. 1 full parameters tuning method, i.e., **Fedkseed** (Qin et al., 2024), which uses a finite set of random seeds for zeroth-order optimization, reducing the communication overhead between the server and clients. 5 prompt-based tuning method, i.e., **P-Tuning v2** (Liu et al., 2021a), **FedPrompt** (Zhao et al., 2023), **FedPepTAO** (Che et al., 2023), **PromptFL** (Guo et al., 2024), **PE-FL** (Zhao et al., 2024), which integrate lightweight trainable blocks into the frozen foundational models (FMs) and fine-tune additional parameters for localized model adaptation, enhancing computational and communication efficiency while allowing customization based on specific local data characteristics or user preferences and 5 reparameterization-based methods, i.e., **SLoRA** (Babakniya et al., 2023), **HetLoRA** (Cho et al., 2023), **FedLoRA** (Yi et al., 2024), **FFA-LoRA** (Sun et al., 2024),**FLoRA** (Wang et al., 2024), which hypothesize that fine-tuning adaptations can be reparameterized into optimization within low-rank subspaces, addressing performance gaps due to data heterogeneity and optimizing performance across diverse client devices. For detailed descriptions of each baseline, please refer to the appendix 7.1.

**Variants of the RAFFT Model.** We evaluate two versions of the RAFFT model to showcase the strengths of distinct technical approaches. The first variant, which we term the RAFFT-RGD Model, solely utilizes the Riemannian gradient. This model primarily focuses on the fundamental capability to manage gradient descent within the geometric constraints of the RAFFT manifold, using FedAvg as the aggregation method. The second variant is the RAFFT-MR model, which does not use the Riemannian theory fix rank during training. This means rankings will drop during local updates.

| | alpha = 5 | | | alpha = 3 | | | alpha = 1 | | |
| --- | --- | --- | --- | --- | --- | --- | --- | --- | --- |
| Method | Accuracy | Loss | Time | Accuracy | Loss | Time | Accuracy | Loss | Time |
| Centralized LoRA | 71.90 | 0.7170 | 3,329 | 72.16 | 0.6129 | 3,395 | 67.08 | 0.6977 | 3,369 |
| AdaLoRA | 71.76 | 0.6936 | 3,535 | 71.35 | 0.6031 | 3,515 | 68.11 | 0.7818 | 3,529 |
| SLoRA | 71.75 | 0.6843 | 3,309 | 71.06 | 0.7023 | 3,380 | 68.14 | 0.7474 | 3,316 |
| HetLoRA | 71.76 | 0.6939 | 3,355 | 71.26 | 0.5861 | 3,331 | 65.96 | 0.6800 | 3,437 |
| FedLoRA | 71.56 | 0.7149 | 3,282 | 70.72 | 0.7149 | 3,384 | 68.19 | 0.6634 | 3,421 |
| FFA-LoRA | 71.47 | 0.7135 | 3,321 | 70.24 | 0.7369 | 3,321 | 65.76 | 0.6831 | 3,358 |
| FLoRA | 70.95 | 0.7266 | 3,401 | 71.09 | 0.6335 | 3,347 | 65.85 | 0.6890 | 3,361 |
| RAFFT | 74.54 | 0.5757 | 3,304 | 73.22 | 0.5765 | 3,281 | 70.04 | 0.6504 | 3,285 |
| RAFFT-RGD | 72.96 | 0.5945 | 3,288 | 72.52 | 0.6381 | 3,292 | 67.14 | 0.7159 | 3,299 |
| RAFFT-MR | 72.88 | 0.6070 | 3,291 | 72.02 | 0.6435 | 3,307 | 67.24 | 0.6879 | 3,311 |

Table 2: Performance comparison of different methods on ViT+CIFRA10.

**Evaluation metrics.** In line with protocols established in previous studies on federated learning for large language models (Babakniya et al., 2023; Cho et al., 2023; Sun et al., 2024), we employ three crucial metrics to evaluate the efficacy of parameter efficiency in fine-tuning for classification tasks: Loss, Accuracy, and Time.Loss measures the model's prediction error, providing insight into the effectiveness of the learning process. Accuracy gauges the model's performance on classifying new, unseen data, reflecting the practical applicability of the federated learning model. Time is evaluated to determine the speed of convergence and the overall computational demand, which are critical in federated settings where computational resources and time are often limited.

**Accuracy of classification with Riemannian Manifold.** Tables 1 and 2 show the classification accuracy of our method, which uses Riemannian parameter matching and a Riemannian optimizer across two datasets. For comparison, the baseline reflects the accuracy of FedPEFT. Our Non-IID dataset was partitioned using a Dirichlet distribution. Our method consistently outperforms 13 other approaches, addressing both "client drift" and "rank drift." Notably, we see accuracy improvements of up to 10.94% on MPQA for LLaMA-3B, 23.21% for LLaMA-7B, and 14.11% on SST-2 for RoBERTa. In image classification, the ViT model sees gains up to 16.61% on CIFAR-100. These results highlight the potential of our approach as a strong alternative to existing FedPEFT methods. For more details, see Appendix 7.5.

**Ablation study.** Tables 1-2 further compare the accuracy across three datasets with two variants of our RAFFT model, which differ in their aggregation methods and optimizer updates. Our observations reveal that our method consistently achieves optimal precision and minimal error on the SST-2, MRPC, and MPQA datasets, significantly outperforming the two variants. A plausible explanation for this superiority is that our RAFFT approach performs Riemannian parameter matching, effectively addressing "client drift" issues and thereby enhancing the efficacy of rank-adaptive LoRA in federated learning. Additionally, our method leverages the Riemannian Gradient Descent (RGD) technique to ensure that each iteration of federated learning by the server maintains locally updated weight matrices with a fixed optimal rank. This not only mitigates rank drift but also accelerates the aggregation process in RAFFT, enhancing the overall performance.

**Execution Time.** Tables 1-2 report the total running time for training large models across all clients for both variants of our RAFFT method and all comparison methods on three datasets. Compared to other FedPEFT methods that require operations on the full weight matrix during aggregation, our RAFFT approach achieves higher efficiency. Apart from these methods, our RAFFT method maintains similar efficiency while ensuring higher accuracy rates.

**Impact of number of client** Figure 2 (c,d) evaluates the performance effects with varying numbers of clients by changing N from 40 to 80. We observe that as N increases, the performance curve of RAFFT generally trends downward. Notably, when the number of clients reaches 80, the accuracy of RAFFT significantly decreases. A plausible explanation for this is that as the number of clients increases, the amount of data allocated to each client decreases, leading to lower quality local model

updates. Additionally, the increase in the number of clients exacerbates the heterogeneity issue. Data distributions may vary significantly across different clients, resulting in substantial differences in the model parameters trained by each client. This increased training complexity ultimately affects the model's performance. As the number of clients increases, the overall training time tends to increase. This is likely due to the higher communication overhead and synchronization delays introduced when coordinating a larger number of clients.

**Impact of initial rank** Based on Figure 2 (a,b), we evaluate the performance effect when setting the initial ranking value r to 2, 4, 8, and 16. Theoretically, the highest rank should yield the best performance. However, as ovserved in the experiments from the LoRA Hu et al. (2022) and AdaLoRA Zhang et al. (2023b) papers, there is no clear trend between performance and rank. Our results are consistent with this observation, showing that the best rank varies depending on the task. For example, rank 16 performed best for SST-2, MRPC, and MPQA, while rank 8 was optimal for CIFAR-10. Additionally, higher ranks increase training time due to greater computational complexity. When the ranking is higher, the model has more parameters to learn and optimize, which essentially requires more computing resources and time. Thus, different tasks require different ranks, and our approach is adaptive to the rank, allowing it to find the appropriate rank for each specific task.

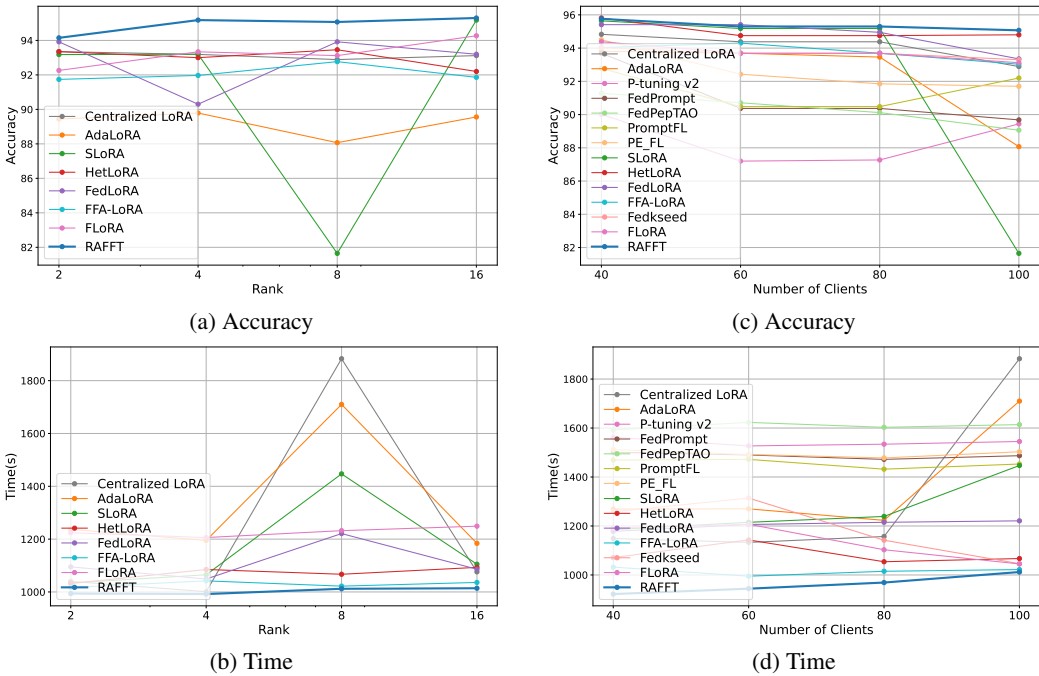

Figure 2: Accuracy and Time (s) with various LoRA ranks and client numbers on Roberta-SST2

**Impact of select rank threshold** Based on Figure Appendix 7.6, we evaluate the performance effect of setting the selection ranking threshold to 0.6, 0.7, 0.8, 0.9, and 1. We observe that as the threshold increases, the performance improves slightly initially and performs best at a threshold of 0.9. Beyond this point, performance starts to degrade. This trend suggests that a threshold of 0.9 is optimal, probably because it strikes a balance between filtering out irrelevant information and retaining useful information. When the threshold is set to 1, it means there is no level truncation which may introduce too much noise or redundant information, leading to performance degradation.

## 6 CONCLUSIONS

In this work, we propose rank-adaptive LoRA-based federated fine-tuning algorithm for foundation models. First, we design a novel Riemannian parameter matching method with the SVD on a low-rank matrix to avoid the client-drift issue to enhance the effectiveness of FFT-FM, while further improving its efficiency. Second, we develop a Riemannian gradient descent method on Riemannian manifold to guarantee the local parameter matrices with fixed optimal rank in each iteration, to speed up the convergence.

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

# 7 APPENDIX

## 7.1 RELATED WORK

**Parameter Efficient Fine Tuning (PEFT).** As the scale of large language models (LLMs) continues to grow, the cost associated with the standard full fine-tuning paradigm has become extremely high. To address this issue, parameter-efficient fine-tuning (PEFT) methods Houlsby et al. (2019) have been proposed. These methods introduce a minimal set of additional trainable parameters to enhance model performance while keeping the majority of the pre-trained parameters frozen. Generally, PEFT approaches can be classified into two main categories based on how they manipulate model parameters. The first category, additive PEFT, involves the incorporation of new trainable modules or parameters to adjust the model architecture, minimizing the parameter count for downstream tasks. This includes Adapter Tuning (Houlsby et al., 2019), where small, trainable adapter modules are inserted within the model architecture to allow for task-specific adaptation without altering the original model weights, and methods such as prefix-tuning (Devlin et al., 2019), P-tuning (Liu et al., 2022), P-tuning v2 (Liu et al., 2021a), and prompt tuning (Lester et al., 2021), which append learnable vectors at the beginning of input sequences. The second category involves reparameterized PEFT techniques that build low-rank representations of original model parameters, incorporating additional trainable parameters during training. This category includes Low-Rank Decomposition, techniques like LoRA Hu et al. (2022) and KronA (Edalati et al., 2022), which decompose original model parameters into low-rank and residual matrices, and LoRA derivatives such as AdaLoRA Zhang et al. (2023b) and SoRA (Liu et al., 2024b), which dynamically select the rank of LoRA to construct adaptable low-rank representations. Collectively, these methods represent the cutting-edge in efficient model training techniques, enabling the deployment of large language models in resource-constrained environments without sacrificing performance.

**Additive PEFT Methods in Federated Learning.** These methods are specifically designed to minimize computational demands and associated communication overheads. The additive methods integrate lightweight trainable blocks into the frozen foundational models (FMs) and fine-tune additional parameters for localized model adaptation. This approach not only enhances computational and communication efficiency but also allows for the customization of heterogeneous models based on specific local data characteristics or user preferences. Recent work has introduced Fed-CLIP, a method that leverages lightweight adapters to enhance CLIP's performance in federated

settings, significantly reducing computational and communication costs while ensuring fast generalization and personalization (Lu et al., 2023). Similarly, FedDAT has been developed to fine-tune foundation models in a multi-modal, heterogeneous federated learning environment, utilizing Dual-Adapter Teachers to efficiently handle data heterogeneity and improve knowledge transfer (Chen et al., 2024). The Client-Customized Adaptation approach further tailors federated learning to address client heterogeneity through hypernetworks that generate client-specific adapters, enhancing both the efficiency and efficacy of pre-trained model adaptation (Kim et al., 2023). In addressing the challenges of multilingual neural machine translation, a communication-efficient framework has been proposed that minimizes the need for heavy data transmission by transferring only lightweight adapters, thus maintaining performance despite substantial discrepancies in data distribution (Liu et al., 2023c). A practical approach toward federated learning in NLP, AdaFL1, identifies and configures adapter modules to optimize training efficiency and model performance across various NLP tasks (Cai et al., 2023a). Joint efforts in federated learning and personalization for on-device ASR have shown that integrating personalized adapters with federated training substantially reduces word error rates, demonstrating the effectiveness of combined federated learning and personalization strategies (Jia et al., 2023). Furthermore, the FedPETuning framework explores parameter-efficient tuning methods within federated learning, offering insights into privacy protection, performance efficiency, and resource constraints (Zhang et al., 2023c). Research into image reconstruction from PEFT gradients within federated learning frameworks reveals potential security vulnerabilities, suggesting that even lightweight adapter gradients can be exploited to reconstruct training data (Zhou et al., 2024). The novel concept of Dual-Personalizing Adapter for Federated Foundation Models introduces test-time personalization to address distribution shifts effectively, enhancing both global and local adaptation (Yang et al., 2024). FedPEAT combines emulator-assisted tuning with parameter-efficient fine-tuning within a federated context to tackle the challenges of deploying large foundation models efficiently (Chua et al., 2023). Adaptive model pruning within federated learning frameworks demonstrates a significant reduction in computation and communication latency, enabling efficient learning even over wireless networks (Liu et al., 2024a). The development of a communication-efficient federated learning framework for industrial human-robot interaction addresses data privacy and heterogeneity while reducing communication costs (Khalid et al., 2023). Moreover, new approaches in parameter-efficient fine-tuning aim to reduce latency and storage requirements without compromising the model's performance, indicating a shift towards more sustainable and efficient federated learning methodologies (Liao et al., 2023; Tobaben et al., 2023; Cai et al., 2023a; Shysheya et al., 2023). The other is promt tuning. prompt tuning incorporates trainable task-specific continuous cue vectors at the input layer (Liu et al., 2022). PromptFL Guo et al. (2024) ships an off-the-shelf CLIP, to distributed clients who would cooperatively train shared soft prompts based on very few local data. FedPerfix Sun et al. (2023a) use a local adapter to generate prefixes and aggregate raw self-attention layers.Efforts to mitigate communication overhead in deploying large pre-trained models across decentralized networks have introduced techniques like model split aggregation, which significantly reduce the parameter transmission cost while maintaining robust defenses against adversarial attacks (Zhao et al., 2023). Other studies have emphasized the reduction of communication costs by employing compact, federated visual prompts specifically in contexts like medical imaging, where data heterogeneity often leads to catastrophic forgetting (Feng et al., 2023; Li et al., 2023). The PromptFL framework exemplifies another novel approach, shifting from traditional model training to prompt-based federated learning, leveraging foundational models to improve local training efficiency on limited data (Guo et al., 2024). Concurrently, text-driven prompt generation methods have been developed to adapt vision-language models to federated settings, demonstrating superior generalization capabilities across seen and unseen classes (Qiu et al., 2024). Multilingual and domain-specific adaptations further extend the applicability of prompt tuning in federated learning, offering significant improvements in data efficiency and robustness across various languages, especially in low-resource scenarios (Zhao et al., 2024; Bai et al., 2024b; Deng et al., 2024). Meanwhile, research on personalization and robustness trade-offs in federated systems has explored the implications of localized fine-tuning on model performance, revealing that strategic prompt tuning can significantly enhance both the personalization and robustness of federated models under diverse conditions (Collins et al., 2023; Lin et al., 2023). Innovative methods such as Federated Black-Box Prompt Tuning have also been proposed to address privacy concerns associated with large pre-trained models, emphasizing parameter-efficient strategies that reduce memory and communication requirements without compromising the model effectiveness (Lin et al., 2023; Sun et al., 2023b). The exploration of federated prompt tuning continues to evolve, tackling not only linguistic and domain-specific challenges but also aiming at enhancing the deployment efficiency

and personalization of large-scale models across heterogeneous environments (Wang et al., 2023; Yu et al., 2023; Cai et al., 2023b; Che et al., 2023).

**Reparametrization-based PEFT Methods in Federated Learning.** The hypothesis behind this class of methods is that fine-tuning adaptions can be reparameterized into optimization within low-rank subspaces (Aghajanyan et al., 2021). Low-Rank Adaptation (LoRA) (Hu et al., 2022), as a popular PEFT method from the area of LLMs, reduces the number of trainable parameters for downstream tasks by representing the weight updates with two smaller matrices (called update matrices) through low-rank decomposition (Liu et al., 2023a). For instance, FedIT Zhang et al. (2024) leverages LoRA to improve the response quality of LLMs by utilizing diverse instructions from different clients. Noticeably, LoRA and its variants have also exhibited considerable potential in addressing the challenges inherent in data heterogeneity among clients in FL. Researchers have developed Heterogeneous LoRA, which introduces a novel approach by employing low-rank approximations with heterogeneous ranks across clients to balance overfitting risks and convergence speed, significantly enhancing federated fine-tuning efficiency and efficacy (Cho et al., 2023). Similarly, the SLoRA method tackles the challenges posed by high data heterogeneity in federated environments by employing a data-driven initialization technique that effectively bridges the performance gap between traditional fine-tuning and parameter-efficient methods (Babakniya et al., 2023). The application of LoRA in privacy-preserving federated settings has also been refined through Federated Freeze A LoRA (FFA-LoRA), which optimizes the stability and efficiency by adjusting the update mechanisms for low-rank matrices (Sun et al., 2024).FedRA introduces a random allocation strategy that leverages heterogeneous client capabilities to optimize federated tuning, highlighting an approach that can dynamically adapt to varying client resources without requiring each to support the full model (Su et al., 2023). The pFedLoRA framework emerges as a solution to model-heterogeneous personalized federated learning by using homogeneous small adapters for efficient knowledge exchange across diverse federated clients, proving to significantly outperform existing methods in terms of computational and communication overheads (Yi et al., 2024). Moreover, the DP-LoRA algorithm has been tailored for differentially private federated learning, enabling secure, efficient large language model fine-tuning across sensitive domains by incorporating a Gaussian noise mechanism for privacy preservation (Liu et al., 2023b). In scenarios with severe resource constraints, Low-Parameter Federated Learning (LP-FL) combines few-shot prompt learning and LoRA techniques to mitigate the high costs associated with large language models, demonstrating robust performance even in limited data environments (Jiang et al., 2023). The automated FedPipe system has also been developed to enhance the parameter-efficient fine-tuning of large language models, employing low-rank adapters and a novel parameter quantization strategy to optimize both training efficiency and memory usage (Fang et al., 2024). Lastly, FedMS introduces a novel two-stage federated learning algorithm that integrates a mixture of sparsely activated foundation models to cater to personalized needs, employing a unique activation strategy to manage computational demands effectively (Wu et al., 2023). Heroes presents a lightweight federated learning framework employing a novel neural composition and adaptive local update mechanism, tailored for heterogeneous edge networks. This method significantly reduces traffic consumption and enhances the speed of federated learning by adaptively configuring neural components and update frequencies according to the capabilities and resources of participating clients (Yan et al., 2023). Meanwhile, efforts to facilitate large-vocabulary neural language model training in resource-constrained environments incorporate Differential Privacy and Partial Embedding Updates. This approach, combined with Low-Rank Adaptation and Noise Contrastive Estimation, allows for effective language model training on compute-constrained devices while ensuring privacy and reducing memory demands (Xu et al., 2023).

**Rank-Adaptive Low-Rank Adaption for FFT-FM.** This category of algorithms aim to dynamically allocate the rank among low-rank parameter matrices based on their importance score. DyLoRA Valipour et al. (2023) addresses these issues by introducing dynamic, search-free rank adjustment during training, allowing for flexible adaptation across a range of ranks with minimal retraining and significantly accelerates training times across various tasks without substantial sacrifices in performance. QDyLoRA Rajabzadeh et al. (2024) evolves this concept into quantized settings, facilitating dynamic rank adjustments within a single training session on constrained hardware, thus maintaining competitive performance while optimizing hardware utilization efficiently. DP-DyLoRA Xu et al. (2024) integrates dynamic low-rank adaptation with differential privacy under federated learning settings to enhance privacy while mitigating typical performance degradation associated with such measures, making it feasible to deploy large models in privacy-sensitive environments. Structure-Aware Low-Rank Adaptation (SaLoRA) Hu et al. (2023) fine-tunes LLMs

by adapting the rank of updates based on the inherent structural properties of the model's layers, optimizing the allocation of trainable parameters to reflect varying importance across different model components. Sparse Low-rank Adaptation (SoRA) Ding et al. (2023) introduces an innovative method to dynamically adjust the intrinsic rank during the adaptation process using a gating mechanism that optimizes parameter sparsity, effectively balancing the trade-off between parameter efficiency and computational overhead. Hyperparameter Optimization for Large Language Model Instruction-Tuning Tribes et al. (2023)challenges optimizing LoRA's parameters through advanced blackbox optimization techniques to enhance the model's alignment with human-like processing abilities. ALoRA Liu et al. (2024c) pushes boundaries by dynamically allocating ranks during the fine-tuning process based on evaluated importance of different model components, thus allowing for more targeted and efficient adaptation tailored to specific downstream tasks. Lastly, IncreLoRA Zhang et al. (2023a) proposes an incremental parameter allocation method that adaptively adjusts ranks based on real-time assessments of module importance during training, showing great potential particularly in resource-constrained environments by optimizing parameter efficiency and model performance. Several pioneer rank-adaptive LoRA algorithms for FFT-FM iteratively aggregate local full parameter metrics into a global one and perform the SVD on the latter to find the optimal rank (Chai et al., 2022; Wu et al., 2021; Niu et al., 2023).

To our best knowledge, the common characteristics of the above rank-adaptive low-rank adaption methods is to (1) either fail to transplanting from the centralized setting to the federated environment (SVD-based rank-adaptive LoRA methods Zhang et al. (2023b); Ding et al. (2023); Zhang et al. (2023a)) due to the client-drift issue (Sun et al., 2024). (2) or raise extremely expensive cost in centralize learning and federated learning (rank sampling-based algorithms Valipour et al. (2023); Rajabzadeh et al. (2024); Xu et al. (2024) due to multi-time model fine-tuning at different ranks and federated rank-adaptive LoRA algorithms Chai et al. (2022); Wu et al. (2021); Niu et al. (2023) with iterative SVD on large-scale full parameter metrics). This work is the first to offer a rank-adaptive low-rank adaptation solution for FFT-FM with Riemannian manifold theory, by eliminating the client-drift issue and the multi-time model FT, and reducing the SVD of large-scale full parameter matrix to the SVD of a low-rank matrix with low dimensions, for further enhancing both effectiveness and efficiency of FFT-FM.

## 7.2 APPROXIMATION MATRIX UPPER BOUND

**Lemma 1** (Orthogonal Matrix and Frobenius Norm Preservation). *Let $U \in \mathbb{R}^{n \times n}$ be an orthogonal matrix, i.e., $U^\top U = I$. For any $A \in \mathbb{R}^{n \times m}$, the Frobenius norm $\|A\|_F$ satisfies:*

$$\|U^\top A\|_F = \|A\|_F. \tag{25}$$

*Please refer to Golub & Van Loan (2013) for the detailed proof.*

**Lemma 2** (Cauchy-Schwarz Inequality for Frobenius Norm). *For any $A, B \in \mathbb{R}^{n \times m}$, the Frobenius norms $\|A\|_F$ and $\|B\|_F$ satisfy:*

$$|tr(A^\top B)| \leq \|A\|_F \|B\|_F. \tag{26}$$

*where $tr(\cdot)$ is the trace of a matrix.*

*Please refer to Horn & Johnson (2012) for the detailed proof.*

**Lemma 3** (Condition Number and Off-Diagonal Error Amplification). *Let $S \in \mathbb{R}^{n \times n}$ be a symmetric positive definite matrix with condition number $\kappa(S) = \lambda_{\max}(S)/\lambda_{\min}(S)$. Then the Frobenius norm of the off-diagonal part of the inverse matrix is bounded by:*

$$\|S^{-1}_{\text{off-diagonal}}\|_F \leq \kappa(S) \|S_{\text{off-diagonal}}\|_F. \tag{27}$$

*where $S^{-1}$ is the inverse of the matrix $S$.*

*Please refer to Trefethen & Bau (1997) for the detailed proof.*

The following theorems provide upper bounds on the Frobenius norm error for off-diagonal elements after an orthogonal transformation.

**Theorem 1.** *We assume that the Frobenius norm of $\mathbf{R}^k$ is bounded by a constant $H$, and its eigenvalues lie within the interval $[\lambda_{R_{\min}}, \lambda_{R_{\max}}]$. Similarly, the Frobenius norm of $\mathbf{S}^k$ is bounded by a constant $N$, with eigenvalues in the range $[\lambda_{S_{\min}}, \lambda_{S_{\max}}]$. Additionally, the matrix $\Delta W$ has rank $r$. Given these conditions, the total approximation error bound in Equation 14 is derived as follows*

$$\|\mathbf{U}^1 \times \frac{1}{K}\left(\mathbf{\Sigma}^1 + \sum_{k=2}^{K} \tilde{\mathbf{\Sigma}}^k\right) \times \mathbf{V}^1 - \frac{1}{K}\left(\sum_{k=1}^{K} \mathbf{U}^k \mathbf{\Sigma}^k \mathbf{V}^k\right)\|_F^2$$
$$= \sum_{k=2}^{K}\left(2r(1 - H - N) + (1 + (\frac{\lambda_{R_{\max}}}{\lambda_{R_{\min}}})^2)H^2 + (1 + (\frac{\lambda_{S_{\max}}}{\lambda_{S_{\min}}})^2)N^2\right) \tag{28}$$

*Proof.* From Equation 14, we observe that the total approximation error consists of three components: the approximation error from the matrices $\mathbf{U}^k \mathbf{R}^k$, the approximation error from $\mathbf{V}^k \mathbf{S}^k$, and the approximation error due to the diagonal optimal solution of the $\tilde{\mathbf{\Sigma}}^k$ matrix. We calculate the error between the two components by computing the square of the Frobenius norm between them. These components can be expressed as follows:

$$\|\mathbf{U}^1 \times \frac{1}{K}\left(\mathbf{\Sigma}^1 + \sum_{k=2}^{K} \tilde{\mathbf{\Sigma}}^k\right) \times \mathbf{V}^1 - \frac{1}{K}\left(\sum_{k=1}^{K} \mathbf{U}^k \mathbf{\Sigma}^k \mathbf{V}^k\right)\|$$
$$= \sum_{k=2}^{K}(\|\mathbf{U}^1 - \mathbf{U}^k \mathbf{R}^k\|_F^2 + \|\mathbf{V}^1 - \mathbf{S}^k \mathbf{V}^k\|_F^2$$
$$+ \|(\mathbf{R_{ii}}^k)^{-1}\mathbf{\Sigma_{ii}}^k(\mathbf{S_{ii}}^k)^{-1} - (\mathbf{R}^k)^{-1}\mathbf{\Sigma}^k(\mathbf{S}^k)^{-1}\|_F^2) \tag{29}$$

The approximation error analysis for both $\mathbf{U}^k \mathbf{R}^k$ and $\mathbf{V}^k \mathbf{S}^k$ is similar. Therefore, we focus on $\mathbf{U}^k \mathbf{R}^k$ as an example. By expanding the Frobenius norm of the difference, we obtain the following expression:

$$\|\mathbf{U}^1 - \mathbf{U}^k \mathbf{R}^k\|_F^2 = \mathrm{tr}\left((\mathbf{U}^1)^T \mathbf{U}^1\right) - 2\mathrm{tr}\left((\mathbf{U}^1)^T \mathbf{U}^k \mathbf{R}^k\right) + \mathrm{tr}\left((\mathbf{R}^k)^T(\mathbf{U}^k)^T \mathbf{U}^k \mathbf{R}^k\right) \tag{30}$$

For the first term, since $\mathbf{U}^1$ is an orthogonal matrix, we have $(\mathbf{U}^1)^T \mathbf{U}^1 = I$, and thus $\mathrm{tr}(I) = r$. Similarly, since $U^k$ is an orthogonal matrix, the third term can be simplified as $tr(\mathbf{R}^k)^T \mathbf{R}^k)) = \|\mathbf{R}^k\|_F^2 \leq H^2$ For the second term, as shown in Equation (15), we have $-2\mathrm{tr}(\mathbf{P}^k \mathbf{\Lambda}^k (\mathbf{Q}^k)^T \mathbf{R}^k)$. Using the Cauchy-Schwarz inequality, we can further bound this term as follows:

$$\mathrm{tr}(\mathbf{\Lambda}^k(\mathbf{Q}^k)^T \mathbf{R}^k \mathbf{P}^k) \leq \|\mathbf{\Lambda}^k\|_F \cdot \|(\mathbf{Q}^k)^T \mathbf{R}^k \mathbf{P}^k\|_F \leq rH \tag{31}$$

Then we get approximation error between $\mathbf{U}^k \mathbf{R}^k$ and $\mathbf{U}^1$

$$\|\mathbf{U}^1 - \mathbf{U}^k \mathbf{R}^k\|_F^2 = r - 2rH + H^2 \tag{32}$$

Similarly, the approximation error between $\mathbf{V}^k \mathbf{S}^k$ and $\mathbf{V}^1$ be bounded as follows

$$\|\mathbf{V}^1 - \mathbf{S}^k \mathbf{V}^k\|_F^2 = r - 2rN + N^2 \tag{33}$$

The error between the optimal diagonal solution and the original complete matrix primarily arises from the off-diagonal elements. Since $\mathbf{\Sigma}^k$ is a diagonal matrix, the off-diagonal error is determined solely by the off-diagonal components of $\mathbf{R}^k$ and $\mathbf{S}^k$. Then we have

$$\|(\mathbf{R_{ii}}^k)^{-1}\mathbf{\Sigma_{ii}}^k(\mathbf{S_{ii}}^k)^{-1} - (\mathbf{R}^k)^{-1}\mathbf{\Sigma}^k(\mathbf{S}^k)^{-1}\|_F^2$$
$$= \sum_{i \neq j}((\mathbf{R}^k)_{ij}^{-1}(\mathbf{S}^k)_{ji}^{-1})^2 \leq \|(\mathbf{R}^k)_{ij}^{-1}\|_F^2\|(\mathbf{S}^k)_{ji}^{-1}\|_F^2 \tag{34}$$

Based on the condition number of the matrix, we know that the norm of the off-diagonal part of the inverse matrix can be amplified by at most $\kappa$ times the norm of the off-diagonal part of the original matrix. Since $\mathbf{R}^k$ and $\mathbf{S}^k$ are symmetric positive definite matrices, the condition number is determined by the ratio of the largest to smallest eigenvalues, then we get

$$\|(\mathbf{R_{ii}}^k)^{-1}\mathbf{\Sigma_{ii}}^k(\mathbf{S_{ii}}^k)^{-1} - (\mathbf{R}^k)^{-1}\mathbf{\Sigma}^k(\mathbf{S}^k)^{-1}\|_F^2 \leq (\frac{\lambda_{R_{\max}}}{\lambda_{R_{\min}}})^2 H^2 + (\frac{\lambda_{S_{\max}}}{\lambda_{S_{\min}}})^2 N^2 \tag{35}$$

$\square$

## 7.3 RIEMANN GRADIENT AND RETRACTION

**Lemma 4** (Partitioning and Submatrix Inversion). *Let $W \in \mathbb{R}^{m \times n}$ with rank $r$. If $w_{11} \in \mathbb{R}^{r \times r}$ is an invertible submatrix of $W$, then $W$ can be partitioned as:*

$$W = \begin{bmatrix} w_{11} & w_{12} \\ w_{21} & w_{22} \end{bmatrix} \tag{36}$$

*where $w_{12}$, $w_{21}$, and $w_{22}$ are submatrices. The submatrix $X \in \mathbb{R}^{r \times (n-r)}$ satisfies:*

$$X = w_{11}^{-1} w_{12}, \quad w_{22} = w_{21} X = w_{21} w_{11}^{-1} w_{12} \tag{37}$$

*Please refer to Boumal (2023) for detailed proof.*

**Lemma 5** (Differentiability of Local Defining Function). *Let $h : \mathbb{R}^{(m-r) \times (n-r)} \to \mathbb{R}^{m \times n}$ be defined as:*

$$h(Y) = Y_{22} - Y_{21} Y_{11}^{-1} Y_{12} \tag{38}$$

*where $Y_{ij}$ are submatrices of $Y$. Then $h$ is smooth and has an inverse mapping $h^{-1}$ such that $\ker(Dh(Y))$ is a linear subspace that spans $\mathbb{R}^{m \times n}$.*

*Please refer to Boumal (2023) for detailed proof.*

**Lemma 6** (Tangent Space on the Stiefel Manifold). *Let $\mathbf{U}^k \in \mathbb{R}^{m \times r}$ and $\mathbf{V}^k \in \mathbb{R}^{n \times r}$ lie on the Stiefel manifolds $St(m, r)$ and $St(n, r)$, respectively. For any $\Omega \in Skew(r)$ and $B \in \mathbb{R}^{(m-n) \times r}$, the tangent space velocities $\mathbf{U}^{k'}(0)$ and $\mathbf{V}^{k'}(0)$ are given by:*

$$\mathbf{U}^{k'}(0) = \mathbf{U}^k \Omega + \mathbf{U}_\perp^k B, \quad \mathbf{V}^{k'}(0) = \mathbf{V}^k \Omega' + \mathbf{V}_\perp^k C \tag{39}$$

*where $\mathbf{U}_\perp^k$ and $\mathbf{V}_\perp^k$ are orthogonal complements.*

*Please refer to Absil et al. (2008) for detailed proof.*

**Lemma 7** (Orthogonal Projection onto Tangent Space). *Let $\Delta W^k = U^k \Sigma^k (V^k)^\top$ represent a point on $M_r$. The orthogonal projection of a matrix $Z^k \in \mathbb{R}^{m \times n}$ onto the tangent space $T_{M_r}$ at $\Delta W^k$ is given by:*

$$Proj_{\Delta W^k}(Z^k) = U^k G (V^k)^\top + U_\perp^k (V_\perp^k)^\top + U^k (V_\perp^k)^\top \tag{40}$$

*where $G$ is a general matrix related to the variation of $U^k$ and $V^k$ along their tangent directions, $U_\perp^k$ and $V_\perp^k$ denote the components of $Z^k$ orthogonal to $U^k$ and $V^k$, respectively.*

*Please refer to Absil et al. (2008) for detailed proof.*

**Lemma 8** (Retraction for Fixed-Rank Matrices). *Let $H \in \mathbb{R}^{m \times n}$ represent the perturbation in the ambient space. The retraction function $f$ maps the tangent space $T_M$ to the manifold $M$ by minimizing the Frobenius norm distance:*

$$f(H) = \arg \min_{\Delta W \in M} \| \Delta W^k + H - \Delta W \|_F^2 \tag{41}$$

*Under the constraint of fixed rank $r$, the retraction is given by the truncated singular value decomposition of $\Delta W^k + H$.*

*Please refer to Golub & Van Loan (2013) for detailed proof.*

We define the set of weight parameter matrices of size $m \times n$ with rank $r$ as:

$$M_r = \{ \Delta \mathbf{W}^k \in \mathbb{R}^{m \times n} : \operatorname{rank}(\Delta \mathbf{W}^k) = r \} \tag{42}$$

which is an embedded submanifold of $\mathbb{R}^{m \times n}$. For any $\Delta \mathbf{W}^k \in \mathbb{R}_r^{m \times n}$, we define a continuous, smooth, local defining function. Given that the rank of $\Delta \mathbf{W}^k$ is $r$, it contains an invertible submatrix of size $r \times r$. Consequently, $\Delta \mathbf{W}^k$ can be partitioned as:

$$\Delta \mathbf{W}^k = \begin{bmatrix} \mathbb{R}^{r \times r} & \mathbb{R}^{r \times (n-r)} \\ \mathbb{R}^{(m-r) \times r} & \mathbb{R}^{(m-r) \times (n-r)} \end{bmatrix} \tag{43}$$

We assume that $w_{11} \in \mathbb{R}^{r \times r}$ is an invertible matrix, $w_{12} \in \mathbb{R}^{r \times (n-r)}$, $w_{21} \in \mathbb{R}^{(m-r) \times r}$, $w_{22} \in \mathbb{R}^{(m-r) \times (n-r)}$ Given a matrix $w \in \mathbb{R}^{m \times n}$ with rank $r$, its last $n-r$ columns are linear combinations of the first $r$ columns. This implies the existence of a matrix $X \in \mathbb{R}^{r \times (n-r)}$ such that:

$$\begin{bmatrix} w_{12} \\ w_{22} \end{bmatrix} = \begin{bmatrix} w_{11} \\ w_{21} \end{bmatrix} X \tag{44}$$

Here, $w_{11}$ is an invertible $r \times r$ matrix, and $w_{12}, w_{21}, w_{22}$ are submatrices of $X$ partitioned accordingly.

From this relationship, it follows that:

$$X = w_{11}^{-1} w_{12} \tag{45}$$

and consequently,

$$w_{22} = w_{21} X = w_{21} w_{11}^{-1} w_{12} \tag{46}$$

This decomposition shows that under the assumption of $w_{11}$ being invertible, the described relationship between the blocks of $w$ is both necessary and sufficient for $w$ to maintain a rank of $r$. Consider $U$ to be a subset of $\mathbb{R}^{m \times n}$ where the upper-left $r \times r$ submatrix is invertible. This subset $U$, being an open set in $\mathbb{R}^{m \times n}$, serves as the domain for the candidate local defining function $h$ given by:

$$h : U \to \mathbb{R}^{(m-r)\times(n-r)}, \quad Y = \begin{bmatrix} Y_{11} & Y_{12} \\ Y_{21} & Y_{22} \end{bmatrix} \mapsto Y_{22} - Y_{21}Y_{11}^{-1}Y_{12} \tag{47}$$

This mapping $h$ is smooth within $U$ and has the inverse mapping $h^{-1}(0) = \mathbb{R}^{m \times n} \cap U$, indicating that the pre-image of zero under $h$ intersects with $U$. Moreover, the differential of $h$ at any matrix $Y$ in $U$, for $V \in \mathbb{R}^{m \times n}$, is:

$$Dh(Y)[V] = V_{22} - V_{21}Y_{11}^{-1}Y_{12} + Y_{21}Y_{11}^{-1}V_{11}Y_{11}^{-1}Y_{12} - Y_{21}Y_{11}^{-1}V_{12} \tag{48}$$

utilizing the identity for differentiating a matrix inverse:

$$D(M \mapsto M^{-1})(M)[H] = -M^{-1}HM^{-1} \tag{49}$$

Please refer to the book Boumal (2023) for detailed proof.

The codomain of $Dh(Y)$ spans $\mathbb{R}^{(m-r)\times(n-r)}$, meaning any matrix in this space can be achieved by choosing an appropriate $V$. Setting $V_{11}, V_{12}, V_{21}$ to zero simplifies $Dh(Y)[V]$ to $V_{22}$, showing that $Dh(Y)$ is surjective. Thus, $h$ serves as a local defining function for the smooth submanifold around $w$ in $\mathbb{R}^{m \times n}$. If the top-left submatrix of size $r \times r$ is non-invertible, another local defining function can be constructed similarly for different submatrix choices.

These local defining functions collectively establish that $\mathbb{R}_r^{m \times n}$ is an embedded submanifold of $\mathbb{R}^{m \times n}$, endowed with dimension:

$$\dim \mathbb{R}_r^{m \times n} = \dim \mathbb{R}^{m \times n} - \dim \mathbb{R}^{(m-r)\times(n-r)} = mn - (m-r)(n-r) = r(m+n-r) \tag{50}$$

**Definition 1** (Tangent Spaces and Embedded Submanifolds). *Boumal (2023) Let $M$ be a subset of $\mathcal{E}$ and $w \in M$. The tangent space at $w$, denoted $T_w M$, consists of velocity vectors at $w$ of all smooth curves $c : I \to M$ that pass through $w$ at $t = 0$, where $I$ is an open interval containing $t = 0$. Formally,*

$$T_w M = \{c'(0) \mid c : I \to M \text{ is smooth and } c(0) = w\} \tag{51}$$

*For an embedded submanifold $M$ of $\mathcal{E}$, $T_w M$ coincides with the subspace defined by $\ker Dh(w)$ if $M$ is not open within $\mathcal{E}$, otherwise $T_w M = \mathcal{E}$. This structure highlights that $T_w M$ not only captures the linear approximation to $M$ at $w$ but also conforms to the embedding properties of submanifolds within $\mathcal{E}$.*

We recognize that each tangent space possesses a dimension as delineated in equation 50, it is sufficient to exhibit a linear subspace of that dimension which is included in the tangent space. Returning to the 19 of the tangent space , we undertake the explicit construction of smooth curves on $\mathbb{R}_r^{m \times n}$. We have $\Delta \mathbf{W}^k = \mathbf{U}^k \mathbf{\Sigma}^k (\mathbf{V}^k)^T$, define $U(t)$ as a smooth trajectory within $St(m, r)$ initiating at $U(0) = \mathbf{U}^k$, and similarly for $V(t)$ within $St(n, r)$ starting from $V(0) = \mathbf{V}^k$. Additionally, let $S(t)$ describe a trajectory within the invertible $r \times r$ matrices, forming an open submanifold, starting from $S(0) = \mathbf{\Sigma}^k$. Consequently, the curve $c(t) = U(t)S(t)V(t)^T$ forms a smooth trajectory in $\mathbb{R}_r^{m \times n}$ with $c(0) = \Delta \mathbf{W}^k$, where its initial velocity $c'(0)$ represents a tangent vector at $w$:

$$c'(0) = U'(0)\mathbf{\Sigma}^k(\mathbf{V}^k)^T + \mathbf{U}^k S'(0)(\mathbf{V}^k)^T + \mathbf{U}^k\mathbf{\Sigma}^k V'(0)^T \in T_w \mathbb{R}_r^{m \times n} \tag{52}$$

Given that $U(t)$ traverses smoothly through $\mathbf{U}^k$ on $St(m, r)$, the velocity $U'(0)$ lies within its tangent space at $\mathbf{U}^k$. For any vector within $T_U St(m, r)$, a corresponding smooth curve $U(t)$ can exhibit such velocity at $t = 0$. Referencing the specified relationship for the tangent space at $\Delta \mathbf{W}^k$ in the

Stiefel manifold Absil & Malick (2012) $T_{\Delta \mathbf{W}^k}St(n, p) = \{\Delta \mathbf{W}^k \Omega + \Delta \mathbf{W}_{\perp}^k B : \Omega \in \text{Skew}(p), B \in \mathbb{R}^{(n-p) \times p}\}$, it implies for any $\Omega \in \text{Skew}(p)$ and $B \in \mathbb{R}^{(n-p) \times p}$ that:

$$U'(0) = \mathbf{U}^k \Omega + \mathbf{U}_{\perp}^k B \tag{53}$$

where $\mathbf{U}_{\perp}^k$ ensures orthogonality with $\mathbf{U}^k$. Likewise, for any $\Omega' \in \text{Skew}(r)$ and $C \in \mathbb{R}^{(n-r) \times r}$, the expression for $V'(0)$ becomes:

$$V'(0) = \mathbf{V}^k \Omega' + \mathbf{V}_{\perp}^k C \tag{54}$$

with $\mathbf{V}_{\perp}^k$ maintaining orthogonality with $\mathbf{V}^k$. As $S(t)$ evolves smoothly within its manifold, the derivative $S'(0)$ can adopt any matrix configuration within $\mathbb{R}^{r \times r}$. with $\mathbf{V}_{\perp}^k$ such that $\mathbf{V}^k \mathbf{V}_{\perp}^k$ is orthogonal. Since $S(t)$ is a smooth curve on an open submanifold of $\mathbb{R}^{r \times r}$, we can choose $S'(0)$ to be any matrix $A \in \mathbb{R}^{r \times r}$. Finally, incorporating the relationships with $\mathbf{U}_{\perp}^k$ and $\mathbf{V}_{\perp}^k$, we observe that $U'(0)$ can be represented as $\mathbf{U}^k \Omega + \mathbf{U}_{\perp}^k B$ and $V'(0)$ as $\mathbf{V}^k \Omega' + \mathbf{V}_{\perp}^k C$, with $\Omega, \Omega' \in \text{Skew}(r)$ and $B, C \in \mathbb{R}^{(m-r) \times r}$. These adjustments ensure the continuation of $U(t)$ and $V(t)$ as smooth curves, affirming the orthogonality of $\mathbf{U}^k$ and $\mathbf{V}^k$ to their respective complements. Overall, this analysis demonstrates that the following velocities are within the tangent space of $\mathbb{R}_r^{m \times n}$ at $w$:

$$\begin{aligned} c'(0) &= (\mathbf{U}^k \Omega_2 + \mathbf{U}_{\perp}^k B)\mathbf{\Sigma}^k (\mathbf{V}^k)^T + \mathbf{U}^k \mathbf{\Sigma}^k ((\mathbf{V}^k)\Omega_2' + \mathbf{V}_{\perp}^k C)^T \\ &= \mathbf{U}^k(\Omega_2 + A - \mathbf{\Sigma}^k \Omega_2')(\mathbf{V}^k)^T + \mathbf{U}_{\perp}^k B(\mathbf{V}^k)^T + \mathbf{U}^k(\mathbf{V}_{\perp}^k C)^T \end{aligned} \tag{55}$$

with $A = \mathbf{\Sigma}^k \Omega_2 - \Omega_2 \mathbf{\Sigma}^k$, where $\Omega_2, \Omega_2' \in \text{Skew}(r)$, $B \in \mathbb{R}^{(m-r) \times r}$, and $C \in \mathbb{R}^{(n-r) \times r}$ are arbitrary matrices. Since $\mathbf{\Sigma}^k$ is invertible, it follows that any matrix of the form

$$\mathbf{U}^k M(\mathbf{V}^k)^T + \mathbf{U}_p^k(\mathbf{V}^k)^T + \mathbf{U}^k(\mathbf{V}_p^k)^T \tag{56}$$

with $M \in \mathbb{R}^{r \times r}$, $\mathbf{U}_p^k \in \mathbb{R}^{m \times r}$, $\mathbf{V}_p^k \in \mathbb{R}^{n \times r}$ such that $(\mathbf{U}^k)^T \mathbf{U}_p^k = (\mathbf{V}^k)^T \mathbf{V}_p^k = 0$ is tangent at $\Delta \mathbf{W}^k$. The conditions on $\mathbf{U}_p^k$ and $\mathbf{V}_p^k$ impose $2r^2$ linear constraints, thereby defining a linear subspace of $T_w \mathbb{R}_r^{m \times n}$ with dimension

$$r^2 + mr + nr - 2r^2 = r(m + n - r) \tag{57}$$

Then the singular value decomposition $(\mathbf{U}^k, \mathbf{\Sigma}^k, \mathbf{V}^k)$ of the matrix $\Delta \mathbf{W}^k$, the tangent space $T_{M_r}$ at the point $\Delta \mathbf{W}^k$ on the manifold $M_r$ can be represented as:

$$\begin{aligned} T_M &= \left\{ (\mathbf{U}^k\ \mathbf{U}_{\perp}^k) \begin{bmatrix} \mathbb{R}^{r \times r} & \mathbb{R}^{r \times (n-r)} \\ \mathbb{R}^{(m-r) \times r} & 0^{(m-r) \times (n-r)} \end{bmatrix} (\mathbf{V}^k\ \mathbf{V}_{\perp}^k)^T \right\} \\ &= \left\{ \mathbf{U}^k \mathbf{G}(\mathbf{V}^k)^T + \mathbf{U}_p^k(\mathbf{V}^k)^T + \mathbf{U}^k(\mathbf{V}_p^k)^T \right. \\ &\quad \left. : \mathbf{G} \in \mathbb{R}^{r \times r}, \mathbf{U}_p^k \in \mathbb{R}^{m \times r}, (\mathbf{U}_p^k)^T \mathbf{U}^k = 0, \mathbf{V}_p^k \in \mathbb{R}^{n \times r}, (\mathbf{V}_p^k)^T \mathbf{V}^k = 0 \right\} \end{aligned} \tag{58}$$

With $\mathbb{R}_r^{m \times n}$ still endowed with the standard inner product, we now consider the orthogonal projectors of $\mathbb{R}_r^{m \times n}$. From equation 19, the normal space at $\Delta \mathbf{W}^k = \mathbf{U}^k \mathbf{\Sigma}^k (\mathbf{V}^k)^T$ is defined as:

$$\mathbf{N}^k = \{\mathbf{U}^k \mathbf{\Sigma}^k (\mathbf{V}^k)^T : \Delta \mathbf{W}^k \in \mathbb{R}^{(m-r) \times (n-r)}\} \tag{59}$$

The orthogonal projection of a matrix $\mathbf{Z}^k \in \mathbb{R}^{m \times n}$ onto the tangent space $T_{\Delta \mathbf{W}^k} \mathbb{R}_r^{m \times n}$ can be expressed as:

$$\mathbf{Z}^k - \text{Proj}_{\Delta \mathbf{W}^k}(\mathbf{Z}^k) = \mathbf{U}^k \Delta \mathbf{W}_{\perp}^k (\mathbf{V}^k)^T \tag{60}$$

for some orthogonal complement $\Delta \mathbf{W}_{\perp}^k$, and subsequently,

$$\text{Proj}_{\Delta \mathbf{W}^k}(\mathbf{Z}^k) = \mathbf{U}^k \mathbf{G}(\mathbf{V}^k)^T + \mathbf{U}_{\perp}^k(\mathbf{V}_p^k)^T + \mathbf{U}^k(\mathbf{V}_p^k)^T \tag{61}$$

for matrices $\mathbf{G}, \mathbf{U}_p^k, \mathbf{V}_p^k$ such that $(\mathbf{U}^k)^T \mathbf{U}_p^k = (\mathbf{V}^k)^T \mathbf{V}_p^k = 0$. These matrices satisfy:

$$\mathbf{Z}^k = \mathbf{U}^k \mathbf{G}(\mathbf{V}^k)^T + \mathbf{U}_p^k(\mathbf{V}^k)^T + \mathbf{U}^k(\mathbf{V}_p^k)^T + \mathbf{U}_{\perp}^k \Delta \mathbf{W}_{\perp}^k (\mathbf{V}^k)_{\perp}^T \tag{62}$$

Define the projectors $P_U^k = \mathbf{U}^k(\mathbf{U}^k)^T$, $P_V^k = \mathbf{V}^k(\mathbf{V}^k)^T$, and their complements $P_{U_{\perp}}^k = I_m - P_U^k$, $P_{V_{\perp}}^k = I_n - P_V^k$. Then, we derive:

$$P_U^k P_V^k \mathbf{Z}^k = \mathbf{U}^k \mathbf{G}(\mathbf{V}^k)^T, \quad P_{U_{\perp}}^k P_V^k \mathbf{Z}^k = U_p^k(\mathbf{V}^k)^T, \quad \text{and} \quad P_U^k P_{V_{\perp}}^k \mathbf{Z}^k = \mathbf{U}^k(\mathbf{V}_p^k)^T \tag{63}$$

This leads to a reformulated projection:

$$\text{Proj}_{\Delta\mathbf{W}^k}(\mathbf{Z}^k) = P_U^k \mathbf{Z}^k P_V^k + P_{U_\perp}^k \mathbf{Z}^k P_V^k + P_U \mathbf{Z}^k P_{V_\perp}^k \tag{64}$$

yielding the complete orthogonal projection:

$$\text{Proj}_w(\mathbf{Z}^k) = \mathbf{U}^k((\mathbf{U}^k)^T \mathbf{Z}^k \mathbf{V}^k)(\mathbf{V}^k)^T + (I_m - \mathbf{U}^k(\mathbf{U}^k)^T)\mathbf{Z}^k \mathbf{V}^k(\mathbf{V}^k)^T + \mathbf{U}^k(\mathbf{U}^k)^T \mathbf{Z}^k(I_n - \mathbf{V}^k(\mathbf{V}^k)^T) \tag{65}$$

Here, $M = (\mathbf{U}^k)^T \mathbf{Z}^k \mathbf{V}^k$, $U_p^k = \mathbf{Z}^k \mathbf{V}^k - \mathbf{U}^k M$, and $V_p^k = (\mathbf{Z}^k)^T \mathbf{U}^k - \mathbf{V}^k M^T$, indicating that these components form a tangent vector at $w$ represented by the variation $\mathbf{Z}^k$.

Therefore, the gradient of $(\mathbf{U}^k, \mathbf{\Sigma}^k, \mathbf{V}^k)$ in Riemannian space can be expressed as:

$$\Gamma_{\mathbf{U}^k} L^k = (\mathbf{I}_m - \mathbf{U}^k(\mathbf{U}^k)^T)\nabla_{\mathbf{U}^k} L^k, \qquad \Gamma_{\mathbf{V}^k} L^k = (\mathbf{I}_n - \mathbf{V}^k(\mathbf{V}^k)^T)\nabla_{\mathbf{V}^k} L^k,$$

$$\Gamma_{\mathbf{\Sigma}^k} L^k = ((\mathbf{U}^k)^T \nabla_{\mathbf{U}^k} L^k - \nabla_U(L^k)^T \mathbf{U}^k)\mathbf{\Sigma}^k + \mathbf{\Sigma}^k(\mathbf{V}^k \nabla_{\mathbf{V}^k}(L^k)^T - \nabla_{\mathbf{V}^k} L^k(\mathbf{V}^k)^T) + \nabla_{\mathbf{\Sigma}^k} L^k \tag{66}$$

According to the theorem2, we can ensure that the rank remains unchanged during the local training process.

**Theorem 2.** *Let $\Delta\mathbf{W}^k$ be a point on a Riemannian manifold $M$ consisting of all matrices with a fixed rank $r$ in $\mathbb{R}^{m \times n}$. Suppose $\nabla L^k(\Delta\mathbf{W}^k)$ is the Euclidean gradient of the loss function $L^k$ at $\Delta\mathbf{W}^k$, and $\Gamma_{\mathbf{U}^k} L^k$, $\Gamma_{\mathbf{\Sigma}^k} L^k$, and $\Gamma_{\mathbf{V}^k} L^k$ represent the components of the Riemannian gradient at $\Delta\mathbf{W}^k$. The RGD optimization with a learning rate $\eta_t$ at round $t$ ensures that the local update*

$$\left((\mathbf{U}^k)^{(t+1)}, (\mathbf{\Sigma}^k)^{(t+1)}, (\mathbf{V}^k)^{(t+1)}\right) = f\left(-\eta_t \left(\left(\Gamma_{\mathbf{U}^k} L^k\right)^{(t)}, \left(\Gamma_{\mathbf{\Sigma}^k} L^k\right)^{(t)}, \left(\Gamma_{\mathbf{V}^k} L^k\right)^{(t)}\right)\right) \tag{67}$$

*preserves the rank $r$ of $\Delta\mathbf{W}^k$, maintaining the structure within the manifold $M$, where $f$ is a retraction $f : T_M \to M$ mapping the tangent space at $\Delta\mathbf{W}^k$ to the manifold.*

*Proof.* To construct a retraction $f$ for updating within a Riemannian manifold, we utilize metric projection. This method involves making a step in the ambient space, denoted by $H$, and then projecting back to the manifold to minimize the distance in the Frobenius norm, ensuring that the updated matrix retains the same rank as the original. The retraction function is formulated as:

$$f(H) = \arg\min_{\Delta\mathbf{W}^k \in \mathbb{R}_{m \times n}^r} \|\Delta\mathbf{W}^k + H - \Delta\mathbf{W}^k\|_F^2 \tag{68}$$

According to the Eckart-Young-Mirsky theorem Golub et al. (1987), the optimal solution to this optimization problem, under the constraint of rank $r$, is achieved by the truncated singular value decomposition of $\Delta\mathbf{W}^k + H$. According to quation 19, the $\Delta\mathbf{W}^k + H$ can be represented as:

$$\Delta\mathbf{W}^k + H = (\mathbf{U}^k)^{(t)}((\mathbf{\Sigma}^k)^{(t)} + (-\eta_t \left(\Gamma_{\mathbf{\Sigma}^k} L^k\right)^{(t)}))((\mathbf{V}^k)^{(t)})^T + U_p((\mathbf{V}^k)^{(t)})^T + (\mathbf{U}^k)^{(t)} V_p^T$$

$$= [(\mathbf{U}^k)^{(t)} \quad U_p] \begin{bmatrix} ((\mathbf{\Sigma}^k)^{(t)} + (-\eta_t \left(\Gamma_{\mathbf{\Sigma}^k} L^k\right)^{(t)})) & I_r \\ I_r & 0 \end{bmatrix} [(\mathbf{V}^k)^{(t)} \quad V_p]^T. \tag{69}$$

Where $I_r$ is the $r$-dimensional identity matrix. QR factorizations of the augmented matrices $(\mathbf{U}^k)^{(t)}$ and $U_p$, as well as $(\mathbf{V}^k)^{(t)}$ and $V_p$, are given by:

$$Q_U R_U = [(\mathbf{U}^k)^{(t)} U_p], \quad Q_V R_V = [(\mathbf{V}^k)^{(t)} V_p] \tag{70}$$

Here, $Q_U$ and $Q_V$ represent the orthogonal matrices resulting from the QR factorizations of the left and right matrices, ensuring that columns are orthonormal. $R_U$ and $R_V$ are upper triangular matrices which correspond to the R components in the QR factorization process.

Employing the QR factorizations, the perturbed matrix $W + H$ is expressed as:

$$\Delta\mathbf{W}^k + H = Q_U R_U \begin{bmatrix} ((\mathbf{\Sigma}^k)^{(t)} + (-\eta_t \left(\Gamma_{\mathbf{\Sigma}^k} L^k\right)^{(t)})) & I_r \\ I_r & 0 \end{bmatrix} R_V^T Q_V^T. \tag{71}$$

By performing singular value decomposition on the middle part, we obtain $\tilde{U}\tilde{\Sigma}\tilde{V}^T$, where $\tilde{U}$, $\tilde{V}$, and $\tilde{\Sigma}$ represent the orthogonal matrices and diagonal matrix of singular values, respectively.

The retraction $f(H)$ of the perturbation $H$ at the point $\Delta \mathbf{W}^k$ on the manifold is given by:

$$f(H) = (Q_U \tilde{U}) \tilde{\Sigma} (Q_V \tilde{V})^T \tag{72}$$

This expression concludes that the triplet $(Q_U \tilde{U}, \tilde{\Sigma}, Q_V \tilde{V})$ represents the retracted point on the rank-$r$ manifold $\mathbb{R}_r^{m \times n}$. The final outcome of our optimization process on the manifold, representing the updated components for the next iteration, is:

$$\left( (\mathbf{U}^k)^{(t+1)}, (\mathbf{\Sigma}^k)^{(t+1)}, (\mathbf{V}^k)^{(t+1)} \right) = (Q_U \tilde{U}, \tilde{\Sigma}, Q_V \tilde{V}) \tag{73}$$

$\square$

### 7.4 CONVERGENCE ON RIEMANNIAN MANIFOLD

**Definition 2** (L-smoothness on manifolds). *A function $L$ is said to be Lipschitz smooth on a manifold $\mathcal{M}$ if there exists a constant $c \geq 0$ such that the following inequality holds:*

$$\|grad \, L(y) - P_{y \to x} grad \, L(x)\| \leq cd(x, y), \tag{74}$$

*where $P_{y \to x}$ is the parallel transport operator along the geodesic connecting $y$ and $x$, and $d(x, y)$ is the geodesic distance between $x$ and $y$.*

*For a complete Riemannian manifold, the following inequality holds:*

$$L(y) \leq L(x) + \langle g_x, Exp_x^{-1}(y) \rangle_x + \frac{L_g}{2} d^2(x, y), \quad \forall x, y \in \mathcal{M} \tag{75}$$

*Please refer to the paper Zhang & Sra (2016) for detailed proof.*

**Definition 3** (Geodesic convexity). *A function $L \in C^1(\mathcal{M})$ is said to be geodesically convex if for all $x, y \in \mathcal{M}$, there exists a geodesic $\gamma$ such that $\gamma(0) = x$, $\gamma(1) = y$, and:*

$$f(\gamma(t)) \leq (1 - t)L(x) + tL(y), \quad \forall t \in [0, 1] \tag{76}$$

*Or equivalently,*

$$L(y) \geq L(x) + \langle grad \, L(x), Exp_x^{-1}(y) \rangle_x \tag{77}$$

*Please refer to the paper Zhang & Sra (2016) for detailed proof.*

**Assumption 1.** *For all client $k$, we assume that the function $L_i$ is geodesically $c_L$-Lipschitz continuous. Therefore, the function $l$ is also geodesically $c_L$-Lipschitz continuous.*

**Assumption 2.** *For all client $k$, we assume that the function $L_i$ is geodesically $c_g$-smooth, which implies that the function $L$ is geodesically $c_g$-smooth.*

**Assumption 3.** *We assume that the manifold under consideration is complete and that there exists a compact subset $\mathcal{W} \subset \mathcal{M}$ with diameter bounded by $M$, such that all the iterates of Algorithm 1 and the optimal points lie within $\mathcal{W}$. The sectional curvature of $\mathcal{W}$ is bounded within the interval $[\kappa_{\min}, \kappa_{\max}]$. Furthermore, we define the following key geometric constant that captures the impact of the manifold's curvature (Zhang et al., 2017)*

$$\zeta = \begin{cases} \frac{\sqrt{|\kappa_{\min}|} M}{\tanh\left(\sqrt{|\kappa_{\min}|} M\right)} & \text{if } \kappa_{\min} < 0, \\ 1 & \text{if } \kappa_{\min} \geq 0. \end{cases} \tag{78}$$

**Theorem 3** (Nonconvex). *Suppose that optimization problem in Eq.(18) satisfies Assumption 1 and 2. We run Algorithm 1, where the selected clients perform gradient descent with a fixed stepsize, and we set $\eta_t \leq \frac{1}{c_g}$. Then, the output of Algorithm 1 satisfies the following*

$$\min_{0 \leq t \leq C} \mathbb{E}\left[ \| \left( (\Gamma_{\mathbf{U}^k} L^k)^{(t)}, (\Gamma_{\mathbf{\Sigma}^k} L^k)^{(t)}, (\Gamma_{\mathbf{V}^k} L^k)^{(t)} \right) \|^2 \right] \leq \frac{2c_g}{C} \left( L((\Delta \mathbf{W})^1) - L((\Delta \mathbf{W}^*)) \right)$$
$$\tag{79}$$

*where $x^*$ denotes the point on the manifold $\mathcal{M}$ that minimizes the function $f(x) x^* = \arg\min_{x \in \mathcal{M}} L(x)$.*

*Proof.* According to Equation 14, our aggregation method is equivalent to the aggregation of the matrix $\Delta \mathbf{W}$. However, in Euclidean space, it is challenging to accurately measure the distance between two points before and after the update. Therefore, following the approach from [12, 13], we project $\Delta \mathbf{W}$ back to the tangent space using the inverse retraction for aggregation. To measure the distance between the updated point $x_{t+1}$ and the previous point $x_t$, we use the following distance formula

$$d(x_{t+1}, x_t) = \left\| \mathrm{f}_{x_t}^{-1}(x_{t+1}) \right\| = \left\| \frac{1}{k} \sum_{i \in S_t} \mathrm{f}_{x_t}^{-1}(x^{(i)}) \right\| \tag{80}$$

According to the update rule of formula 22 we have:

$$(\Delta \mathbf{W}^k)^{(t+1)} = \left( (\mathbf{U}^k)^{(t+1)}, (\mathbf{\Sigma}^k)^{(t+1)}, (\mathbf{V}^k)^{(t+1)} \right)$$

$$= f \left( -\eta_t \left( \left( \Gamma_{\mathbf{U}^k} L^k \right)^{(t)}, \left( \Gamma_{\mathbf{\Sigma}^k} L^k \right)^{(t)}, \left( \Gamma_{\mathbf{V}^k} L^k \right)^{(t)} \right) \right) \tag{81}$$

$$f_{(\Delta \mathbf{W}^k)^{(t)}}^{-1} \left( (\Delta \mathbf{W}^k)^{(t+1)} \right) = \left( -\eta_t \left( \left( \Gamma_{\mathbf{U}^k} L^k \right)^{(t)}, \left( \Gamma_{\mathbf{\Sigma}^k} L^k \right), \left( \Gamma_{\mathbf{V}^k} L^k \right)^{(t)} \right) \right)$$

Using Lipschitz smooth ($c_g$-smooth) of $\mathbf{L_i}$ based on Definition 2, we have

$$L((\Delta \mathbf{W}^k)^{(t+1)}) - L((\Delta \mathbf{W}^k)^{(t)}) \le \left\langle f_{(\Delta \mathbf{W}^k)^{(t)}}^{-1} \left( (\Delta \mathbf{W}^k)^{(t+1)} \right), \left( \left( \Gamma_{\mathbf{U}^k} L^k \right)^{(t)}, \left( \Gamma_{\mathbf{\Sigma}^k} L^k \right), \left( \Gamma_{\mathbf{V}^k} L^k \right)^{(t)} \right) \right\rangle$$

$$+ \frac{c_g}{2} d^2((\Delta \mathbf{W}^k)^{(t+1)}, (\Delta \mathbf{W}^k)^{(t)})$$

$$= \left\langle \frac{1}{K} \sum_{k=1}^{K} f_{(\Delta \mathbf{W}^k)^{(t)}}^{-1} \left( (\Delta \mathbf{W}^k)^{(t+1)} \right), \left( \left( \Gamma_{\mathbf{U}^k} L^k \right)^{(t)}, \left( \Gamma_{\mathbf{\Sigma}^k} L^k \right), \left( \Gamma_{\mathbf{V}^k} L^k \right)^{(t)} \right) \right\rangle$$

$$+ \frac{c_g}{2} \left\| \frac{1}{k} \sum_{k=1}^{K} f_{(\Delta \mathbf{W}^k)^{(t)}}^{-1} \left( (\Delta \mathbf{W}^k)^{(t+1)} \right) \right\|^2$$

$$= -\eta_t \| \left( \left( \Gamma_{\mathbf{U}^k} L^k \right)^{(t)}, \left( \Gamma_{\mathbf{\Sigma}^k} L^k \right)^{(t)}, \left( \Gamma_{\mathbf{V}^k} L^k \right)^{(t)} \right) \|^2$$

$$+ \frac{\eta_t^2 c_g}{2} \| \left( \left( \Gamma_{\mathbf{U}^k} L^k \right)^{(t)}, \left( \Gamma_{\mathbf{\Sigma}^k} L^k \right)^{(t)}, \left( \Gamma_{\mathbf{V}^k} L^k \right)^{(t)} \right) \|^2 \tag{82}$$

Then we have

$$\mathbb{E} \left[ L((\Delta \mathbf{W}^k)^{(t+1)}) - L((\Delta \mathbf{W}^k)^{(t)}) \right] \le -\eta_t \mathbb{E} \left[ \| \left( \left( \Gamma_{\mathbf{U}^k} L^k \right)^{(t)}, \left( \Gamma_{\mathbf{\Sigma}^k} L^k \right)^{(t)}, \left( \Gamma_{\mathbf{V}^k} L^k \right)^{(t)} \right) \|^2 \right]$$

$$+ \mathbb{E} \left[ \frac{\eta_t^2 c_g}{2} \| \left( \left( \Gamma_{\mathbf{U}^k} L^k \right)^{(t)}, \left( \Gamma_{\mathbf{\Sigma}^k} L^k \right)^{(t)}, \left( \Gamma_{\mathbf{V}^k} L^k \right)^{(t)} \right) \|^2 \right] \tag{83}$$

By taking $\eta_t \le \frac{1}{c_g}$, then we have

$$\mathbb{E} \left[ L((\Delta \mathbf{W}^k)^{(t+1)}) - L((\Delta \mathbf{W}^k)^{(t)}) \right] \le -\frac{1}{c_g} \mathbb{E} \left[ \| \left( \left( \Gamma_{\mathbf{U}^k} L^k \right)^{(t)}, \left( \Gamma_{\mathbf{\Sigma}^k} L^k \right)^{(t)}, \left( \Gamma_{\mathbf{V}^k} L^k \right)^{(t)} \right) \|^2 \right]$$

$$+ \mathbb{E} \left[ \frac{(\frac{1}{c_g})^2 c_g}{2} \| \left( \left( \Gamma_{\mathbf{U}^k} L^k \right)^{(t)}, \left( \Gamma_{\mathbf{\Sigma}^k} L^k \right)^{(t)}, \left( \Gamma_{\mathbf{V}^k} L^k \right)^{(t)} \right) \|^2 \right]$$

$$= -\frac{1}{2c_g} \| \left( \left( \Gamma_{\mathbf{U}^k} L^k \right)^{(t)}, \left( \Gamma_{\mathbf{\Sigma}^k} L^k \right)^{(t)}, \left( \Gamma_{\mathbf{V}^k} L^k \right)^{(t)} \right) \|^2 \tag{84}$$

Summing this inequality over t from 1 to C, we have

$$\frac{1}{2c_g} \sum_{t=1}^{C} \| \left( \left( \Gamma_{\mathbf{U}^k} L^k \right)^{(t)}, \left( \Gamma_{\mathbf{\Sigma}^k} L^k \right)^{(t)}, \left( \Gamma_{\mathbf{V}^k} L^k \right)^{(t)} \right) \|^2 \le L((\Delta \mathbf{W})^1) - L((\Delta \mathbf{W})^R) \tag{85}$$

$$\le L((\Delta \mathbf{W})^1) - L((\Delta \mathbf{W}^*))$$

$\square$

**Theorem 4** (Convex). *Suppose that optimization problem in Eq.(18) satisfies Assumptions 1, 2, and 3, where the local functions L are geodesically convex in W (see Definition 3). We consider Algorithm 1, where the selected clients perform gradient descent with a fixed stepsize, and the stepsize is set such that $\eta_t \le \frac{1}{2c_g}$. Then, the output of Algorithm 1 satisfies the following*

$$\mathbb{E}\left[L(\Delta \mathbf{W}^{(C)}) - L(\Delta \mathbf{W}^*)\right] \le \frac{\zeta c_g d^2(\Delta \mathbf{W}, \Delta \mathbf{W}^*)}{(\zeta + C - 2)} = \frac{\zeta c_g d^2(\Delta(\mathbf{U_0}, \mathbf{\Sigma_0}, \mathbf{v_0}), \Delta(\mathbf{U}^*, \mathbf{\Sigma}^*, \mathbf{v}^*))}{(\zeta + C - 2)} \tag{86}$$

*where $x^*$ denotes the point on the manifold $\mathcal{M}$ that minimizes the function $f(x)$ $x^* = \arg\min_{x \in \mathcal{M}} L(x)$ and $\zeta$ is defined in Assumptions 3*

*Proof.*

**Lemma 9** ( Zhang & Sra (2016)[Corollary 8]). *For any Riemannian manifold $\mathcal{M}$ where the sectional curvature is lower bounded by $\kappa_{\min}$ and any point $x, x^{(t)} \in \mathcal{M}$, the update $x^{(t+1)} = Exp_{x^{(t)}}(-\alpha g^{(t)})$ with $g^{(t)} \in T_{x^{(t)}}\mathcal{M}$ satisfies*

$$\left\langle -g^{(t)}, Exp_{x^{(t)}}^{-1}(x) \right\rangle \le \frac{1}{2\alpha}\left(dist^2(x, x^{(t)}) - dist^2(x, x^{(t+1)})\right) + \frac{\zeta(\kappa_{\min}, dist(x, x^{(t)}))\alpha}{2}\|g^{(t)}\|^2, \tag{87}$$

*Where $Exp_{x^{(t)}}(x)$ is the exponentail map that projects a tangent vctor $x$ onto the manifold, $Exp_{x^{(t)}}^{-1}(x)$ is the inverse exponential map that maps a point $x$ back to the tangent space, $\alpha$ is the learning rate controlling the magnitude of the update, $dist(x, x^{(t+1)})$ denotes the geodesic distance between points $x$ and $x^{(t+1)}$ on the manifold, and $\zeta(\kappa, c) = \frac{\sqrt{|\kappa|c}}{\tanh\left(\sqrt{|\kappa|c}\right)}$ is the curvature adjustment for distance.*

From Lemma 9 we have

$$\left\langle \frac{1}{K}\sum_{k=1}^{K} f_{(\Delta \mathbf{W})^t}^{-1}\left((\Delta \mathbf{W}^k)\right), f_{(\Delta \mathbf{W})^t}^{-1}\left((\Delta \mathbf{W})\right) \right\rangle \le \frac{1}{2}\left(d^2(\Delta \mathbf{W})^t, \Delta \mathbf{W})) - d^2(\Delta \mathbf{W})^{(t+1)}, \Delta \mathbf{W}))\right)$$

$$+ \frac{\zeta}{2}\left\|\frac{1}{k}\sum_{k=1}^{K} f_{(\Delta \mathbf{W})^t}^{-1}\left((\Delta \mathbf{W}^k)\right)\right\|^2 \tag{88}$$

According to the formula 82 we get

$$-\eta_t \left\langle \left(\frac{1}{K}\sum_{k=1}^{K}\left(\left(\Gamma_{\mathbf{U}^k}L^k\right)^{(t)}, \left(\Gamma_{\mathbf{\Sigma}^k}L^k\right)^{(t)}, \left(\Gamma_{\mathbf{V}^k}L^k\right)^{(t)}\right)\right), f_{(\Delta \mathbf{W})^t}^{-1}\left((\Delta \mathbf{W})\right) \right\rangle \tag{89}$$

$$\le \frac{1}{2}\left(d^2(\Delta \mathbf{W})^t, \Delta \mathbf{W})) - d^2(\Delta \mathbf{W})^{(t+1)}, \Delta \mathbf{W}))\right) + \frac{\zeta}{2}\left\|\frac{1}{k}\sum_{k=1}^{K} f_{(\Delta \mathbf{W})^t}^{-1}\left((\Delta \mathbf{W}^k)\right)\right\|^2$$

Based on the geodesic convexity of $L_i$ and inequality 89, we define $\Delta_t = L(\Delta \mathbf{W})^t) - L(\Delta \mathbf{W}^*)$ and $\Delta_t^k = L_k(\Delta \mathbf{W})^t) - L_k(\Delta \mathbf{W}^*)$, and we have:

$$\Delta_t^k \le -\langle \left(\left(\Gamma_{\mathbf{U}^k}L^k\right)^{(t)}, \left(\Gamma_{\mathbf{\Sigma}^k}L^k\right)^{(t)}, \left(\Gamma_{\mathbf{V}^k}L^k\right)^{(t)}\right), f_{(\Delta \mathbf{W})^t}^{-1}(\Delta \mathbf{W}^*)\rangle. \tag{90}$$

Summing this inequality over $k = 1, \ldots, K$, we get:

$$\Delta_t \le -\frac{1}{n}\left\langle \left(\frac{1}{K}\sum_{k=1}^{K}\left(\left(\Gamma_{\mathbf{U}^k}L^k\right)^{(t)}, \left(\Gamma_{\mathbf{\Sigma}^k}L^k\right)^{(t)}, \left(\Gamma_{\mathbf{V}^k}L^k\right)^{(t)}\right)\right), f_{(\Delta \mathbf{W})^t}^{-1}(\Delta \mathbf{W}^*) \right\rangle$$

$$\le \frac{1}{2\eta}\left(d^2((\Delta \mathbf{W})^t, \Delta \mathbf{W}^*)) - d^2((\Delta \mathbf{W})^{(t+1)}, \Delta \mathbf{W}^*))\right) + \frac{\zeta}{2\eta}\left\|\frac{1}{k}\sum_{k=1}^{K} f_{(\Delta \mathbf{W})^t}^{-1}\left((\Delta \mathbf{W}^k)\right)\right\|^2$$

$$\le \frac{1}{2\eta}\left(d^2((\Delta \mathbf{W})^t, \Delta \mathbf{W}^*)) - d^2((\Delta \mathbf{W})^{(t+1)}, \Delta \mathbf{W}^*))\right) + \frac{\zeta\eta}{2n}\| \left(\left(\Gamma_{\mathbf{U}^k}L^k\right)^{(t)}, \left(\Gamma_{\mathbf{\Sigma}^k}L^k\right)^{(t)}, \left(\Gamma_{\mathbf{V}^k}L^k\right)^{(t)}\right)\|^2. \tag{91}$$

From Inequality82 we have

$$\Delta_{(t+1)} - \Delta_t \le (-\eta + \frac{\eta^2 c_g}{2})\| \left( \left(\Gamma_{\mathbf{U}^k} L^k\right)^{(t)}, \left(\Gamma_{\mathbf{\Sigma}^k} L^k\right)^{(t)}, \left(\Gamma_{\mathbf{V}^k} L^k\right)^{(t)} \right) \|^2 \tag{92}$$

we multiply 92 by $\zeta$ and add it into 91, then we get:

$$\zeta\Delta_{(t+1)} - (\zeta-1)\Delta_t \le \zeta\left( \frac{\eta}{2n} - \eta + \frac{\eta^2 c_g}{2} \right) \| \left( \left(\Gamma_{\mathbf{U}^k} L^k\right)^{(t)}, \left(\Gamma_{\mathbf{\Sigma}^k} L^k\right)^{(t)}, \left(\Gamma_{\mathbf{V}^k} L^k\right)^{(t)} \right) \|^2$$
$$+ \frac{1}{2\eta}\left( d^2((\Delta\mathbf{W})^t, \Delta\mathbf{W}^*)) - d^2((\Delta\mathbf{W})^{(t+1)}, \Delta\mathbf{W}^*)) \right) \tag{93}$$

We take $\eta \le \frac{1}{2c_g}$ and $\left( \frac{\eta}{2n} - \eta + \frac{\eta^2 c_g}{2} \right) \le 0$, thus

$$\zeta\Delta_{(t+1)} - (\zeta-1)\Delta_t \le \frac{1}{2\eta}\left( d^2((\Delta\mathbf{W})^t, \Delta\mathbf{W}^*)) - d^2((\Delta\mathbf{W})^{(t+1)}, \Delta\mathbf{W}^*)) \right) \tag{94}$$

Summing this up over t from 0 to C-1 we get

$$\zeta\Delta_C + \sum_{t=0}^{C-1} \Delta_t \le (\zeta-1)\Delta_1 + \frac{d^2(\Delta\mathbf{W}, \Delta\mathbf{W}^*)}{2\eta} \le \frac{\zeta d^2(\Delta\mathbf{W}, \Delta\mathbf{W}^*)}{2\eta} \tag{95}$$

Where the last inequality follows from $\eta \le \frac{1}{2c_g}$ and $\Delta^0 \le c_g dist(\Delta\mathbf{W}, \Delta\mathbf{W}^*)$. Taking expectation for 92 then we get

$$(\zeta + C - 2)\mathbb{E}[\Delta_C] \le \frac{\zeta d^2(\Delta\mathbf{W}, \Delta\mathbf{W}^*)}{2\eta} \le \zeta c_g d^2(\Delta\mathbf{W}, \Delta\mathbf{W}^*) \tag{96}$$

thus

$$\Delta_C \le \frac{\zeta c_g d^2(\Delta\mathbf{W}, \Delta\mathbf{W}^*)}{(\zeta + C - 2)} = \frac{\zeta c_g d^2(\Delta(\mathbf{U_0}, \mathbf{\Sigma_0}, \mathbf{V_0}), \Delta(\mathbf{U}^*, \mathbf{\Sigma}^*, \mathbf{V}^*))}{(\zeta + C - 2)} \tag{97}$$

$\square$

### 7.5 EXPERIMENTAL DETAILS

**Environment.** The experiments were conducted on a compute server running on Red Hat Enterprise Linux 7.2 with 2 CPUs of Intel Xeon E5-2650 v4 (at 2.66 GHz) and 8 GPUs of NVIDIA GeForce GTX 2080 Ti (with 11 GB of GDDR6 on a 352-bit memory bus and memory bandwidth in the neighborhood of 620GB/s) and 4 GPUs of NVIDIA H100 (each with 80GB of HBM2e memory on a 5120-bit memory bus, offering a memory bandwidth of approximately 3TB/s),256GB of RAM, and 1TB of HDD. Overall, the experiments took about 10 days in a shared resource setting. We expect that a consumer-grade single-GPU machine could complete the full set of experiments in around 21-23 days, if its full resources were dedicated. The codes were implemented in Python 3.7.10 and PyTorch 1.9.0. Since the datasets used are all public datasets and our methodologies and the hyperparameter settings are explicitly described in section 5 and 7.5, our codes and experiments can be easily reproduced on top of a GPU server.

**Training.** We study text classification model on three standard text datasets: SST-2 Socher et al. (2013),MRPC Dolan & Brockett (2005) and MPQA (Wiebe et al., 2005). The above three text datasets are all public datasets, which allow researchers to use non-commercial research and educational purposes. We use 7606 examples as training data and 1000 examples as test data for MPQA. We use 3668 examples as training data and 408 as test data for MRPC. We use 20000 examples as training data and 872 examples as test data for SST-2. We train LLaMA-3B,LLaMA-7B and Roberta on this three dataset for text classification. We also train a ViT on CIFAR-10, CIFAR-100 and Tiny-imagnet for image classification. The neural networks are trained with Kaiming initialization using RGD for 50 epochs with an initial learning rate of 4e-4 and batch size 8. In addition, we run each experiment for 3 trials for obtaining more stable results.

**Implementation.** For 11 state-of-the-art federated large language models of LoRA Hu et al. (2022), AdaLoRA Zhang et al. (2023b), P-Tuningv2 Liu et al. (2021a), FedPrompt Zhao et al. (2023), Fed-PepTAO Che et al. (2023), PromptFL Guo et al. (2024),PE-FL Zhao et al. (2024), SLoRA Babakniya

et al. (2023), HetLoRA Cho et al. (2023), FedLoRA Yi et al. (2024)and FFA-LoRA Sun et al. (2024), we utilized the same model architecture as the official open-source implementation and default parameter settings provided by the original authors for FedPEFT in all experiments. All hyperparameters are standard values from reference codes or prior works. We validate the performance of different FedPEFT methods with a range of rank $\in \{2, 4, 8, 16\}$. All models were trained for 10, 25 and 50 epochs, with a batch size of 8, and a learning rate of 4e-4. We use the Dirichlet distribution with concentration parameters $\alpha \in \{1, 3, 5\}$ to partition the data into non-IID splits. Each device is then assigned a certain number of samples based on the corresponding Dirichlet distribution. The above open-source codes from the GitHub are licensed under the MIT License, which only requires preservation of copyright and license notices and includes the permissions of commercial use, modification, distribution, and private use.

For our RAFFT model, we performed hyperparameter selection by performing a parameter rank $r \in \{2, 4, 8, 16\}$, Dirichlet alpha $\alpha \in \{1, 3, 5\}$, training epochs of the FedPEFT model $\in \{10, 25, 50, 100\}$, select rank and learning rate$\in \{1e^{-4}, 3e^{-4}, 4e^{-4}, 5e^{-5}\}$. We select the best parameters over 50 epochs of training and evaluate the model at test time.

**Hyperparameter settings.** Unless otherwise explicitly stated, we used the following default parameter settings in the experiments. As shown in Table 37

| Parameter | Value |
|---|---|
| Training data on SST-2 | 20,000 |
| Test data ratio on SST-2 | 872 |
| Training data on MRPC | 3,668 |
| Test data on MRPC | 408 |
| Training data on MPQA | 7,606 |
| Test data on MPQA | 1,000 |
| Training data on CIFRA10 | 50,000 |
| Test data on CIFRA10 | 10,000 |
| Training data on CIFRA100 | 5,000 |
| Test data on CIFRA100 | 1,000 |
| Training data on Tiny-ImageNet | 100,000 |
| Test data on Tiny-ImageNet | 10,000 |
| Select rank threshold $\alpha$ | 0.9 |
| Training epochs of the FedPEFT model | 50 |
| Batch size for training the model | 8 |
| Learning rate | 4e-4 |

Table 3: Model parameters and settings

## 7.6 ADDITIONAL EXPERIMENTS

In this section, we present additional experimental results beyond those described in Section 3 to demonstrate the advantages of our proposed method. We considered a Federated Learning (FL) environment with 100 devices and a parameter server, randomly sampling 10 devices in each epoch to perform local updates. We utilized three widely-used NLP tasks, including SST-2 Socher et al. (2013), MRPC Dolan & Brockett (2005), and MPQA (Wiebe et al., 2005). Evaluations were conducted on the RoBERTa-LARGE Liu et al. (2019b) model for the MRPC and SST-2 datasets, the LLaMA 3B Touvron et al. (2023) model for the MRPC and SST-2 datasets, and the LLaMA 7B Radford et al. (2019) model for the MRPC and SST-2 datasets. Additionally, we experimented with the ViT model on image classification datasets CIFAR-10 and CIFAR-100. For all methods, the backbone models remained frozen. These comprehensive evaluations across multiple models and datasets illustrate the robustness and effectiveness of our fixed rank approach compared to 11 baseline methods across three common tasks.

| Method | alpha = 5 | | | alpha = 3 | | | alpha = 1 | | |
|---|---|---|---|---|---|---|---|---|---|
| | Accuracy | Loss | Time | Accuracy | Loss | Time | Accuracy | Loss | Time |
| Centralized LoRA | 89.65 | 0.2510 | 1,442 | 89.85 | 0.2857 | 1,419 | 86.25 | 0.3776 | 1,433 |
| AdaLoRA | 85.85 | 0.4134 | 1,774 | 82.00 | 2.2912 | 1,748 | 81.95 | 2.3407 | 1,763 |
| P-tuning v2 | 87.45 | 0.3563 | 1,581 | 86.80 | 0.3256 | 1,614 | 86.50 | 0.3509 | 1,623 |
| FedPrompt | 86.00 | 0.5352 | 1,463 | 85.45 | 0.4666 | 1,479 | 84.80 | 0.4345 | 1,537 |
| FedPepTAO | 89.20 | 0.2854 | 1,608 | 89.05 | 0.2545 | 1,565 | 88.45 | 0.2749 | 1,592 |
| PromptFL | 85.75 | 0.5358 | 1,696 | 83.75 | 0.4847 | 1,516 | 83.35 | 0.4236 | 1,498 |
| PE_FL | 85.50 | 0.5331 | 1,663 | 85.25 | 0.4669 | 1,568 | 83.35 | 0.4274 | 1,505 |
| SLoRA | 90.05 | 0.2545 | 1,473 | 89.75 | 0.2546 | 1,446 | 86.70 | 0.2579 | 1,488 |
| HetLoRA | 87.70 | 0.3377 | 1,207 | 87.15 | 0.3159 | 1,212 | 87.25 | 0.3227 | 1,239 |
| FedLoRA | 89.95 | 0.2473 | 1,416 | 89.95 | 0.2416 | 1,517 | 89.60 | 0.2492 | 1,399 |
| FFA-LoRA | 87.50 | 0.3580 | 1,193 | 86.95 | 0.3386 | 1,168 | 87.30 | 0.3443 | 1,166 |
| Fedkseed | 81.90 | 0.6226 | 1,193 | 82.35 | 0.5921 | 1,186 | 82.75 | 0.5833 | 1,176 |
| FLoRA | 82.25 | 0.5549 | 1,216 | 83.35 | 0.5286 | 1,213 | 84.25 | 0.5227 | 1,154 |
| RAFFT | 91.45 | 0.1692 | 905 | 91.60 | 0.1701 | 902 | 91.20 | 0.1602 | 905 |
| RAFFT-RGD | 90.05 | 0.2311 | 869 | 89.85 | 0.2392 | 894 | 89.35 | 0.2281 | 890 |
| RAFFT-MR | 90.15 | 0.2299 | 890 | 90.10 | 0.2288 | 885 | 89.50 | 0.2209 | 884 |

Table 4: Performance comparison of different methods on Roberta+MPQA.

| Method | alpha = 5 | | | alpha = 3 | | | alpha = 1 | | |
|---|---|---|---|---|---|---|---|---|---|
| | Accuracy | Loss | Time | Accuracy | Loss | Time | Accuracy | Loss | Time |
| Centralized LoRA | 88.40 | 0.2249 | 3,351 | 88.65 | 0.2203 | 3,339 | 88.00 | 0.2362 | 3,229 |
| AdaLoRA | 83.60 | 0.3318 | 3,134 | 84.85 | 0.3343 | 3,136 | 83.45 | 0.3179 | 3,137 |
| P-tuning v2 | 88.60 | 0.4830 | 3,360 | 86.85 | 0.3565 | 3,360 | 88.50 | 0.4606 | 3,358 |
| FedPrompt | 86.70 | 0.4537 | 3,358 | 87.70 | 0.4110 | 3,387 | 87.70 | 0.4110 | 3,354 |
| FedPepTAO | 88.90 | 0.4001 | 3,345 | 87.20 | 0.3535 | 3,335 | 87.70 | 0.5189 | 3,354 |
| PromptFL | 88.90 | 0.4000 | 3,352 | 88.50 | 0.2580 | 3,352 | 88.85 | 0.4520 | 3,360 |
| PE_FL | 88.25 | 0.3837 | 3,357 | 88.60 | 0.2580 | 3,353 | 88.60 | 0.2580 | 3,356 |
| SLoRA | 84.55 | 0.3028 | 3,893 | 81.35 | 0.3215 | 3,912 | 81.73 | 0.3579 | 3,851 |
| HetLoRA | 88.80 | 0.2685 | 2,848 | 88.95 | 0.2501 | 2,842 | 88.60 | 0.2310 | 2,842 |
| FedLoRA | 84.55 | 0.3028 | 2,792 | 81.35 | 0.3215 | 2,801 | 81.65 | 0.3579 | 2,805 |
| FFA-LoRA | 85.25 | 0.2552 | 2,789 | 84.35 | 0.2745 | 2,813 | 84.15 | 0.2776 | 2,811 |
| Fedkseed | 88.81 | 0.3979 | 3,356 | 88.83 | 0.2865 | 3,937 | 88.55 | 0.4165 | 3,356 |
| FLoRA | 87.05 | 0.3469 | 2,915 | 84.80 | 0.3443 | 2,957 | 84.84 | 0.4316 | 3,105 |
| RAFFT | 91.50 | 0.2769 | 2,718 | 91.35 | 0.2330 | 2,728 | 91.00 | 0.2418 | 2,754 |
| RAFFT-RGD | 90.15 | 0.3123 | 2,681 | 89.50 | 0.2930 | 2,678 | 88.90 | 0.2741 | 2,683 |
| RAFFT-MR | 89.95 | 0.2962 | 2,699 | 88.90 | 0.2899 | 2,701 | 88.70 | 0.2775 | 2,691 |

Table 5: Performance comparison of different methods on LLaMA 3B+MPQA.

| Method | alpha = 5 | | | alpha = 3 | | | alpha = 1 | | |
|---|---|---|---|---|---|---|---|---|---|
| | Accuracy | Loss | Time | Accuracy | Loss | Time | Accuracy | Loss | Time |
| Centralized LoRA | 81.32 | 0.4122 | 1,407 | 80.77 | 0.4217 | 1,411 | 80.70 | 0.4519 | 1,411 |
| AdaLoRA | 81.13 | 0.4309 | 1,563 | 80.29 | 0.4373 | 1,554 | 79.16 | 0.4818 | 1,570 |
| P-tuning v2 | 78.48 | 0.4325 | 1,049 | 79.41 | 0.4116 | 1,057 | 77.59 | 0.3935 | 1,064 |
| FedPrompt | 75.87 | 0.6784 | 1,004 | 75.87 | 0.6346 | 979 | 74.98 | 3.8650 | 1,022 |
| FedPepTAO | 81.83 | 0.3691 | 1,066 | 80.24 | 0.3577 | 1,053 | 81.21 | 0.3722 | 1,066 |
| PromptFL | 75.84 | 0.5917 | 1,001 | 75.97 | 0.7182 | 977 | 75.29 | 0.7722 | 1,022 |
| PE_FL | 76.18 | 0.5868 | 1,024 | 74.98 | 3.8320 | 1,052 | 75.45 | 0.6071 | 1,022 |
| SLoRA | 80.43 | 0.4559 | 1,044 | 80.81 | 0.4152 | 1,052 | 81.29 | 0.4075 | 1,043 |
| HetLoRA | 80.52 | 0.4470 | 1,086 | 80.33 | 0.4404 | 1,168 | 79.51 | 0.3834 | 1,146 |
| FedLoRA | 81.64 | 0.5644 | 1,124 | 82.99 | 0.4428 | 1,204 | 82.21 | 0.4141 | 1,193 |
| FFA-LoRA | 77.74 | 0.4846 | 1,149 | 78.22 | 0.4395 | 1,026 | 77.02 | 0.4180 | 1,070 |
| Fedkseed | 81.34 | 0.7509 | 1,083 | 81.29 | 0.675 | 1,058 | 81.40 | 0.6975 | 1,036 |
| FLoRA | 81.70 | 0.6994 | 1,105 | 82.38 | 0.6346 | 1,037 | 81.29 | 0.6579 | 1,175 |
| RAFFT | 85.49 | 0.2758 | 818 | 84.09 | 0.3583 | 863 | 83.85 | 0.3623 | 890 |
| RAFFT-RGD | 82.8 | 0.4166 | 802 | 82.78 | 0.3928 | 822 | 82.12 | 0.4140 | 822 |
| RAFFT-MR | 82.25 | 0.4259 | 798 | 83.12 | 0.4039 | 819 | 81.95 | 0.4232 | 851 |

Table 6: Performance comparison of different methods on Roberta+MRPC.

| Method | alpha = 5 | | | alpha = 3 | | | alpha = 1 | | |
|---|---|---|---|---|---|---|---|---|---|
| | Accuracy | Loss | Time | Accuracy | Loss | Time | Accuracy | Loss | Time |
| Centralized LoRA | 92.89 | 0.3394 | 1,883 | 93.35 | 0.3276 | 1,862 | 92.89 | 0.3166 | 1,877 |
| AdaLoRA | 88.07 | 1.7253 | 1,710 | 87.96 | 1.7722 | 1,711 | 87.16 | 1.6742 | 1,720 |
| P-tuning v2 | 89.44 | 0.6417 | 1,545 | 90.36 | 0.5675 | 1,504 | 90.60 | 0.4632 | 1,533 |
| FedPrompt | 89.68 | 0.7164 | 1,487 | 93.81 | 0.5829 | 1,460 | 89.33 | 0.5689 | 1,495 |
| FedPepTAO | 91.06 | 0.4388 | 1,614 | 91.74 | 0.4267 | 1,620 | 91.51 | 0.3499 | 1,581 |
| PromptFL | 92.20 | 0.5044 | 1,453 | 93.12 | 0.5096 | 1,444 | 93.92 | 0.3634 | 1,489 |
| PE_FL | 93.00 | 0.4628 | 1,503 | 93.00 | 0.4732 | 1,498 | 93.69 | 0.3523 | 1,470 |
| SLoRA | 92.66 | 0.2616 | 1,447 | 92.88 | 0.1956 | 1,473 | 90.25 | 0.2588 | 1,501 |
| HetLoRA | 93.46 | 0.2785 | 1,067 | 92.43 | 0.5086 | 1,095 | 93.46 | 0.2091 | 1,128 |
| FedLoRA | 93.35 | 0.3002 | 1,221 | 93.58 | 0.2877 | 1,192 | 93.69 | 0.2878 | 1,248 |
| FFA-LoRA | 92.77 | 0.3048 | 1,022 | 91.62 | 0.8016 | 1,087 | 92.43 | 0.4128 | 1,065 |
| Fedkseed | 93.23 | 0.3662 | 1,124 | 93.23 | 0.3633 | 1,251 | 93.00 | 0.3636 | 1,103 |
| FLoRA | 93.46 | 0.3175 | 1,158 | 93.46 | 0.3254 | 1,186 | 93.12 | 0.3307 | 1,251 |
| RAFFT | 95.07 | 0.1894 | 1,012 | 95.41 | 0.1720 | 1,012 | 95.18 | 0.1605 | 1,020 |
| RAFFT-RGD | 93.12 | 0.2654 | 957 | 92.88 | 0.3290 | 952 | 93.35 | 0.2026 | 961 |
| RAFFT-MR | 93.00 | 0.2921 | 962 | 92.88 | 0.3280 | 951 | 93.46 | 0.2337 | 958 |

Table 7: Performance comparison of different methods on Roberta+SST2.

| Method | alpha = 5 | | | alpha = 3 | | | alpha = 1 | | |
|---|---|---|---|---|---|---|---|---|---|
| | Accuracy | Loss | Time | Accuracy | Loss | Time | Accuracy | Loss | Time |
| Centralized LoRA | 82.07 | 0.3670 | 4,331 | 81.67 | 0.4179 | 4,255 | 81.36 | 0.4257 | 4,291 |
| AdaLoRA | 81.22 | 0.8810 | 4,086 | 81.22 | 0.8473 | 4,069 | 81.22 | 0.7275 | 4,078 |
| P-tuning v2 | 81.34 | 4.1410 | 4,023 | 81.34 | 4.0730 | 4,103 | 80.82 | 4.1940 | 4,062 |
| FedPrompt | 81.18 | 4.3970 | 4,120 | 81.46 | 0.5310 | 4,094 | 81.22 | 0.4466 | 4,109 |
| FedPepTAO | 81.34 | 4.1410 | 3,891 | 81.34 | 0.5334 | 4,093 | 81.22 | 0.4460 | 4,125 |
| PromptFL | 79.19 | 5.7040 | 4,120 | 81.34 | 0.5884 | 4,093 | 81.22 | 0.4490 | 4,125 |
| PE_FL | 73.99 | 6.8350 | 4,114 | 81.34 | 0.5792 | 4,101 | 81.22 | 0.4447 | 4,111 |
| SLoRA | 81.78 | 0.3714 | 4,955 | 81.89 | 0.3811 | 4,952 | 80.15 | 0.4276 | 4,966 |
| HetLoRA | 76.80 | 4.6050 | 3,994 | 78.22 | 6.8990 | 3,995 | 78.27 | 7.0250 | 4,012 |
| FedLoRA | 82.07 | 0.4880 | 4,093 | 82.14 | 0.4748 | 4,207 | 81.97 | 0.3731 | 4,017 |
| FFA-LoRA | 80.76 | 3.2140 | 4,186 | 79.03 | 7.0660 | 4,008 | 78.78 | 7.1700 | 3,991 |
| Fedkseed | 74.92 | 6.5395 | 4,114 | 82.00 | 0.5822 | 4,100 | 81.84 | 0.4495 | 4,110 |
| FLoRA | 81.64 | 2.0501 | 4,912 | 81.83 | 0.5054 | 4,985 | 81.68 | 0.4092 | 4,963 |
| RAFFT | 82.63 | 0.4869 | 3,904 | 82.41 | 0.5877 | 3,915 | 82.63 | 0.4738 | 3,892 |
| RAFFT-RGD | 82.07 | 0.5319 | 3,971 | 81.95 | 0.5170 | 3,978 | 81.95 | 0.5623 | 3,956 |
| RAFFT-MR | 82.20 | 0.5133 | 3,956 | 81.98 | 0.6992 | 3,985 | 81.98 | 0.6992 | 3,985 |

Table 8: Performance comparison of different methods on LLaMA3B+MRPC.

| Method | alpha = 5 | | | alpha = 3 | | | alpha = 1 | | |
|---|---|---|---|---|---|---|---|---|---|
| | Accuracy | Loss | Time | Accuracy | Loss | Time | Accuracy | Loss | Time |
| Centralized LoRA | 94.61 | 0.1793 | 3,899 | 95.41 | 0.1268 | 3,897 | 94.38 | 0.1752 | 3,921 |
| AdaLoRA | 87.16 | 0.8543 | 3,532 | 88.12 | 0.9312 | 3,528 | 86.47 | 1.0330 | 3,548 |
| P-tuning v2 | 93.23 | 0.2659 | 3,891 | 94.15 | 0.2156 | 3,900 | 92.43 | 0.2635 | 3,888 |
| FedPrompt | 93.69 | 0.3066 | 3,881 | 94.15 | 0.2606 | 3,893 | 94.04 | 0.2540 | 3,894 |
| FedPepTAO | 93.12 | 0.2619 | 3,778 | 94.15 | 0.2162 | 3,896 | 94.04 | 0.2540 | 3,894 |
| PromptFL | 93.92 | 0.3336 | 3,906 | 94.15 | 0.2606 | 3,893 | 92.32 | 0.2612 | 3,893 |
| PE_FL | 93.92 | 0.3063 | 3,924 | 94.15 | 0.2588 | 3,901 | 93.92 | 0.2627 | 3,890 |
| SLoRA | 94.04 | 0.2177 | 5,878 | 94.27 | 0.1750 | 5,792 | 93.81 | 0.1809 | 5,924 |
| HetLoRA | 94.04 | 0.2637 | 3,893 | 93.92 | 0.2726 | 3,638 | 94.50 | 0.2174 | 3,634 |
| FedLoRA | 94.61 | 0.2022 | 3,664 | 94.72 | 0.1556 | 3,723 | 95.18 | 0.1804 | 3,704 |
| FFA-LoRA | 93.12 | 0.2330 | 3,789 | 93.12 | 0.2419 | 3,593 | 93.46 | 0.2061 | 3,606 |
| Fedkseed | 93.92 | 0.3074 | 3,918 | 94.69 | 0.2604 | 3,900 | 94.80 | 0.2649 | 3,890 |
| FLoRA | 94.36 | 0.2370 | 3,853 | 94.90 | 0.19 | 3,904 | 94.00 | 0.2198 | 3,922 |
| RAFFT | 95.64 | 0.2048 | 3,672 | 95.53 | 0.1458 | 3,733 | 95.64 | 0.2029 | 3,749 |
| RAFFT-RGD | 93.81 | 0.2365 | 3,596 | 94.15 | 0.2314 | 3,634 | 93.92 | 0.2604 | 3,671 |
| RAFFT-MR | 93.80 | 0.2389 | 3,604 | 94.03 | 0.2353 | 3,615 | 93.57 | 0.2529 | 3,648 |

Table 9: Performance comparison of different methods on LLaMA3B+SST2.

| Method | alpha = 5 | | | alpha = 3 | | | alpha = 1 | | |
|---|---|---|---|---|---|---|---|---|---|
| | Accuracy | Loss | Time | Accuracy | Loss | Time | Accuracy | Loss | Time |
| Centralized LoRA | 80.32 | 0.3386 | 7,783 | 80.02 | 0.3900 | 7,691 | 80.09 | 0.3255 | 7,722 |
| AdaLoRA | 81.18 | 0.7412 | 7,522 | 81.07 | 0.7650 | 7,509 | 81.07 | 0.6333 | 7,518 |
| P-tuning v2 | 81.07 | 0.9424 | 7,329 | 81.05 | 0.9306 | 7,261 | 81.07 | 0.8343 | 7,257 |
| FedPrompt | 81.34 | 0.5585 | 7,735 | 81.35 | 0.5322 | 7,667 | 81.63 | 0.4787 | 7,685 |
| FedPepTAO | 81.22 | 0.8325 | 7,328 | 81.11 | 0.9207 | 7,270 | 81.07 | 0.8174 | 7,253 |
| PromptFL | 81.46 | 0.5137 | 7,735 | 81.34 | 0.5329 | 7,673 | 81.07 | 0.8174 | 7,253 |
| PE_FL | 81.46 | 0.5348 | 7,741 | 81.35 | 0.5334 | 7,666 | 81.53 | 0.4791 | 7,678 |
| SLoRA | 81.66 | 0.5086 | 7,400 | 81.93 | 0.4805 | 7,314 | 81.30 | 0.3968 | 7,342 |
| HetLoRA | 80.88 | 0.5677 | 7,400 | 80.47 | 0.5675 | 7,342 | 80.88 | 0.4960 | 7,349 |
| FedLoRA | 81.60 | 0.4918 | 7,328 | 81.75 | 0.4760 | 7,492 | 81.60 | 0.4917 | 7,303 |
| FFA-LoRA | 80.90 | 0.8238 | 7,394 | 80.37 | 0.8623 | 7,337 | 81.26 | 0.7849 | 7,331 |
| Fedkseed | 81.81 | 0.5464 | 7,740 | 82.06 | 0.5397 | 7,666 | 81.74 | 0.5715 | 7,285 |
| FLoRA | 81.74 | 0.6461 | 7,314 | 80.98 | 0.7247 | 7,311 | 79.16 | 0.6914 | 7,338 |
| RAFFT | 83.62 | 0.4950 | 7,285 | 82.58 | 0.6641 | 7,216 | 83.11 | 0.6454 | 7,298 |
| RAFFT-RGD | 80.85 | 0.7416 | 7,291 | 80.73 | 0.7416 | 7,188 | 80.89 | 0.8807 | 7,301 |
| RAFFT-MR | 81.55 | 0.5580 | 7,295 | 82.03 | 0.6274 | 7,246 | 81.64 | 0.7618 | 7,288 |

Table 10: Performance comparison of different methods on LLaMA7B+MRPC.

| Method | alpha = 5 | | | alpha = 3 | | | alpha = 1 | | |
|---|---|---|---|---|---|---|---|---|---|
| | Accuracy | Loss | Time | Accuracy | Loss | Time | Accuracy | Loss | Time |
| Centralized LoRA | 94.61 | 0.1541 | 7,332 | 94.84 | 0.1400 | 7,442 | 95.18 | 0.1677 | 6,922 |
| AdaLoRA | 93.35 | 0.2736 | 7,129 | 93.58 | 0.2273 | 7,154 | 78.89 | 1.0824 | 6,811 |
| P-tuning v2 | 90.02 | 0.2629 | 6,790 | 85.32 | 0.2731 | 6,807 | 94.04 | 0.2220 | 6,825 |
| FedPrompt | 82.91 | 0.3996 | 6,797 | 85.89 | 0.2773 | 6,813 | 93.92 | 0.2249 | 6,810 |
| FedPepTAO | 90.14 | 0.2632 | 6,719 | 90.14 | 0.2215 | 6,822 | 94.50 | 0.2258 | 6,811 |
| PromptFL | 88.42 | 0.2809 | 6,827 | 84.98 | 0.2677 | 6,813 | 93.92 | 0.2220 | 6,816 |
| PE_FL | 86.93 | 0.3327 | 6,797 | 84.29 | 0.2869 | 6,809 | 93.92 | 0.2262 | 6,815 |
| SLoRA | 89.22 | 0.2458 | 6,770 | 81.94 | 0.4805 | 7,314 | 81.29 | 0.3968 | 7,342 |
| HetLoRA | 91.40 | 0.2757 | 6,818 | 90.25 | 0.2588 | 6,860 | 95.18 | 0.1805 | 6,818 |
| FedLoRA | 94.45 | 0.1840 | 6,979 | 94.38 | 0.1599 | 6,498 | 95.18 | 0.1805 | 6,454 |
| FFA-LoRA | 94.38 | 0.2463 | 6,810 | 92.66 | 0.2617 | 6,838 | 92.89 | 0.1956 | 6,817 |
| Fedkseed | 86.77 | 0.3502 | 6,799 | 85.40 | 0.2927 | 6,809 | 94.18 | 0.2298 | 6,814 |
| FLoRA | 92.66 | 0.2216 | 6,610 | 92.77 | 0.1940 | 6,637 | 93.96 | 0.2028 | 6,916 |
| RAFFT | 96.33 | 0.1982 | 6,681 | 95.53 | 0.1949 | 6,618 | 95.41 | 0.2015 | 6,744 |
| RAFFT-RGD | 92.78 | 0.2951 | 6,731 | 93.69 | 0.2795 | 6,731 | 93.12 | 0.3003 | 6,738 |
| RAFFT-MR | 94.15 | 0.2758 | 6,724 | 93.92 | 0.2323 | 6,692 | 93.69 | 0.3629 | 6,819 |

Table 11: Performance comparison of different methods on LLaMA7B+SST2.

| Method | alpha = 5 | | | alpha = 3 | | | alpha = 1 | | |
|---|---|---|---|---|---|---|---|---|---|
| | Accuracy | Loss | Time | Accuracy | Loss | Time | Accuracy | Loss | Time |
| Centralized LoRA | 46.43 | 1.7763 | 3,340 | 46.32 | 1.8492 | 3,394 | 43.43 | 2.0657 | 3,314 |
| AdaLoRA | 46.15 | 1.7674 | 3,548 | 46.10 | 1.8546 | 3,488 | 42.81 | 2.1023 | 3,454 |
| SLoRA | 44.10 | 1.9510 | 3,372 | 43.34 | 1.9870 | 3,306 | 41.39 | 2.1920 | 3,362 |
| HetLoRA | 46.31 | 1.8132 | 3,347 | 45.68 | 1.8629 | 3,321 | 43.53 | 2.0552 | 3,305 |
| FedLoRA | 50.08 | 1.5120 | 3,404 | 44.94 | 1.8940 | 3,290 | 42.40 | 2.1020 | 3,371 |
| FFA-LoRA | 45.99 | 1.8065 | 3,363 | 46.06 | 1.8210 | 3,363 | 43.19 | 1.9970 | 3,334 |
| FLoRA | 40.39 | 1.5424 | 3,288 | 45.36 | 1.6339 | 3,326 | 41.93 | 1.811 | 3,418 |
| RAFFT | 48.44 | 1.3240 | 3,273 | 48.36 | 1.3390 | 3,295 | 45.82 | 1.5210 | 3,314 |
| RAFFT-RGD | 47.11 | 1.5020 | 3,308 | 46.96 | 1.5480 | 3,280 | 43.33 | 1.6940 | 3,297 |
| RAFFT-MR | 47.21 | 1.4959 | 3,625 | 46.47 | 1.5746 | 3,301 | 43.75 | 1.7250 | 3,309 |

Table 12: Performance comparison of different methods on ViT+CIFRA100.

| Method | alpha = 5 | | | alpha = 3 | | | alpha = 1 | | |
|---|---|---|---|---|---|---|---|---|---|
| | Accuracy | Loss | Time | Accuracy | Loss | Time | Accuracy | Loss | Time |
| Centralized LoRA | 26.92 | 2.9963 | 11,497 | 26.18 | 3.0743 | 11,139 | 24.34 | 3.2956 | 11,260 |
| AdaLoRA | 27.10 | 3.0361 | 12,278 | 26.72 | 3.0535 | 11,820 | 24.38 | 3.2249 | 11,908 |
| SLoRA | 26.76 | 3.2740 | 11,176 | 25.76 | 3.3610 | 11,755 | 24.18 | 3.6510 | 11,878 |
| HetLoRA | 26.78 | 3.0303 | 11,487 | 25.90 | 3.0713 | 11,317 | 24.14 | 3.2865 | 11,278 |
| FedLoRA | 26.80 | 3.2570 | 11,643 | 26.24 | 3.3519 | 11,720 | 24.32 | 3.6429 | 11,960 |
| FFA-LoRA | 26.42 | 3.2642 | 11,700 | 26.02 | 3.3664 | 11,787 | 24.06 | 3.6111 | 11,797 |
| FLoRA | 23.96 | 3.5034 | 11,564 | 23.56 | 3.5504 | 11,508 | 21.82 | 3.8373 | 11,755 |
| RAFFT | 28.82 | 2.9140 | 11,236 | 27.88 | 2.9350 | 11,195 | 25.12 | 3.1270 | 11,225 |
| RAFFT-RGD | 27.74 | 3.0150 | 11,159 | 26.90 | 3.0470 | 11,598 | 24.08 | 3.2850 | 11,591 |
| RAFFT-MR | 27.16 | 3.0480 | 10,938 | 27.18 | 3.0370 | 11,245 | 24.58 | 3.2260 | 10,957 |

Table 13: Performance comparison of different methods on ViT+IamgeNet.

**Accuracy of classification using Riemannian manifolds on different numbers of clients.** To comprehensively investigate the applicability of our RAFFT method for classification tasks under different client numbers, we evaluated the method with 40, 60, 80, and 100 clients while keeping other settings constant. Tables 14-19 present the classification accuracy on three datasets using Riemannian manifold techniques. We observed consistent results, indicating that the three variants of our RAFFT method achieved the best accuracy in most experiments. This demonstrates the superior performance of RAFFT under varying client numbers. A plausible explanation is that the rigorous mathematical analysis based on Riemannian manifold theory significantly enhances the effectiveness and applicability of our RAFFT method across different scenarios.

| Method | N = 40 | | | N = 60 | | | N = 80 | | | N = 100 | | |
|---|---|---|---|---|---|---|---|---|---|---|---|---|
| | Accuracy | Loss | Time | Accuracy | Loss | Time | Accuracy | Loss | Time | Accuracy | Loss | Time |
| Centralized LoRA | 91.70 | 0.1511 | 1,481 | 90.95 | 0.1604 | 1,489 | 91.40 | 0.2203 | 1,472 | 89.65 | 0.2510 | 1,433 |
| AdaLoRA | 89.60 | 0.2471 | 1,669 | 89.15 | 0.2639 | 1,730 | 88.05 | 0.3008 | 1,710 | 85.85 | 0.4136 | 1,763 |
| P-tuning v2 | 89.25 | 0.2481 | 1,588 | 88.10 | 0.2930 | 1,610 | 87.60 | 0.3337 | 1,631 | 87.45 | 0.3563 | 1,581 |
| FedPrompt | 87.90 | 0.4247 | 1,485 | 87.15 | 0.4846 | 1,584 | 85.25 | 0.5106 | 1,495 | 86.00 | 0.5352 | 1,463 |
| FedPepTAO | 89.95 | 0.1627 | 1,586 | 89.60 | 0.2348 | 1,683 | 89.20 | 0.2421 | 1,583 | 89.20 | 0.2854 | 1,608 |
| PromptFL | 87.70 | 0.4394 | 1,479 | 86.50 | 0.4824 | 1,575 | 84.90 | 0.5149 | 1,499 | 85.75 | 0.5358 | 1,696 |
| PE_FL | 88.05 | 0.3954 | 1,507 | 86.15 | 0.4736 | 1,489 | 85.65 | 0.4727 | 1,529 | 85.50 | 0.5331 | 1,663 |
| SLoRA | 91.15 | 0.1598 | 1,410 | 90.70 | 0.2509 | 1,463 | 91.00 | 0.2743 | 1,469 | 90.05 | 0.2544 | 1,488 |
| HetLoRA | 88.10 | 0.3145 | 1,154 | 86.80 | 0.3668 | 1,221 | 84.00 | 0.4775 | 1,276 | 87.70 | 0.3377 | 1,207 |
| FedLoRA | 91.30 | 0.1522 | 1,493 | 90.75 | 0.2478 | 1,539 | 91.10 | 0.2744 | 1,571 | 89.95 | 0.2473 | 1,399 |
| FFA-LoRA | 87.55 | 0.3448 | 1,065 | 86.00 | 0.4245 | 1,098 | 84.45 | 0.3580 | 1,119 | 87.50 | 0.3580 | 1,193 |
| Fedkseed | 88.80 | 0.3138 | 1,114 | 86.90 | 0.4047 | 1,141 | 84.95 | 0.5082 | 1,168 | 82.75 | 0.5833 | 1,176 |
| FLoRA | 89.05 | 0.2966 | 1,202 | 87.75 | 0.3707 | 1,151 | 85.65 | 0.4662 | 1,129 | 84.25 | 0.5227 | 1,154 |
| RAFFT | 91.80 | 0.1065 | 842 | 92.00 | 0.1285 | 905 | 91.70 | 0.1469 | 885 | 91.45 | 0.1692 | 905 |

Table 14: Performance comparison of different methods on Roberta+MPQA with varying numbers of clients (N).

| Method | N = 40 | | | N = 60 | | | N = 80 | | | N = 100 | | |
|---|---|---|---|---|---|---|---|---|---|---|---|---|
| | Accuracy | Loss | Time | Accuracy | Loss | Time | Accuracy | Loss | Time | Accuracy | Loss | Time |
| Centralized LoRA | 85.21 | 0.2362 | 1,416 | 83.55 | 0.3623 | 1,395 | 83.72 | 0.3956 | 1,443 | 81.32 | 0.4122 | 1,407 |
| AdaLoRA | 83.37 | 0.2942 | 1,545 | 83.62 | 0.3217 | 1,506 | 83.75 | 0.3641 | 1,514 | 81.13 | 0.4309 | 1,563 |
| P-tuning v2 | 80.63 | 0.2974 | 1,037 | 78.13 | 0.4139 | 1,088 | 79.11 | 0.4611 | 1,059 | 78.48 | 0.4325 | 1,049 |
| FedPrompt | 76.29 | 0.5435 | 1,003 | 75.29 | 0.6265 | 1,003 | 75.13 | 0.7028 | 992 | 75.87 | 0.6784 | 1,004 |
| FedPepTAO | 84.98 | 0.2090 | 1,042 | 81.40 | 0.2732 | 1,080 | 81.68 | 0.3586 | 1,120 | 80.85 | 0.3982 | 1,049 |
| PromptFL | 75.55 | 0.6329 | 1,022 | 74.98 | 0.6772 | 1,045 | 75.34 | 0.7687 | 972 | 75.84 | 0.5917 | 1,001 |
| PE_FL | 83.26 | 0.2482 | 1,045 | 80.75 | 0.2792 | 1,087 | 81.27 | 0.3896 | 1,037 | 76.18 | 0.5686 | 1,024 |
| SLoRA | 85.43 | 0.3257 | 1,029 | 83.57 | 0.3722 | 1,066 | 81.39 | 0.3846 | 1,021 | 80.43 | 0.4559 | 1,044 |
| HetLoRA | 81.68 | 0.4203 | 1,035 | 78.83 | 0.4678 | 1,143 | 76.77 | 0.5138 | 1,150 | 80.52 | 0.4470 | 1,086 |
| FedLoRA | 85.07 | 0.4072 | 1,083 | 84.11 | 0.4674 | 1,114 | 82.92 | 0.5052 | 1,075 | 81.64 | 0.5645 | 1,095 |
| FFA-LoRA | 81.06 | 0.3960 | 1,116 | 80.87 | 0.4550 | 1,033 | 78.30 | 0.5054 | 1,036 | 77.74 | 0.4846 | 1,060 |
| Fedkseed | 81.32 | 0.6024 | 1,075 | 81.28 | 0.6224 | 1,084 | 81.34 | 0.6945 | 1,041 | 81.40 | 0.6975 | 1,153 |
| FLoRA | 82.38 | 0.5507 | 1,153 | 81.52 | 0.6001 | 1,031 | 81.38 | 0.6627 | 1,052 | 81.29 | 0.6579 | 1,188 |
| RAFFT | 86.80 | 0.2629 | 994 | 85.55 | 0.2914 | 1,005 | 85.70 | 0.3949 | 997 | 84.75 | 0.3519 | 1,097 |

Table 15: Performance comparison of different methods on Roberta+MRPC with varying numbers of clients (N).

| Method | N = 40 | | | N = 60 | | | N = 80 | | | N = 100 | | |
|---|---|---|---|---|---|---|---|---|---|---|---|---|
| | Accuracy | Loss | Time | Accuracy | Loss | Time | Accuracy | Loss | Time | Accuracy | Loss | Time |
| Centralized LoRA | 94.83 | 0.2296 | 1,149 | 94.38 | 0.2550 | 1,133 | 94.38 | 0.2523 | 1,157 | 92.89 | 0.3394 | 1,883 |
| AdaLoRA | 93.81 | 0.2471 | 1,269 | 93.69 | 0.2498 | 1,270 | 93.46 | 0.2424 | 1,223 | 88.07 | 1.7253 | 1,710 |
| P-tuning v2 | 90.02 | 0.6433 | 1,562 | 88.07 | 0.7700 | 1,527 | 87.27 | 0.7822 | 1,534 | 89.44 | 0.6417 | 1,545 |
| FedPrompt | 93.69 | 0.4831 | 1,501 | 90.13 | 0.6522 | 1,490 | 90.37 | 0.7105 | 1,472 | 89.68 | 0.7164 | 1,487 |
| FedPepTAO | 91.28 | 0.4281 | 1,592 | 90.83 | 0.5123 | 1,623 | 90.71 | 0.5028 | 1,603 | 91.06 | 0.4388 | 1,614 |
| PromptFL | 92.78 | 0.4751 | 1,469 | 91.86 | 0.5922 | 1,472 | 90.48 | 0.6318 | 1,432 | 92.20 | 0.5044 | 1,453 |
| PE_FL | 94.50 | 0.4647 | 1,512 | 92.43 | 0.5441 | 1,492 | 91.85 | 0.5617 | 1,478 | 93.00 | 0.4628 | 1,503 |
| SLoRA | 95.64 | 0.2084 | 1,185 | 95.30 | 0.2302 | 1,215 | 95.18 | 0.2233 | 1,239 | 81.65 | 3.4859 | 1,447 |
| HetLoRA | 93.69 | 0.2472 | 1,071 | 93.23 | 0.2734 | 1,143 | 93.12 | 0.3029 | 1,054 | 93.46 | 0.2785 | 1,067 |
| FedLoRA | 95.41 | 0.2875 | 1,178 | 95.05 | 0.2962 | 1,206 | 94.95 | 0.2701 | 1,215 | 93.35 | 0.3002 | 1,221 |
| FFA-LoRA | 93.00 | 0.3431 | 1,032 | 92.32 | 0.4621 | 995 | 92.77 | 0.5617 | 1,015 | 93.46 | 0.3048 | 1,022 |
| Fedkseed | 94.03 | 0.2437 | 1,258 | 93.58 | 0.2739 | 1,314 | 93.69 | 0.2736 | 1,142 | 93.00 | 0.2724 | 1,046 |
| FLoRA | 94.38 | 0.2389 | 1,192 | 93.69 | 0.2623 | 1,207 | 93.69 | 0.2544 | 1,103 | 93.12 | 0.3307 | 1,045 |
| RAFFT | 95.76 | 0.1561 | 922 | 95.30 | 0.1665 | 944 | 95.30 | 0.1396 | 969 | 95.07 | 0.1894 | 1012 |

Table 16: Performance comparison of different methods on Roberta+SST2 with varying numbers of clients (N).

| Method | N = 40 | | | N = 60 | | | N = 80 | | | N = 100 | | |
|---|---|---|---|---|---|---|---|---|---|---|---|---|
| | Accuracy | Loss | Time | Accuracy | Loss | Time | Accuracy | Loss | Time | Accuracy | Loss | Time |
| Centralized LoRA | 71.95 | 0.6527 | 3,278 | 71.77 | 0.6565 | 3,321 | 71.48 | 0.6874 | 3,286 | 71.90 | 0.7170 | 3,329 |
| AdaLoRA | 72.34 | 0.6616 | 3,518 | 72.12 | 0.6467 | 3,627 | 71.83 | 0.6873 | 3,590 | 71.76 | 0.6936 | 3,535 |
| SLoRA | 72.76 | 0.6372 | 3,301 | 72.39 | 0.6652 | 3,294 | 71.77 | 0.6634 | 3,330 | 71.75 | 0.6843 | 3,309 |
| HetLoRA | 72.22 | 0.6614 | 3,310 | 71.81 | 0.6831 | 3,312 | 71.30 | 0.7046 | 3,350 | 71.76 | 0.6939 | 3,355 |
| FedLoRA | 72.35 | 0.6800 | 3,269 | 71.64 | 0.6867 | 3,343 | 71.53 | 0.6873 | 3,301 | 71.56 | 0.7149 | 3,282 |
| FFA-LoRA | 71.79 | 0.6435 | 3,269 | 71.91 | 0.6848 | 3,362 | 71.53 | 0.6993 | 3,310 | 71.47 | 0.7135 | 3,321 |
| FLoRA | 72.59 | 0.5949 | 3,409 | 72.25 | 0.6060 | 3,373 | 71.03 | 0.5980 | 3,350 | 70.95 | 0.7266 | 3,401 |
| RAFFT | 75.71 | 0.5021 | 3,279 | 75.67 | 0.5252 | 3,286 | 74.61 | 0.5232 | 3,301 | 74.54 | 0.5757 | 3,304 |

Table 17: Performance comparison of different methods on ViT+CIFRA10 with varying numbers of clients (N).

| Method | N = 40 | | | N = 60 | | | N = 80 | | | N = 100 | | |
|---|---|---|---|---|---|---|---|---|---|---|---|---|
| | Accuracy | Loss | Time | Accuracy | Loss | Time | Accuracy | Loss | Time | Accuracy | Loss | Time |
| Centralized LoRA | 45.69 | 1.7550 | 3,267 | 45.88 | 1.7610 | 3,295 | 45.46 | 1.8640 | 3,251 | 46.43 | 1.7763 | 3,340 |
| AdaLoRA | 44.28 | 1.8590 | 3,535 | 44.53 | 1.8950 | 3,489 | 43.54 | 1.7670 | 3,538 | 46.15 | 1.7674 | 3,548 |
| SLoRA | 44.78 | 1.8630 | 3,304 | 45.16 | 1.8620 | 3,237 | 44.52 | 1.9560 | 3,311 | 44.10 | 1.9510 | 3,372 |
| HetLoRA | 45.80 | 1.7540 | 3,278 | 45.90 | 1.7990 | 3,315 | 45.50 | 1.8280 | 3,334 | 46.31 | 1.8132 | 3,347 |
| FedLoRA | 46.13 | 1.7310 | 3,335 | 45.72 | 1.7500 | 3,286 | 44.95 | 1.8430 | 3,289 | 45.26 | 1.5120 | 3,404 |
| FFA-LoRA | 45.75 | 1.7200 | 3,331 | 45.85 | 1.7780 | 3,280 | 45.28 | 1.8550 | 3,404 | 45.99 | 1.8065 | 3,363 |
| FLoRA | 48.11 | 1.6190 | 3,329 | 48.09 | 1.6210 | 3,494 | 46.76 | 1.6030 | 3,344 | 40.39 | 1.5424 | 3,288 |
| RAFFT | 48.94 | 1.1740 | 3,193 | 48.85 | 1.2130 | 3,122 | 48.27 | 1.3310 | 3,284 | 48.44 | 1.3240 | 3,273 |

Table 18: Performance comparison of different methods on ViT+CIFRA100 with varying numbers of clients (N).

| Method | N = 40 | | | N = 60 | | | N = 80 | | | N = 100 | | |
|---|---|---|---|---|---|---|---|---|---|---|---|---|
| | Accuracy | Loss | Time | Accuracy | Loss | Time | Accuracy | Loss | Time | Accuracy | Loss | Time |
| Centralized LoRA | 28.46 | 3.0426 | 11,604 | 26.34 | 3.1904 | 11,737 | 26.82 | 3.2050 | 11,546 | 26.92 | 2.9963 | 11,497 |
| AdaLoRA | 28.86 | 2.9607 | 12,264 | 26.50 | 3.1529 | 12,481 | 26.64 | 3.1230 | 12,433 | 27.10 | 3.0361 | 12,278 |
| SLoRA | 28.32 | 3.0342 | 11,583 | 26.40 | 3.2080 | 11,834 | 26.92 | 3.2540 | 11,734 | 26.76 | 3.2740 | 11,176 |
| HetLoRA | 28.50 | 2.9923 | 11,838 | 26.58 | 3.2077 | 12,349 | 27.00 | 3.2010 | 12,045 | 26.78 | 3.0303 | 11,487 |
| FedLoRA | 28.30 | 2.9999 | 11,855 | 26.48 | 3.1991 | 12,008 | 26.84 | 3.2300 | 11,922 | 26.80 | 3.2570 | 11,643 |
| FFA-LoRA | 28.44 | 3.0310 | 11,789 | 26.40 | 3.2059 | 11,810 | 26.78 | 3.2140 | 11,941 | 26.42 | 3.2642 | 11,700 |
| FLoRA | 26.96 | 3.499 | 11,586 | 26.68 | 3.4914 | 11,716 | 25.76 | 3.5060 | 11,516 | 23.96 | 3.5034 | 11,564 |
| RAFFT | 29.66 | 2.6210 | 11,038 | 27.36 | 2.9390 | 11,142 | 29.24 | 2.8190 | 11,349 | 28.82 | 2.9140 | 11,236 |

Table 19: Performance comparison of different methods on ViT+IamgeNet with varying numbers of clients (N).

**Accuracy of classification using Riemannian manifolds on different initial rank values.** To comprehensively investigate the applicability of our RAFFT method for classification tasks with different initial rank values, we set the initial ranks to 2, 4, 8, and 16 while keeping other settings constant. Tables 20-25 present the classification accuracy on three datasets using Riemannian manifold techniques. We observed consistent results, indicating that the three variants of our RAFFT method achieved the best accuracy in most experiments. A plausible explanation is that the rigorous mathematical analysis based on Riemannian manifold theory significantly enhances the effectiveness and applicability of our RAFFT method across different scenarios.

| Method | r = 2 | | | r = 4 | | | r = 8 | | | r = 16 | | |
|---|---|---|---|---|---|---|---|---|---|---|---|---|
| | Accuracy | Loss | Time | Accuracy | Loss | Time | Accuracy | Loss | Time | Accuracy | Loss | Time |
| Centralized LoRA | 90.6 | 0.2157 | 1,040 | 90.55 | 0.2153 | 1,038 | 89.65 | 0.2510 | 1,442 | 90.55 | 0.2075 | 1,011 |
| AdaLoRA | 82.46 | 0.8709 | 1,237 | 81.80 | 0.8581 | 1,195 | 85.85 | 0.4136 | 1,774 | 82.10 | 0.8887 | 1,228 |
| SLoRA | 90.10 | 0.2219 | 1,034 | 90.10 | 0.2213 | 1,065 | 90.05 | 0.2545 | 1,473 | 90.10 | 0.2222 | 1,073 |
| HetLoRA | 86.60 | 0.3707 | 1,318 | 83.20 | 0.5256 | 1,203 | 87.70 | 0.3377 | 1,207 | 89.60 | 0.2552 | 1,186 |
| FedLoRA | 90.20 | 0.2170 | 1,095 | 90.30 | 0.2175 | 1,049 | 89.95 | 0.2473 | 1,416 | 90.20 | 0.2177 | 1,140 |
| FFA-LoRA | 87.95 | 0.3290 | 1,158 | 88.55 | 0.3251 | 1,139 | 87.50 | 0.3580 | 1,193 | 87.85 | 0.3413 | 1,140 |
| FLoRA | 84.15 | 0.5238 | 1,159 | 84.10 | 0.5257 | 1,180 | 84.25 | 0.5227 | 1,197 | 84.30 | 0.5250 | 1,186 |
| RAFFT | 91.00 | 0.1900 | 849 | 91.05 | 0.1806 | 848 | 91.45 | 0.1692 | 905 | 91.55 | 0.1500 | 918 |

Table 20: Performance comparison of different methods on Roberta+MPQA with varying rank values (r).

| Method | r = 2 | | | r = 4 | | | r = 8 | | | r = 16 | | |
|---|---|---|---|---|---|---|---|---|---|---|---|---|
| | Accuracy | Loss | Time | Accuracy | Loss | Time | Accuracy | Loss | Time | Accuracy | Loss | Time |
| Centralized LoRA | 82.27 | 0.3894 | 1,417 | 82.49 | 0.3853 | 1,385 | 81.32 | 0.4122 | 1,402 | 81.86 | 0.3862 | 1,401 |
| AdaLoRA | 79.98 | 0.4240 | 1,516 | 82.34 | 0.3992 | 1,528 | 81.14 | 0.4309 | 1,563 | 81.66 | 0.4020 | 1,572 |
| SLoRA | 82.27 | 0.3864 | 1,039 | 82.08 | 0.3794 | 1,012 | 80.43 | 0.4559 | 1,044 | 81.38 | 0.3824 | 1,074 |
| HetLoRA | 75.40 | 0.5103 | 1,045 | 74.78 | 0.5767 | 1,033 | 80.52 | 0.4470 | 1,086 | 74.78 | 0.9160 | 1,082 |
| FedLoRA | 81.45 | 0.5971 | 1,047 | 81.37 | 0.5958 | 1,022 | 81.64 | 0.5645 | 1,056 | 81.33 | 0.5974 | 1,018 |
| FFA-LoRA | 74.81 | 0.5438 | 1,003 | 74.98 | 0.5441 | 1,014 | 77.74 | 0.4846 | 1,060 | 74.91 | 0.5549 | 1,048 |
| FLoRA | 81.16 | 0.6580 | 1,037 | 81.17 | 0.6591 | 1,084 | 81.29 | 0.6579 | 1,010 | 81.67 | 0.6592 | 1,024 |
| RAFFT | 85.16 | 0.3345 | 813 | 84.57 | 0.3000 | 825 | 85.39 | 0.2758 | 818 | 86.04 | 0.3453 | 830 |

Table 21: Performance comparison of different methods on Roberta+MRPC with varying rank values (r).

| Method | r = 2 | | | r = 4 | | | r = 8 | | | r = 16 | | |
|---|---|---|---|---|---|---|---|---|---|---|---|---|
| | Accuracy | Loss | Time | Accuracy | Loss | Time | Accuracy | Loss | Time | Accuracy | Loss | Time |
| Centralized LoRA | 93.35 | 0.2529 | 1,040 | 93.19 | 0.2613 | 1,001 | 92.89 | 0.3394 | 1,883 | 93.12 | 0.2614 | 1,076 |
| AdaLoRA | 89.44 | 1.5001 | 1,237 | 89.79 | 1.4422 | 1,195 | 88.07 | 1.7253 | 1,710 | 89.56 | 1.5087 | 1,184 |
| SLoRA | 93.18 | 0.2571 | 1,034 | 92.30 | 0.2595 | 1,065 | 81.65 | 3.4859 | 1,447 | 95.18 | 0.1595 | 1,105 |
| HetLoRA | 93.35 | 0.2531 | 1,032 | 93.00 | 0.3403 | 1,085 | 93.46 | 0.2785 | 1,067 | 92.20 | 0.7275 | 1,093 |
| FedLoRA | 93.92 | 0.2559 | 1,095 | 90.30 | 0.2175 | 1,049 | 93.92 | 0.2548 | 1,221 | 93.92 | 0.2559 | 1,086 |
| FFA-LoRA | 91.74 | 0.6232 | 1,017 | 91.97 | 0.5818 | 1,042 | 92.77 | 0.3048 | 1,022 | 91.86 | 0.7081 | 1,036 |
| FLoRA | 92.26 | 0.2368 | 1,224 | 93.34 | 0.2369 | 1,206 | 93.12 | 0.3307 | 1,232 | 94.27 | 0.2369 | 1,249 |
| RAFFT | 94.15 | 0.2316 | 994 | 95.18 | 0.1790 | 992 | 95.07 | 0.1894 | 1012 | 95.30 | 0.2130 | 1,014 |

Table 22: Performance comparison of different methods on Roberta+SST2 with varying rank values (r).

| Method | r = 2 | | | r = 4 | | | r = 8 | | | r = 16 | | |
|---|---|---|---|---|---|---|---|---|---|---|---|---|
| | Accuracy | Loss | Time | Accuracy | Loss | Time | Accuracy | Loss | Time | Accuracy | Loss | Time |
| Centralized LoRA | 71.14 | 0.7078 | 3,256 | 72.08 | 0.6924 | 3,241 | 71.90 | 0.717 | 3,329 | 70.69 | 0.7221 | 3,318 |
| AdaLoRA | 71.06 | 0.717 | 3,600 | 71.62 | 0.7060 | 3,510 | 71.76 | 0.6936 | 3,535 | 71.47 | 0.7127 | 3,622 |
| SLoRA | 71.54 | 0.6927 | 3,269 | 72.12 | 0.6871 | 3,337 | 71.75 | 0.6843 | 3,309 | 72.03 | 0.7002 | 3,357 |
| HetLoRA | 71.50 | 0.7071 | 3,358 | 70.66 | 0.7172 | 3,333 | 71.76 | 0.6939 | 3,355 | 70.91 | 0.7045 | 3,361 |
| FedLoRA | 71.87 | 0.7133 | 3,277 | 71.28 | 0.7287 | 3,300 | 71.56 | 0.7149 | 3,282 | 71.83 | 0.7111 | 3,306 |
| FFA-LoRA | 71.90 | 0.6900 | 3,316 | 72.10 | 0.7000 | 3,291 | 71.47 | 0.7135 | 3,321 | 71.22 | 0.7167 | 3,341 |
| FLoRA | 71.75 | 0.5803 | 3,840 | 72.38 | 0.6042 | 3,913 | 70.95 | 0.7266 | 3,901 | 71.34 | 0.5885 | 3,970 |
| RAFFT | 73.67 | 0.5615 | 3,282 | 74.23 | 0.5953 | 3,244 | 74.54 | 0.5757 | 3,304 | 73.97 | 0.579 | 3,301 |

Table 23: Performance comparison of different methods on ViT+CIFRA10 with varying rank values (r).

| Method | r = 2 | | | r = 4 | | | r = 8 | | | r = 16 | | |
|---|---|---|---|---|---|---|---|---|---|---|---|---|
| | Accuracy | Loss | Time | Accuracy | Loss | Time | Accuracy | Loss | Time | Accuracy | Loss | Time |
| Centralized LoRA | 44.45 | 1.8870 | 3,212 | 44.32 | 1.9000 | 3,333 | 46.43 | 1.7763 | 3,340 | 44.64 | 1.8830 | 3,399 |
| AdaLoRA | 44.24 | 1.9410 | 3,519 | 43.76 | 1.9600 | 3,507 | 46.15 | 1.7674 | 3,548 | 43.41 | 1.9920 | 3,559 |
| SLoRA | 43.81 | 1.9670 | 3,382 | 43.42 | 2.0000 | 3,317 | 44.10 | 1.9510 | 3,372 | 43.00 | 1.9810 | 3,437 |
| HetLoRA | 44.87 | 1.8690 | 3,314 | 44.83 | 1.8740 | 3,358 | 46.31 | 1.8132 | 3,347 | 44.46 | 1.9110 | 3,440 |
| FedLoRA | 44.41 | 1.9140 | 3,365 | 44.56 | 1.9090 | 3,351 | 45.26 | 1.8920 | 3,260 | 44.53 | 1.8830 | 3,337 |
| FFA-LoRA | 44.62 | 1.9000 | 3,389 | 44.60 | 1.9220 | 3,375 | 45.99 | 1.8065 | 3,363 | 44.00 | 1.8880 | 3,464 |
| FLoRA | 45.05 | 1.6170 | 3,873 | 45.23 | 1.6100 | 3,813 | 40.39 | 1.5424 | 3,888 | 46.91 | 1.6500 | 3,839 |
| RAFFT | 48.76 | 1.3330 | 3,305 | 48.13 | 1.3220 | 3,255 | 48.44 | 1.3240 | 3,273 | 48.94 | 1.3000 | 3,296 |

Table 24: Performance comparison of different methods on ViT+CIFRA100 with varying rank values (r).

| Method | r = 2 Accuracy | Loss | Time | r = 4 Accuracy | Loss | Time | r = 8 Accuracy | Loss | Time | r = 16 Accuracy | Loss | Time |
|---|---|---|---|---|---|---|---|---|---|---|---|---|
| Centralized LoRA | 27.02 | 3.2640 | 11,425 | 27.20 | 3.2700 | 11,383 | 26.92 | 2.9963 | 11,497 | 26.64 | 3.2910 | 11,542 |
| AdaLoRA | 26.58 | 3.2550 | 12,272 | 26.28 | 3.2700 | 12,256 | 27.10 | 3.0362 | 12,278 | 26.36 | 3.2980 | 12,294 |
| SLoRA | 26.82 | 3.2590 | 11,102 | 26.84 | 3.2520 | 11,145 | 26.76 | 3.2740 | 11,176 | 26.16 | 3.2850 | 11,191 |
| HetLoRA | 26.80 | 3.2540 | 11,502 | 27.08 | 3.2500 | 11,394 | 26.78 | 3.0303 | 11,487 | 27.10 | 3.2540 | 11,674 |
| FedLoRA | 26.56 | 3.2610 | 11,523 | 27.26 | 3.2190 | 11,401 | 26.80 | 3.2570 | 11,643 | 26.28 | 3.2890 | 11,723 |
| FFA-LoRA | 27.00 | 3.2530 | 11,685 | 26.92 | 3.2660 | 11,504 | 26.42 | 3.2642 | 11,700 | 26.68 | 3.2630 | 11,894 |
| FLoRA | 24.36 | 3.477 | 11,817 | 24.48 | 3.4890 | 11,905 | 23.96 | 3.5033 | 11,564 | 23.44 | 3.4843 | 12,218 |
| RAFFT | 29.24 | 2.9230 | 11,295 | 28.40 | 2.9040 | 11,012 | 28.82 | 2.9140 | 11,236 | 28.42 | 2.9330 | 11,096 |

Table 25: Performance comparison of different methods on ViT+IamgeNet with varying rank values (r).

| Model | ACC | | | | |
|---|---|---|---|---|---|
| | $\Theta = 0.6$ | $\Theta = 0.7$ | $\Theta = 0.8$ | $\Theta = 0.9$ | $\Theta = 1$ |
| Roberta+SST-2 | 95.30 | 95.30 | 95.41 | 95.07 | 93.00 |
| Roberta+MRPC | 84.22 | 83.22 | 85.67 | 84.75 | 82.25 |
| Roberta+MPQA | 91.15 | 91.30 | 91.55 | 91.45 | 90.15 |
| LLaMA 7B+SST-2 | 95.41 | 95.53 | 95.41 | 95.64 | 94.15 |
| LLaMA 7B+MRPC | 82.77 | 82.77 | 82.77 | 83.62 | 81.55 |
| LLaMA 7B+MPQA | 91.40 | 91.45 | 91.70 | 92.00 | 90.80 |
| ViT+CIFAR10 | 74.07 | 73.53 | 73.95 | 73.89 | 72.88 |
| ViT+CIFAR100 | 49.16 | 48.22 | 47.97 | 48.44 | 47.21 |
| ViT+ImageNet | 28.92 | 28.64 | 28.66 | 28.82 | 27.16 |

Table 26: Performance comparison of different models with varying Theta values (ACC).

| Model | Time (s) | | | | |
|---|---|---|---|---|---|
| | $\Theta = 0.6$ | $\Theta = 0.7$ | $\Theta = 0.8$ | $\Theta = 0.9$ | $\Theta = 1$ |
| Roberta+SST-2 | 976 | 981 | 968 | 1,012 | 962 |
| Roberta+MRPC | 853 | 850 | 843 | 838 | 798 |
| Roberta+MPQA | 865 | 886 | 883 | 905 | 890 |
| LLaMA 7B+SST-2 | 6,848 | 6,848 | 6,852 | 6,988 | 6,961 |
| LLaMA 7B+MRPC | 7,595 | 7,504 | 7,513 | 7,558 | 7,541 |
| LLaMA 7B+MPQA | 4,840 | 5,025 | 4,916 | 5,010 | 5,021 |
| ViT+CIFAR10 | 3,591 | 3,611 | 3,587 | 3,555 | 3,687 |
| ViT+CIFAR100 | 3,553 | 3,695 | 3,638 | 3,585 | 3,625 |
| ViT+ImageNet | 13,260 | 13,073 | 13,388 | 12,706 | 13,164 |

Table 27: Performance comparison of different models with varying Theta values (Time).

| Model | ACC | | | | |
|---|---|---|---|---|---|
| | LR=0.0001 | LR=0.0005 | LR=0.001 | LR=0.005 | LR=0.01 |
| Roberta+SST-2 | 86.24 | 93.81 | 94.84 | 95.30 | 95.64 |
| Roberta+MRPC | 74.78 | 77.33 | 81.93 | 84.40 | 87.04 |
| Roberta+MPQA | 79.55 | 90.10 | 91.20 | 91.55 | 91.70 |
| LLaMA 7B+SST-2 | 95.18 | 96.10 | 96.33 | 61.70 | 50.92 |
| LLaMA 7B+MRPC | 81.52 | 82.70 | 83.62 | 81.22 | 81.22 |
| LLaMA 7B+MPQA | 91.10 | 92.00 | 91.90 | 67.60 | 67.70 |
| ViT+CIFAR10 | 74.54 | 69.28 | 54.97 | 27.90 | 23.73 |
| ViT+CIFAR100 | 48.44 | 44.07 | 29.38 | 6.55 | 6.18 |
| ViT+ImageNet | 28.82 | 16.80 | 15.40 | 9.28 | 2.72 |

Table 28: Performance comparison of different models with varying Learning Rates (ACC).

| Model | Time (s) | | | | |
|---|---|---|---|---|---|
| | LR=0.0001 | LR=0.0005 | LR=0.001 | LR=0.005 | LR=0.01 |
| Roberta+SST-2 | 979 | 981 | 990 | 980 | 967 |
| Roberta+MRPC | 862 | 847 | 832 | 826 | 825 |
| Roberta+MPQA | 905 | 914 | 920 | 907 | 952 |
| LLaMA 7B+SST-2 | 6,971 | 6,953 | 6,988 | 6,812 | 6,937 |
| LLaMA 7B+MRPC | 7,511 | 7,508 | 7,558 | 7,612 | 7,545 |
| LLaMA 7B+MPQA | 4,886 | 4,906 | 4,878 | 4,988 | 4,949 |
| ViT+CIFAR10 | 3,555 | 3,647 | 3,783 | 3,642 | 3,671 |
| ViT+CIFAR100 | 3,585 | 3,704 | 3,651 | 3,598 | 3,543 |
| ViT+ImageNet | 12,706 | 13,041 | 13,215 | 13,192 | 13,264 |

Table 29: Performance comparison of different models with varying Learning Rates (Time).

| Model | ACC | | | | |
|---|---|---|---|---|---|
| | **Epoch 1** | **Epoch 5** | **Epoch 10** | **Epoch 15** | **Epoch 20** |
| Roberta+SST-2 | 87.79 | 93.81 | 94.72 | 95.18 | 95.07 |
| **Epoch 10** | **Epoch 20** | **Epoch 30** | **Epoch 40** | **Epoch 50** | |
| Roberta+MRPC | 76.60 | 79.82 | 81.81 | 83.40 | 85.67 |
| **Epoch 10** | **Epoch 20** | **Epoch 30** | **Epoch 40** | **Epoch 50** | |
| Roberta+MPQA | 88.85 | 90.15 | 90.70 | 91.20 | 91.45 |
| **Epoch 2** | **Epoch 4** | **Epoch 6** | **Epoch 8** | **Epoch 10** | |
| LLaMA 7B+SST-2 | 89.91 | 94.84 | 95.41 | 95.41 | 95.64 |
| **Epoch 5** | **Epoch 10** | **Epoch 15** | **Epoch 20** | **Epoch 25** | |
| LLaMA 7B+MRPC | 81.29 | 81.29 | 82.28 | 82.28 | 83.62 |
| **Epoch 5** | **Epoch 10** | **Epoch 15** | **Epoch 20** | **Epoch 25** | |
| LLaMA 7B+MPQA | 89.45 | 90.90 | 91.25 | 91.25 | 92.00 |
| **Epoch 40** | **Epoch 80** | **Epoch 120** | **Epoch 160** | **Epoch 200** | |
| ViT+CIFAR10 | 57.69 | 64.74 | 69.42 | 71.85 | 74.54 |
| **Epoch 40** | **Epoch 80** | **Epoch 120** | **Epoch 160** | **Epoch 200** | |
| ViT+CIFAR100 | 29.51 | 37.72 | 42.42 | 45.43 | 48.44 |
| **Epoch 50** | **Epoch 100** | **Epoch 200** | **Epoch 300** | **Epoch 400** | |
| ViT+ImageNet | 13.16 | 17.68 | 22.48 | 25.48 | 28.82 |

Table 30: Performance comparison of different models with varying Training Epochs (ACC).

**Accuracy of classification using Riemannian manifolds on different parameters.** As shown in Table 32-33, RAFFT demonstrates superior performance and stability in extreme non-IID data environments across diverse tasks. For instance, in the LLaMA 7B + MPQA task with $\alpha = 0.5$, RAFFT achieves the highest accuracy of 90.80, outperforming all baseline methods. Similarly, in the ViT + CIFAR-10 task with $\alpha = 0.1$, RAFFT achieves the best accuracy of 61.22, surpassing all other methods. These results highlight RAFFT's robustness and effectiveness in handling high data heterogeneity, further supported by its Riemannian framework, which mitigates both client-drift and rank-drift issues and ensures stability across diverse data distributions. As illustrated in Figure 3, the distribution of samples per class for each client in the MPQA dataset reflects the highly heterogeneous nature of the data.

| Model | Time (s) | | | | |
|---|---|---|---|---|---|
| | **Epoch 1** | **Epoch 5** | **Epoch 10** | **Epoch 15** | **Epoch 20** |
| Roberta+SST-2 | 84 | 324 | 638 | 949 | 1,012 |
| **Epoch 10** | **Epoch 20** | **Epoch 30** | **Epoch 40** | **Epoch 50** | |
| Roberta+MRPC | 173 | 341 | 517 | 655 | 843 |
| **Epoch 10** | **Epoch 20** | **Epoch 30** | **Epoch 40** | **Epoch 50** | |
| Roberta+MPQA | 186 | 360 | 528 | 701 | 905 |
| **Epoch 2** | **Epoch 4** | **Epoch 6** | **Epoch 8** | **Epoch 10** | |
| LLaMA 7B+SST-2 | 1,380 | 2,642 | 4,043 | 5,462 | 6,988 |
| **Epoch 5** | **Epoch 10** | **Epoch 15** | **Epoch 20** | **Epoch 25** | |
| LLaMA 7B+MRPC | 1,547 | 2,994 | 4,602 | 6,012 | 7,558 |
| **Epoch 5** | **Epoch 10** | **Epoch 15** | **Epoch 20** | **Epoch 25** | |
| LLaMA 7B+MPQA | 5,071 | 5,063 | 5,102 | 4,972 | 5,010 |
| **Epoch 40** | **Epoch 80** | **Epoch 120** | **Epoch 160** | **Epoch 200** | |
| ViT+CIFAR10 | 736 | 1,450 | 2,143 | 2,935 | 3,555 |
| **Epoch 40** | **Epoch 80** | **Epoch 120** | **Epoch 160** | **Epoch 200** | |
| ViT+CIFAR100 | 722 | 1,460 | 2,203 | 2,911 | 3,585 |
| **Epoch 200** | **Epoch 250** | **Epoch 300** | **Epoch 350** | **Epoch 400** | |
| ViT+ImageNet | 1,645 | 3,288 | 6,728 | 9,691 | 12,706 |

Table 31: Performance comparison of different models with varying Training Epochs (Time).

| Method | Accuracy (%) | Loss | Time (s) |
|---|---|---|---|
| Centralized LoRA | 82.9000 | 0.2834 | 5432 |
| AdaLoRA | 32.7500 | 10.5591 | 5503 |
| P-tuning v2 | 85.2000 | 0.2912 | 5664 |
| FedPrompt | 87.3500 | 0.2639 | 5393 |
| FedPepTAO | 87.6000 | 0.2584 | 5475 |
| PromptFL | 87.4000 | 0.2574 | 5586 |
| PE_FL | 87.2500 | 0.2661 | 5263 |
| SLoRA | 84.1000 | 0.3372 | 5622 |
| HetLoRA | 84.2300 | 0.3035 | 5229 |
| FedLoRA | 87.7400 | 0.1664 | 5308 |
| FFA-LoRA | 80.0000 | 0.3496 | 5358 |
| Fedkseed | 82.4500 | 0.2918 | 5452 |
| FLoRA | 79.1900 | 0.3016 | 5293 |
| RAFFT | 90.8000 | 0.2766 | 5198 |
| RAFFT-RGD | 86.2100 | 0.2947 | 5151 |
| RAFFT-MR | 86.7300 | 0.3004 | 5177 |

Table 32: Performance comparison of methods on LLaMA 7B + MPQA (25 rounds) with $\alpha = 0.5$.

| Method | Accuracy (%) | Loss | Time (s) |
|--------|--------------|------|----------|
| Centralized LoRA | 60.4800 | 1.0437 | 3578 |
| AdaLoRA | 60.9200 | 1.1428 | 3260 |
| SLoRA | 58.8900 | 0.8824 | 3402 |
| HetLoRA | 59.6300 | 1.0270 | 3387 |
| FedLoRA | 59.1100 | 0.9824 | 3310 |
| FFA-LoRA | 60.2100 | 0.9651 | 3412 |
| FLoRA | 59.3100 | 1.0490 | 3249 |
| RAFFT | 61.2200 | 0.8544 | 3222 |
| RAFFT-RGD | 60.6200 | 0.9478 | 3288 |
| RAFFT-MR | 59.9100 | 1.0032 | 3265 |

Table 33: Performance comparison of methods on ViT + CIFAR-10 with $\alpha = 0.1$.

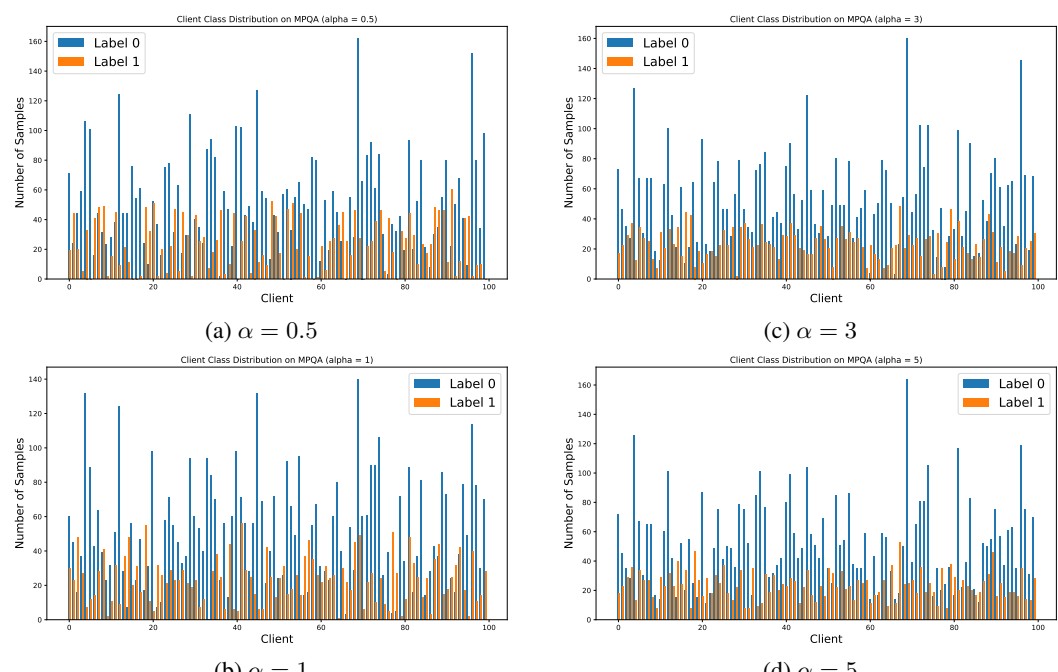

(a) $\alpha = 0.5$

(c) $\alpha = 3$

(b) $\alpha = 1$

(d) $\alpha = 5$

Figure 3: Client class distribution on MPQA dataset for different $\alpha$ values. Each plot shows the number of samples per class across clients, illustrating the heterogeneity in data distribution.

**Resource Efficiency Analysis Using Riemannian Optimization.** The resource usage of our method is comprehensively quantified across multiple dimensions, including FLOPS (measuring computational complexity), throughput (processing rate), communication volume (data transfer during training), training time, and accuracy, as shown in our experimental results. Definitions of these metrics are provided in Bai et al. (2024a).

As shown in Table 34, the resource usage of 16 federated parameter-efficient fine-tuning methods is evaluated on the LLaMA 7B + MPQA task. Our method, RAFFT, achieves the highest throughput of 6229.19 tokens/sec with lower FLOPS (72.73 GFLOPS) compared to other methods (72.76 GFLOPS). This reduction in FLOPS is achieved because the server selects the optimal rank and sets the remaining ranks and their corresponding parameters to zero, thereby reducing unnecessary computations. Additionally, RAFFT significantly reduces the communication volume to 3146112 bytes, the lowest among LoRA-based methods, and achieves the highest accuracy of 0.908. Similarly, on the ViT + CIFAR-10 task, RAFFT achieves the best accuracy of 61.22, with the shortest training time of 3222 seconds and a communication volume of 98352 bytes. These results highlight RAFFT's ability to balance computational and communication efficiency effectively. A plausible explanation for this superior performance is that the Riemannian parameter matching and Riemannian gradient optimization enable faster convergence and improved accuracy, while the adaptive rank se-

lection reduces redundant computation, making RAFFT a highly practical and efficient approach for real-world federated learning scenarios.

| Method | FLOPS (GFLOPS) | Throughput (tokens/sec on H100) | Communication Volume (Bytes) | Training Time (s) | Accuracy (%) |
|---|---|---|---|---|---|
| Centralized LoRA | 72.7557 | 5701.2920 | 4,194,304 | 5432 | 82.90 |
| AdaLoRA | 72.7557 | 6052.1700 | 4,194,816 | 5503 | 32.75 |
| P-tuning v2 | 72.3243 | 5563.1500 | 524,288 | 5664 | 85.20 |
| FedPrompt | 72.3243 | 5100.2800 | 524,288 | 5393 | 87.35 |
| FedPepTAO | 72.3243 | 5829.5450 | 524,288 | 5475 | 87.60 |
| PromptFL | 72.3243 | 5546.1950 | 524,288 | 5586 | 87.40 |
| PE_FL | 72.3243 | 5710.5881 | 524,288 | 5263 | 87.25 |
| SLoRA | 72.7557 | 6199.1755 | 4,194,304 | 5622 | 84.10 |
| HetLoRA | 72.7557 | 5019.9379 | 4,194,304 | 5229 | 84.23 |
| FedLoRA | 72.7557 | 5100.2800 | 6,742,618,112 | 5308 | 87.74 |
| FFA-LoRA | 72.7557 | 5034.1700 | 4,194,304 | 5358 | 80.00 |
| FLoRA | 72.7557 | 4728.9951 | 4,194,304 | 5293 | 79.19 |
| RAFFT | 72.6401 | 6229.1855 | 3,146,112 | 5151 | 90.80 |
| RAFFT-RGD | 72.8020 | 5525.9890 | 4,194,816 | 5198 | 86.21 |
| RAFFT-MR | 72.7557 | 6392.1002 | 3,146,112 | 5177 | 86.73 |

Table 34: Performance comparison of methods on LLaMA 7B + MPQA.

| Method | FLOPS (GFLOPS) | Throughput (images/sec on 2080Ti) | Communication Volume (Bytes) | Training Time (s) | Accuracy (%) |
|---|---|---|---|---|---|
| Centralized LoRA | 41.9654 | 5730.05 | 98,304 | 3578 | 60.48 |
| AdaLoRA | 41.9654 | 4722.30 | 98,352 | 3260 | 60.92 |
| SLoRA | 41.9654 | 5739.46 | 98,304 | 3402 | 58.89 |
| HetLoRA | 41.9654 | 5781.50 | 98,304 | 3387 | 59.63 |
| FedLoRA | 41.9654 | 5663.19 | 5,322,240 | 3310 | 59.11 |
| FFA-LoRA | 41.9654 | 5625.11 | 98,304 | 3412 | 60.21 |
| FLoRA | 41.9654 | 5646.20 | 98,304 | 3249 | 59.31 |
| RAFFT | 41.6335 | 5721.55 | 78,962 | 3222 | 61.22 |
| RAFFT-RGD | 41.9927 | 5315.95 | 98,304 | 3288 | 60.62 |
| RAFFT-MR | 41.9654 | 5681.28 | 78,962 | 3265 | 59.91 |

Table 35: Performance comparison of methods on ViT + CIFAR-10.

**Accuracy of classification using Riemannian manifolds on different parameters.** In order to comprehensively study the applicability of our RAFFT method to classification tasks with different parameter values, in addition to the select rank threshold in section 5 we also evaluate the following parameters: training epoch, learning rate (LR), while keeping other settings unchanged. Table 27-31 shows the classification accuracy using Riemannian manifold techniques on data sets for different models. We observe consistent results, showing that the three variants of our RAFFT method achieve the best accuracy in most experiments. A reasonable explanation is that rigorous mathematical analysis based on Riemannian manifold theory significantly enhances the effectiveness and applicability of our RAFFT method in different scenarios.

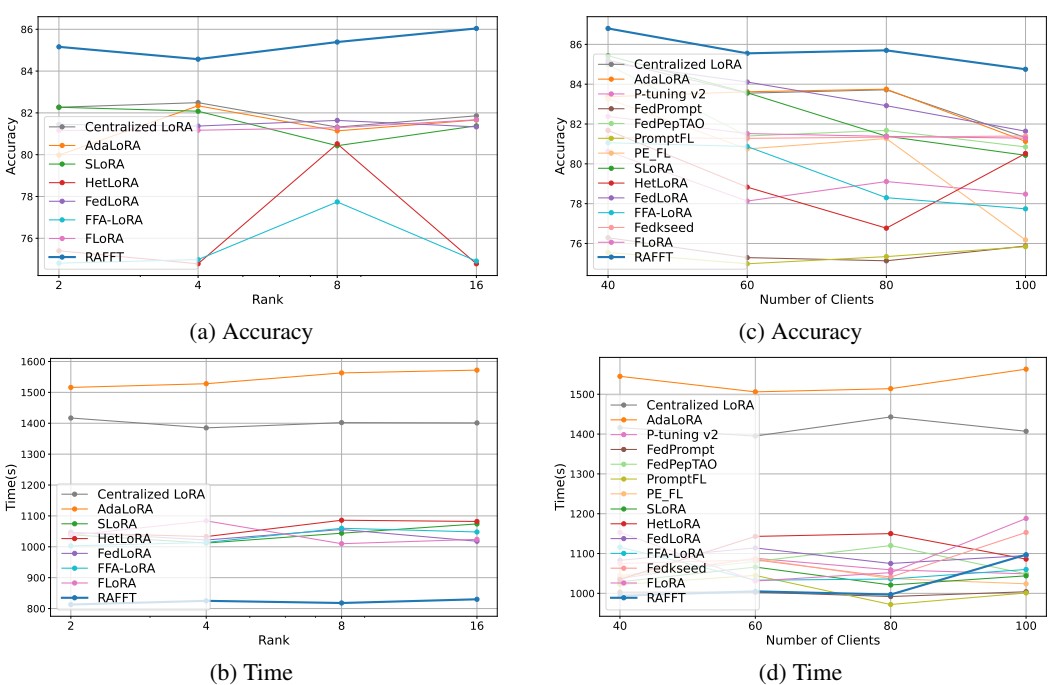

Figure 4: Accuracy and Time (s) with various LoRA ranks and client numbers on Roberta-MRPC

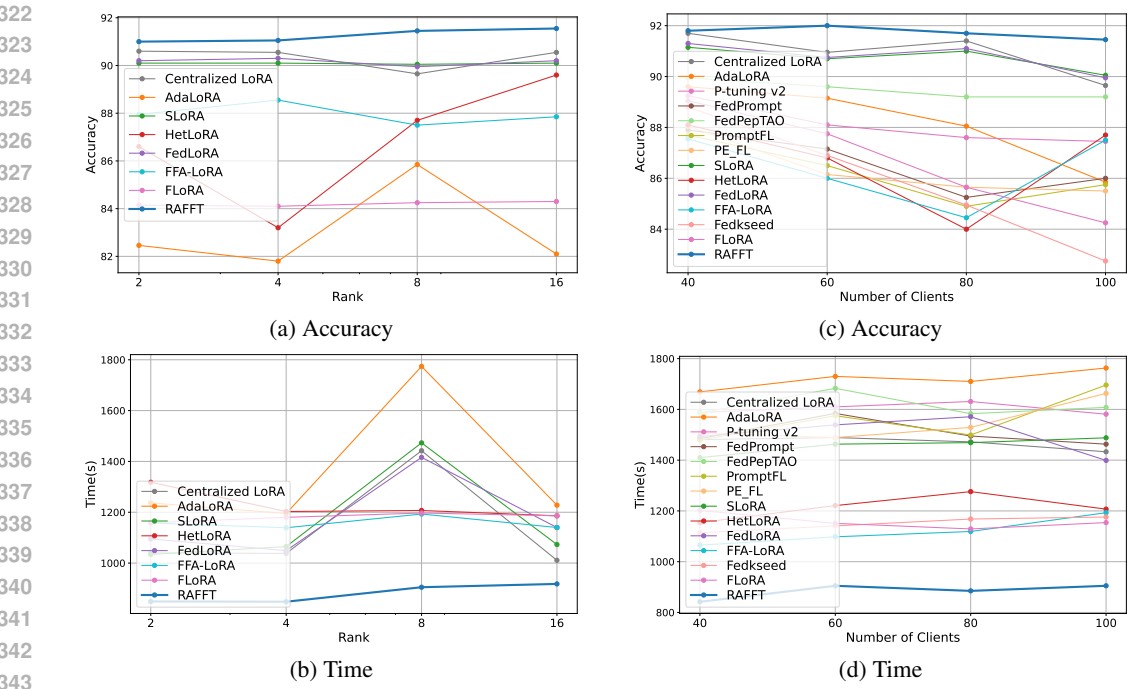

Figure 5: Accuracy and Time (s) with various LoRA ranks and client numbers on Roberta-MPQA

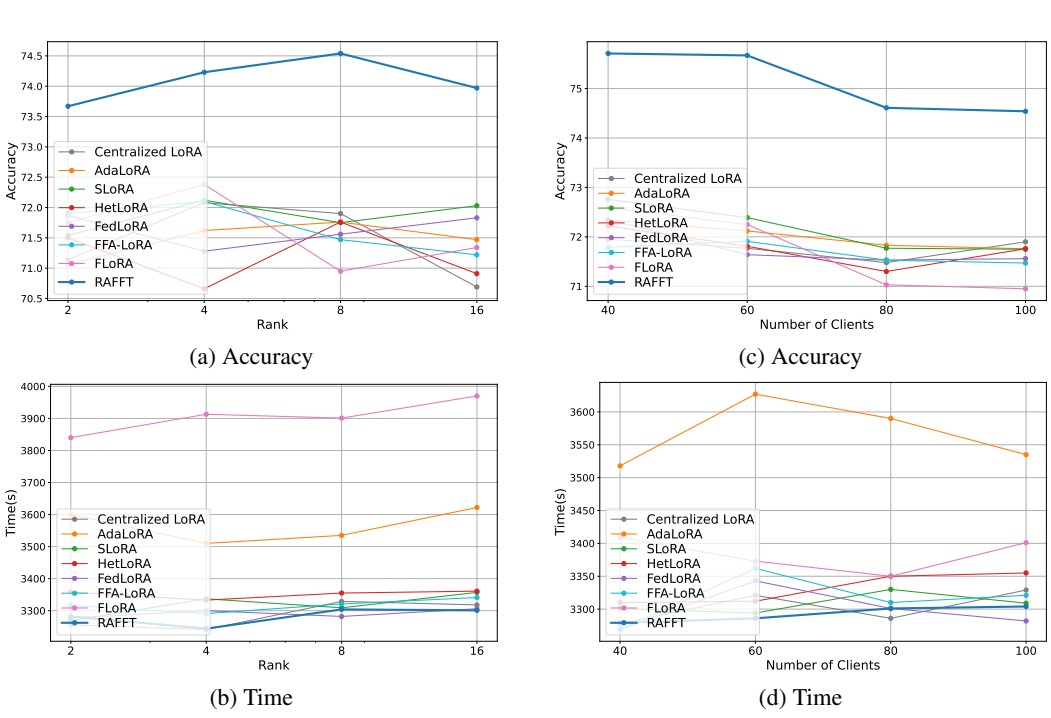

Figure 6: Accuracy and Time (s) with various LoRA ranks and client numbers on ViT-CIFAR-10

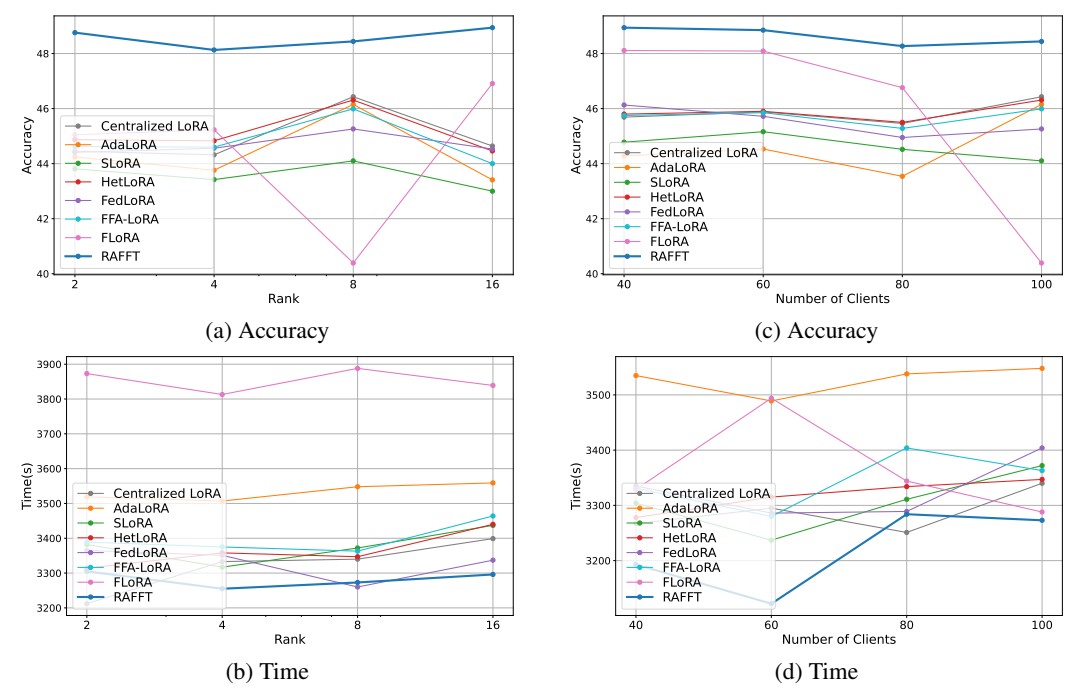

Figure 7: Accuracy and Time (s) with various LoRA ranks and client numbers on ViT-CIFAR-100

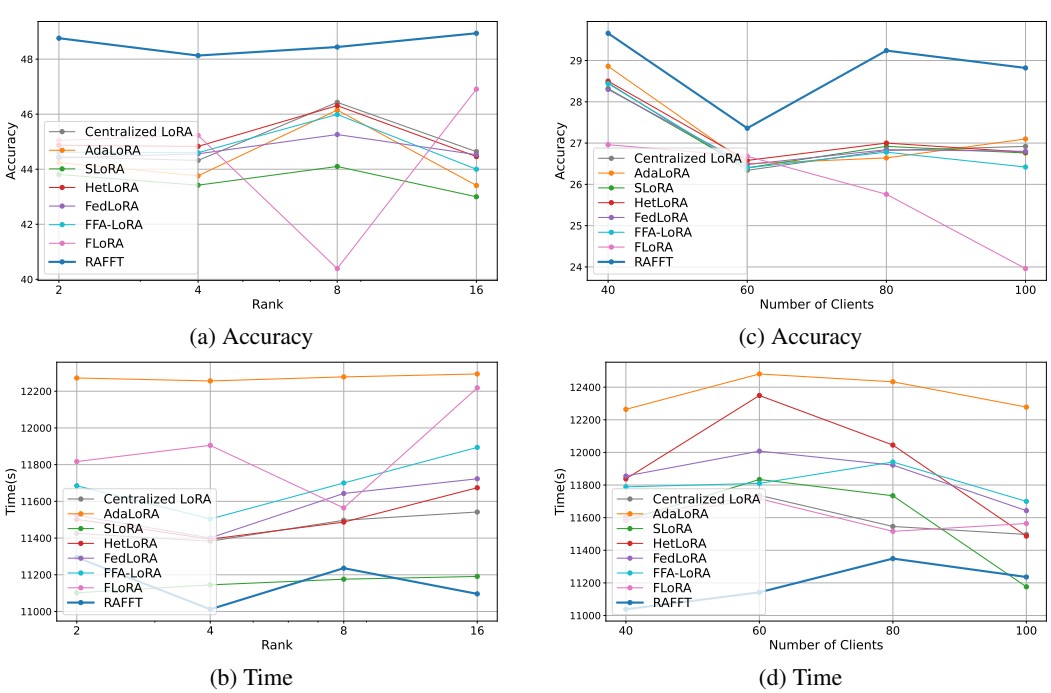

Figure 8: Accuracy and Time (s) with various LoRA ranks and client numbers on ViT-Tiny-ImageNet

## 7.7 POTENTIAL NEGATIVE SOCIETAL IMPACTS AND LIMITATIONS

In this work, the three large language datasets Socher et al. (2013); Dolan & Brockett (2005); Wiebe et al. (2005) and the three image classification datasets are open-release datasets allowing researchers

to use them for non-commercial research and educational purposes. These three datasets are widely used for training/evaluating text classification. All baseline code is open access from GitHub and is licensed under the MIT License, requiring only that the copyright and license notices be retained, and containing permissions for commercial use, modification, distribution, and private use. To the best of our knowledge, this work is the first to theoretically verify the possibility of solving client drift and ranking drift through Riemannian manifold theory. Compared with existing FedPEFT technology, this work explores the equivalence between the theoretical LoRA-based ranking adaptive federated learning method for base model fine-tuning and the standard federated base model federated fine-tuning method. Our model can be used for various large model fine-tuning tasks that require low latency and energy consumption in resource-intensive scenarios. This paper is primarily of a theoretical nature. We expect our findings to produce positive environmental impact, i.e., significantly improve the efficiency and scalability of federated large model fine-tuning by reducing the time and space requirements of large model fine-tuning in federated learning scenarios during training and testing. To our best knowledge, we do not envision any immediate negative societal impacts of our results, such as security, privacy, and fairness issues. An important product of this paper is the theoretical exploration to verify the feasibility of addressing client drift and rank drift through Riemannian manifold theory. Given the large scale of neural networks in practical scenarios and the limitations of current computational hardware, we employ approximate methods for Riemannian distance. This ensures that the rank-adaptive federated learning method based on LoRA for fine-tuning foundation models is equivalent to the standard federated fine-tuning of foundation models on a fully parameterized FedAvg matrix. Our theoretical framework can inspire further improved development and implementations on FedPEFT with better applicability and efficiency from the academic institutions and industrial research labs.

### 7.8 CLINT-DRIFT AND RANK-DRIFT ISSUE DEFINITION

**Client-drift issue:** The client-drift issue is due to the aggregation of low-rank matrices in the FFT-FM. Here, we introduce an illustrative example with two clients to better understand the client-drift issue.

Concretely, if the clients locally fine-tune on full parameters and the server aggregates with FedAvg, the new global model parameters can be expressed as follows.

$$\mathbf{W} = \mathbf{W}^0 + \frac{1}{2}(\Delta\mathbf{W}^1 + \Delta\mathbf{W}^2) \tag{98}$$

In our work, the clients locally fine-tune on both low-rank parameter matrices $\mathbf{U}^k$ and $\mathbf{V}^k$ and diagonal matrix $\mathbf{\Sigma}^k$ to determine the rank. The server uses FedAvg to aggregate the low-rank matrices and diagonal matrix.

$$\Delta\mathbf{W}^k = \mathbf{U}^k\mathbf{\Sigma}^k\mathbf{V}^k, \quad k \in 1, 2 \tag{99}$$

The expected global model parameter updates $\Delta\mathbf{W}$ can be expressed as follows.

$$\Delta\mathbf{W} = \frac{1}{2}(\Delta\mathbf{W}^1 + \Delta\mathbf{W}^2) = \frac{1}{2}(\mathbf{U}^1\mathbf{\Sigma}^1\mathbf{V}^1 + \mathbf{U}^2\mathbf{\Sigma}^2\mathbf{V}^2) \tag{100}$$

After using FedAvg to aggregate the local matrices $\mathbf{U}^k, \mathbf{\Sigma}^k, \mathbf{V}^k$, the server produces

$$\Delta\tilde{\mathbf{W}} = \frac{1}{2}\left(\mathbf{U}^1 + \mathbf{U}^2\right) \times \frac{1}{2}\left(\mathbf{\Sigma}^1 + \mathbf{\Sigma}^2\right) \times \frac{1}{2}\left(\mathbf{V}^1 + \mathbf{V}^2\right)$$

$$= \frac{1}{2}(\mathbf{U}^1\mathbf{\Sigma}^1\mathbf{V}^1 + \mathbf{U}^2\mathbf{\Sigma}^2\mathbf{V}^2) \underline{- \frac{3}{8}\mathbf{U}^1\mathbf{\Sigma}^1\mathbf{V}^1 - \frac{3}{8}\mathbf{U}^2\mathbf{\Sigma}^2\mathbf{V}^2 + \frac{1}{8}\mathbf{U}^1\mathbf{\Sigma}^1\mathbf{V}^2 + \frac{1}{8}\mathbf{U}^1\mathbf{\Sigma}^2\mathbf{V}^1}$$

$$\underline{+ \frac{1}{8}\mathbf{U}^1\mathbf{\Sigma}^2\mathbf{V}^2 + \frac{1}{8}\mathbf{U}^2\mathbf{\Sigma}^1\mathbf{V}^1 + \frac{1}{8}\mathbf{U}^2\mathbf{\Sigma}^1\mathbf{V}^2 + \frac{1}{8}\mathbf{U}^2\mathbf{\Sigma}^2\mathbf{V}^1}$$

$$= \Delta\mathbf{W} \underline{- \frac{3}{8}\mathbf{U}^1\mathbf{\Sigma}^1\mathbf{V}^1 - \frac{3}{8}\mathbf{U}^2\mathbf{\Sigma}^2\mathbf{V}^2 + \frac{1}{8}\mathbf{U}^1\mathbf{\Sigma}^1\mathbf{V}^2 + \frac{1}{8}\mathbf{U}^1\mathbf{\Sigma}^2\mathbf{V}^1}$$

$$\underline{+ \frac{1}{8}\mathbf{U}^1\mathbf{\Sigma}^2\mathbf{V}^2 + \frac{1}{8}\mathbf{U}^2\mathbf{\Sigma}^1\mathbf{V}^1 + \frac{1}{8}\mathbf{U}^2\mathbf{\Sigma}^1\mathbf{V}^2 + \frac{1}{8}\mathbf{U}^2\mathbf{\Sigma}^2\mathbf{V}^1} \tag{101}$$

where the underlined terms denote the difference between $\Delta\tilde{\mathbf{W}}$ and $\Delta\mathbf{W}$.

It is obvious that $\Delta\tilde{\mathbf{W}} \neq \Delta\mathbf{W}$ when $\mathbf{U}^1 \neq \mathbf{U}^2$, $\mathbf{\Sigma}^1 \neq \mathbf{\Sigma}^2$, and $\mathbf{V}^1 \neq \mathbf{V}^2$ due to the data heterogeneity property of federated learning. Therefore, the client-drift issue is raised.

$$\underbrace{\tilde{\mathbf{W}} = \mathbf{W}^0 + \Delta\tilde{\mathbf{W}}}_{\text{Parameter aggregation with SVD-based rank-adaptive LoRA + FedAvg}} \quad\neq\quad \underbrace{\mathbf{W}^0 + \Delta\mathbf{W} = \mathbf{W}}_{\text{Ideal parameter aggregation with FedAvg on full parameter matrices}} \tag{102}$$

The difference between $\Delta\tilde{\mathbf{W}}$ and $\Delta\mathbf{W}$ is mainly due to the noise introduced by the cross-products of LoRA modules from different clients. This difference may become more significant when (1) the number of local update steps between aggregations is large and (2) the local datasets are different across clients.

In the (Sun et al., 2024) paper, when the clients locally fine-tune on low-rank parameters based on LoRA and the server uses FedAvg to aggregate the low-rank matrices, the global model parameters can be expressed as follows.

$$\underbrace{\tilde{\mathbf{W}} = \mathbf{W}^0 + \frac{1}{2}(\mathbf{B}^1 + \mathbf{B}^2) \times \frac{1}{2}(\mathbf{A}^1 + \mathbf{A}^2)}_{\text{Parameter aggregation with LoRA + FedAvg}} \neq \underbrace{\mathbf{W}^0 + \frac{1}{2}(\mathbf{B}^1\mathbf{A}^1 + \mathbf{B}^2\mathbf{A}^2) = \mathbf{W}_0 + \frac{1}{2}(\Delta\mathbf{W}^1 + \Delta\mathbf{W}^2) = \mathbf{W}}_{\text{Ideal parameter aggregation with FedAvg on full parameter matrices}} \tag{103}$$

When the clients use LoRA locally, we have $\Delta\mathbf{W}^k = \mathbf{B}^k\mathbf{A}^k$ on client $k$. The data heterogeneity is a common challenging problem in the federated learning. The collected data on different clients are Non-Independent Identically Distributed (NonIID). Thus, the parameters of local models trained on the non-IID data are quite different from each other, i.e., $\mathbf{B}^1 \neq \mathbf{B}^2$ and $\mathbf{A}^1 \neq \mathbf{A}^2$. This leads to the client-drift issue in the (Sun et al., 2024) paper, i.e., the difference between $\mathbf{W}$ and $\tilde{\mathbf{W}}$ is not equal to 0. In order to address the client-drift issue, the (Sun et al., 2024) paper keeps both $\mathbf{W}^0$ and $\mathbf{A}^0$ frozen and makes only $\mathbf{B}$ trainable.

$$\underbrace{\tilde{\mathbf{W}} = \mathbf{W}^0 + \frac{1}{2}(\mathbf{B}^1 + \mathbf{B}^2) \times \mathbf{A}^0}_{\text{Parameter aggregation with LoRA + FedAvg}} = \underbrace{\mathbf{W}^0 + \frac{1}{2}(\mathbf{B}^1\mathbf{A}^0 + \mathbf{B}^2\mathbf{A}^0) = \mathbf{W}_0 + \frac{1}{2}(\Delta\mathbf{W}^1 + \Delta\mathbf{W}^2) = \mathbf{W}}_{\text{Ideal parameter aggregation with FedAvg on full parameter matrices}} \tag{104}$$

The magic of the (Sun et al., 2024) paper to handle the client-drift issue is to have only one parameter matrix ($\mathbf{B}$) trainable while freezing other parameter matrices. However, this approach cannot be directly utilized to conduct the client-drift issue in our work. Our work has three trainable parameter matrices $\mathbf{U}^k$, $\mathbf{V}^k$, and $\mathbf{\Sigma}^k$. If updating only the diagonal matrix $\mathbf{\Sigma}^k$ and freezing the parameter matrices $\mathbf{U}^k$ and $\mathbf{V}^k$, then the algorithm fails to update the parameter matrices, resulting in poor model performance. If updating either $\mathbf{U}^k$ or $\mathbf{V}^k$ and freezing $\mathbf{\Sigma}^k$, then the rank cannot be optimized, leading to high computational cost (by higher rank) or poor model performance (by lower rank).

Thus, We propose a Riemannian parameter matching method to match the local parameter matrices $\mathbf{U}^k$ and $\mathbf{V}^k$ on other clients with pivots $\mathbf{U}^1$ and $\mathbf{V}^1$ on client 1, in terms of their lengths

and directions, i.e. $\tilde{\mathbf{U}}^k = \mathbf{U}^k\mathbf{R}^k \approx \mathbf{U}^1$ and $\tilde{\mathbf{V}}^k = \mathbf{S}^k\mathbf{V}^k \approx \mathbf{V}^1$. To maintain the consistency between low-rank parameter metrics before and after the Riemannian parameter matching, i.e., $\mathbf{U}^2\boldsymbol{\Sigma}^2\mathbf{V}^2 = \tilde{\mathbf{U}}^2\tilde{\boldsymbol{\Sigma}}^2\tilde{\mathbf{V}}^2 \approx \mathbf{U}^1\tilde{\boldsymbol{\Sigma}}^2\mathbf{V}^1$, for ensuring the effectiveness of FFT-FM with rank-adaptive LoRA, we derive a modified diagonal matrix $\tilde{\boldsymbol{\Sigma}}^k$ for the other clients by performing the SVD on low-dimensional $r \times r$ matrices $(\mathbf{U}^1)^T\mathbf{U}^k$ and $(\mathbf{V}^1)^T\mathbf{V}^k$. Thus, the client-drift issue is resolved.

$$\Delta\tilde{\mathbf{W}} = \frac{1}{2}\left(\mathbf{U}^1 + \tilde{\mathbf{U}}^2\right) \times \frac{1}{2}\left(\boldsymbol{\Sigma}^1 + \tilde{\boldsymbol{\Sigma}}^2\right) \times \frac{1}{2}\left(\mathbf{V}^1 + \tilde{\mathbf{V}^K}\right)$$
$$\approx \mathbf{U}^1 \times \frac{1}{2}\left(\boldsymbol{\Sigma}^1 + \tilde{\boldsymbol{\Sigma}}^2\right) \times \mathbf{V}^1 = \frac{1}{2}\left(\mathbf{U}^1\boldsymbol{\Sigma}^1\mathbf{V}^1 + \mathbf{U}^2\boldsymbol{\Sigma}^2\mathbf{V}^2\right) = \Delta\mathbf{W} \tag{105}$$

Based on the global diagonal matrix $\frac{1}{2}\left(\boldsymbol{\Sigma}^1 + \tilde{\boldsymbol{\Sigma}}^2\right)$, it is easy to find the optimal rank of the global parameter matrix, with aggregation on only the local low-rank matrices $\mathbf{U}^k$, $\tilde{\boldsymbol{\Sigma}}^k$, and $\mathbf{V}^k$.

**Rank-drift issue:** The data heterogeneity is a common challenging problem in the federated learning. The collected data on different clients are Non-Independent Identically Distributed (Non-IID). Thus, the parameters of local models trained on the non-IID data are quite different from each other as well as the parameters of global model. Each local model may oscillate back and forth in different training rounds, leading to unstable and slow convergence and causing suboptimal model performance.

At the same time, due to the heterogeneity of local models, the optimal ranks of local models are quite different from each other as well as the one of global model, raising the rank-drift issue. Similarly, the oscillation of the optimal rank of each local model can slow down the model convergence and degrade the model performance. For example, given two clients 1 and 2, in round 1, the local model on client 1/2 has the optimal rank 5/20. The server aggregates local models from clients to generate a global model based on FedAvg. The global model by model aggregation has the optimal rank 12. In round 2, after the global model are sent back to the clients, the clients begin the training with the global model with the optimal rank 12. Thus, the optimal rank of local models oscillate between 5/20 and 12. The rank oscillation may repeat in each training round, slowing down the model convergence.

The following table 36 presents a case study regarding the rank-drift in federated learning. In round 1, the optimal rank of the global model is 16. In round 2, the optimal rank is changed to 11. In subsequent rounds, the optimal rank continue to oscillate back and forth. The ranks of local models fluctuate around values such as 9.75, 10.17, and 10.53, while the rank of global modelgradually decreases and stabilizes at 8. These results confirm that rank-drift introduces instability into the training process, slowing convergence and degrading performance, which emphasizes the importance of our Riemannian gradient descent optimization approach to stabilize ranks and ensure efficient training.

## 7.9 ALGORITHM

In order to solve the client-drift and rank-drift issues and significantly improves the computational cost, we exploit a Riemannian manifold based method as detailed in Algorithm 1. Within each round, every client performs local updates(line 22). The updates are based on the Riemannian gradient and the retraction function (line 11,line 13). After the updates, each client sends the updated parameters back to the server(line 17). The server then performs Riemannian parameter matching between the U and V matrices of all clients and those of the first client to achieve alignment in both length and direction.(line 24-line 28) The singular value matrix is updated according to Formula7, and the aggregated U, V and $\Sigma$ matrices from all clients are combined.(line 29-line 33) The rank for the next round of training is selected through Formula 16, and the model parameter matrix is updated accordingly.(line 34-line 37)

## 7.10 NOTATION DEFINITION

Below, we provide detailed definitions of the notations used in the main text:

---

**Algorithm 1** Federated Learning with Optimal Rank Selection and Alignment

---

1: **Input:** Client datasets $\{\mathcal{D}_k\}_{k=1}^{K}$, Threshold $\alpha$, The number of local epoch $E$, The number of global round $R$
2: **Output:** Optimally ranked updated matrices $U_R, S_R, V_R$
3: **procedure** SERVER INITIALIZATION
4:     Initialize $S$ as zero matrix
5:     Initialize $U, V$ with random Gaussian values
6: **end procedure**
7: **procedure** CLIENT UPDATE (K,U,S,V) // RUN ON CLIENT K
8:     **for** each local epoch i from 1 to E **do**
9:         batches $\leftarrow (\mathcal{D}_k$ spilt into batches of size)
10:         **for** batch b in batches **do**
11:             computes the Riemannian gradient $P\nabla L^k$ based on Eq.(21)
12:             $(\mathbf{U}^k L^k, \mathbf{\Sigma}^k L^k, \mathbf{V}^k L^k)$
13:             computes $(\mathbf{U}^k)^{(t+1)}, (\mathbf{\Sigma}^k)^{(t+1)}, (\mathbf{V}^k)^{(t+1)}$ based on Eq.(22)
14:             $\left((\mathbf{U}^k)^{(t+1)}, (\mathbf{\Sigma}^k)^{(t+1)}, (\mathbf{V}^k)^{(t+1)}\right) = f\left(-\eta_t \left(\left(\Gamma_{\mathbf{U}^k} L^k\right)^{(t)}, \left(\Gamma_{\mathbf{\Sigma}^k} L^k\right)^{(t)}, \left(\Gamma_{\mathbf{V}^k} L^k\right)^{(t)}\right)\right)$
15:         **end for**
16:     **end for**
17:     return $(\mathbf{U}^k)^{(t+1)}, (\mathbf{\Sigma}^k)^{(t+1)}, (\mathbf{V}^k)^{(t+1)}$ to the server
18: **end procedure**
19: **procedure** SERVER ALIGNMENT AND AGGREGATION
20:     **for** each round t from 1 to C **do**
21:         **for** each client k in K clients **do**
22:             $(\mathbf{U}^k)^{(t+1)}, (\mathbf{\Sigma}^k)^{(t+1)}, (\mathbf{V}^k)^{(t+1)} \leftarrow$ Client Update (k,$(\mathbf{U}^k)^{(t)}, (\mathbf{\Sigma}^k)^{(t)}, (\mathbf{V}^k)^{(t)}$)
23:         **end for**
24:         Compute alignment matrix $\mathbf{S}^k, \mathbf{R}^k$ using $(\mathbf{U}^1)^{(t+1)}, (\mathbf{V}^1)^{(t+1)}$ as reference based on Eq.(9)
25:         **for** $k = 1$ to $K$ **do**
26:             Align $(\mathbf{U}^k)^{(t+1)}$ to $(\mathbf{U}^1)^{(t+1)}$: $(\tilde{\mathbf{U}}^k)^{(t+1)} = (\mathbf{U}^k)^{(t+1)}\mathbf{R}^k$
27:             Align $(\mathbf{V}^k)^{(t+1)}$ to $(\mathbf{V}^1)^{(t+1)}$: $(\tilde{\mathbf{V}}^k)^{(t+1)} = \mathbf{S}^k(\mathbf{V}^k)^{(t+1)}$
28:         **end for**
29:         Update $(\mathbf{\Sigma}^k)^{(t+1)}$ based on Eq.(13)
30:         Aggregate updates:
31:         $\tilde{\mathbf{U}}' = \sum_{k=1}^{K} \frac{n_k}{n} \tilde{\mathbf{U}}^k$
32:         $\tilde{\mathbf{\Sigma}}' = \sum_{k=1}^{K} \frac{n_k}{n} \tilde{\mathbf{\Sigma}}^k$
33:         $\tilde{\mathbf{V}}' = \sum_{k=1}^{K} \frac{n_k}{n} \tilde{\mathbf{V}}^k$
34:         Determine rank $r$ based on Eq.(16):
35:         $\Theta(r) = \frac{\sum_{i=1}^{r} \tilde{\mathbf{\Sigma}}'_{ii}}{\sum_{i=1}^{r_{max}} \tilde{\mathbf{\Sigma}}'_{ii}} \geq \varphi$
36:         Update $\tilde{\mathbf{\Sigma}}'$ to only include top $r$ singular values
37:         Update $\tilde{\mathbf{U}}', \tilde{\mathbf{V}}'$ accordingly
38:     **end for**
39: **end procedure**

---

| Round (1-25) | Client Rank | Server Rank |
|---|---|---|
| 1 | 16.000000 | 16 |
| 2 | 9.206250 | 11 |
| 3 | 9.153125 | 10 |
| 4 | 9.750000 | 10 |
| 5 | 10.165625 | 9 |
| 6 | 10.004688 | 9 |
| 7 | 9.881250 | 9 |
| 8 | 9.951563 | 9 |
| 9 | 10.168750 | 9 |
| 10 | 10.392188 | 8 |
| 11 | 10.467188 | 8 |
| 12 | 10.526563 | 8 |
| 13 | 10.506250 | 8 |
| 14 | 10.518750 | 8 |
| 15 | 10.507813 | 8 |
| 16 | 10.454688 | 8 |
| 17 | 10.393750 | 8 |
| 18 | 10.295313 | 8 |
| 19 | 10.212500 | 8 |
| 20 | 10.175000 | 8 |
| 21 | 10.110938 | 8 |
| 22 | 10.029688 | 8 |
| 23 | 9.981250 | 8 |
| 24 | 9.932813 | 8 |
| 25 | 9.903125 | 8 |

Table 36: Client and Server Ranks for LLaMA 7B + MPQA across 25 Rounds

| Notation | Explanation |
|---|---|
| $\mathbf{W} \in \mathbb{R}^{m \times n}$ | Full parameter matrix of the foundation model |
| $\mathbf{W}^k \in \mathbb{R}^{m \times n}$ | Local model parameter matrix for client $K$. |
| $\mathbf{W}_0 \in \mathbb{R}^{m \times n}$ | Pre-trained foundation model parameters |
| $\Delta\mathbf{W} \in \mathbb{R}^{m \times n}$ | Adapter parameters, used for fine-tuning |
| $\Delta\mathbf{W}^k \in \mathbb{R}^{m \times n}$ | Adapter parameter matrix for client $K$ |
| $\mathbf{U}^k \in \mathbb{R}^{m \times r}, \mathbf{V}^k \in \mathbb{R}^{r \times n}$ | Matrices representing the left and right singular vectors of $\Delta\mathbf{W}^k$ |
| $\boldsymbol{\Sigma}^k \in \mathbb{R}^{r \times r}$ | Diagonal matrix containing the singular values of $\Delta\mathbf{W}^k$ |
| $\mathbf{B} \in \mathbb{R}^{r \times n}, \mathbf{A} \in \mathbb{R}^{m \times r}$ | Low-rank matrices representing adapter parameters $\Delta\mathbf{W} = \mathbf{BA}$ |
| $r \in \mathbb{R}^{r \times r}$ | Rank of decomposition, much smaller than $m$ and $n$ |
| $D = \{D_1, \cdots, D_K\}$ | The set of local training data from $K$ clients |
| $K$ | The total number of clients involved in federated learning |
| $n^k$ | The size of the local dataset $D_k$, i.e., $n^k = |D_k|$. |
| $n$ | The size of the total training data $\mathbf{D}$, i.e., $n = n_1 + \cdots + n_K$. |
| $\mathcal{L}(\mathbf{W})$ | The global loss function to be minimized, aggregated over all clients. |
| $L^k(\mathbf{W})$ | The loss function for client $K$. |
| $l_i(\mathbf{W})$ | The loss function for a single data sample $\{x_i, y_i\}$ |
| $S$ | The set of orthogonal matrices on Riemann manifold |
| $S_p$ | The procrustes representation space on manifold $S$ |
| $\mathbf{R}^k \in \mathbb{R}^{r \times r}, \mathbf{S}^k \in \mathbb{R}^{r \times r}$ | The symmetric positive matrix for Riemannian parameter matching. |
| $\mathbf{P}^k \in \mathbb{R}^{m \times r}, \boldsymbol{\Lambda}^k \in \mathbb{R}^{r \times r}, \mathbf{Q}^k \in \mathbb{R}^{r \times n}$ | Matrices obtained from the SVD of $(\mathbf{U}^1)^T \mathbf{U}^k$ |
| $\mathbf{Y}^k, \mathbf{Z}^k$ | Auxiliary variables introduced to reformulate the Riemann problem. |
| $\tilde{\mathbf{U}}^k \in \mathbb{R}^{m \times r}, \tilde{\mathbf{V}}^k \in \mathbb{R}^{r \times n}$ | Matrices representing the aligned parameters after Riemannian matching |
| $\tilde{\boldsymbol{\Sigma}}^k$ | Modified version of diagonal matrix $\boldsymbol{\Sigma}^k$ for ensuring consistency |
| $\lambda_{R_{\min}}, \lambda_{R_{\max}}$ | Minimum and maximum eigenvalues of the matrix $\mathbf{R}^k$ |
| $\lambda_{S_{\min}}, \lambda_{S_{\max}}$ | Minimum and maximum eigenvalues of the matrix $\mathbf{S}^k$ |
| $H, N$ | Constants bounding the Frobenius norms of matrices $\lambda_{R_{\min}}$ and $\lambda_{S_{\min}}$ |
| $\Theta(r)$ | Singular value contribution rate |
| $\varphi$ | Threshold value for determining rank |
| $R(\mathbf{U}^k, \mathbf{V}^k)$ | Regularization term to enforce orthogonality of $\mathbf{U}^k$ and $\mathbf{V}^k$ |
| $M$ | The set of fixed rank matrices on Riemann manifold |
| $T_M$ | Tangent space of the Riemannian manifold $M$ at the point $\Delta\mathbf{W}$ |
| $\mathbf{G} \in \mathbb{R}^{r \times r}, \mathbf{U}_p^k \in \mathbb{R}^{m \times r}, \mathbf{V}_p^k \in \mathbb{R}^{n \times r}$ | The tangent vector at $\Delta\mathbf{W}$ |
| $\mathbf{U}_\perp^k, \mathbf{V}_\perp^k$ | The orthogonal complements of $\mathbf{U}^k$ and $\mathbf{V}^k$ |
| $\nabla L^k(\Delta\mathbf{W}^k)$ | Gradient of the local objective function $L^k$ in Euclidean space |
| $\Gamma L^k(\Delta\mathbf{W}^k)$ | Riemannian gradient of the local objective function $L^k$ |
| $P$ | Orthogonal projection |
| $\Gamma_{\mathbf{U}^k} L^k, \Gamma_{\boldsymbol{\Sigma}^k} L^k, \Gamma_{\mathbf{V}^k} L^k$ | The components of the Riemannian gradient at $\Delta\mathbf{W}^k$ |
| $\eta_t$ | Learning rate |
| $f()$ | Retraction function |
| $t\,(0 \leq t \leq C)$ | Federated learning round |
| $c_g$ | Geodesic smoothness constant for the function $L^k$ |
| $\zeta$ | Key geometric constant that captures the impact of the manifold's curvature |

Table 37: Definitions of Notations

