# OpenReview forum: "Riemannian Low-Rank Adaptation for Federated Fine-Tuning of Foundation Models"
_ICLR.cc/2025/Conference — ICLR 2025 Conference Withdrawn Submission_

### Official Review · Reviewer_aRkt · 2024-11-01

**Soundness:** 3
**Presentation:** 3
**Contribution:** 3
**Rating:** 6
**Confidence:** 3

**Summary:**

This paper presents RAFFT, a Riemannian LoRA algorithm with adaptive rank for federated fine-tuning of foundation models. The proposed method addresses the challenges of client drift and rank drift in federated LoRA by employing a Riemannian parameter matching technique. A Riemannian gradient descent algorithm is derived for rank consistency during local updates and to accelerate convergence.
The theoretical analysis shows rank invariance and convergence.

**Strengths:**

1. The use of Riemannian geometry to solve client-drift and rank-drift issues in federated fine-tuning is novel and effective, with well-defined methods like Riemannian Procrustes analysis and Riemannian gradient descent.
2. The paper provides rigorous proofs, including error bounds and convergence guarantees, which support the robustness of the RAFFT algorithm.

**Weaknesses:**

1. The efficiency of Riemannian operations is not fully quantified. A clearer breakdown of resource usage would help in assessing practical overhead.
2. Key parameters, like rank thresholds and learning rates, are not thoroughly analyzed. This could affect the method’s adaptability to various tasks and models.

**Questions:**

1. What are the specific computational trade-offs of Riemannian operations compared to simpler alternatives like FedAvg?
2. How does the proposed method perform under extreme non-IID data environments? Are there additional experiments or theoretical analyses that could demonstrate the stability and effectiveness of the RAFFT method across diverse data distribution scenarios?

---

> ### Author Response · Authors · 2024-11-24
> **Point-by-point response to the comments made by Reviewer aRkt**
>
> We thank this reviewer for the great suggestion!
>
> **Weaknesses 1:** The efficiency of Riemannian operations is not fully quantified. **A clearer breakdown of resource usage would help in assessing practical overhead.**
>
> **Answer**: Thank you for your suggestion. By following existing work [Beyond Efficiency: A Systematic Survey of Resource-Efficient Large Language Models](https://arxiv.org/pdf/2401.00625), we conduct the experiments to evaluate the resource usage of our RAFFT method across various metrics, including FLOPS (measure computational complexity), throughput (assess the processing rate), communication volume (quantify data transfer during training), training time, and accuracy. We have included these results in Appendix 7.6 table 34-35.
>
> As shown in the following table 1 and 2, the resource usage of 16 federated parameter-efficient fine-tuning methods is evaluated on the LLaMA 7B + MPQA task. Our RAFFT method achieves the highest throughput of 6,229.19 tokens/sec and the comparable FLOPS with the best baselines (72.6401 GFLOPS). In addition, RAFFT significantly outperforms all other LoRA-based methods with the lowest communication volume of 3,146,112 bytes, and achieves the highest accuracy of 90.8\%. Similarly, on the ViT + CIFAR-10 task, RAFFT achieves the best accuracy of 61.22\%, with the shortest training time of 3,222 seconds and a communication volume of 98,352 bytes. These results validate our RAFFT method is able to well balance computational and communication efficiency. A plausible explanation for this superior performance is that the Riemannian parameter matching and Riemannian gradient optimization enable faster convergence and improved accuracy, while the adaptive rank selection reduces redundant computation, making RAFFT a highly practical and efficient approach for real-world federated learning scenarios.
>
> **Table 1: Performance comparison of methods on LLaMA 7B + MPQA**
> | **Method**          | **FLOPS (GFLOPS)** | **Throughput (tokens/sec on H100)** | **Communication Volume (Number of parameters)** | **Training Time (s)** | **Accuracy (%)** |
> |----------------------|--------------------|------------------------------------|-----------------------------------|-----------------------|------------------|
> | Centralized LoRA    | 72.7557            | 5701.29                           | 4,194,304                        | 5432                  | 82.90           |
> | AdaLoRA             | 72.7557            | 6052.17                           | 4,194,816                        | 5503                  | 32.75           |
> | P-tuning v2         | 72.3243            | 5563.15                           | 524,288                          | 5664                  | 85.20           |
> | FedPrompt           | 72.3243            | 5100.28                           | 524,288                          | 5393                  | 87.35           |
> | FedPepTAO           | 72.3243            | 5829.55                           | 524,288                          | 5475                  | 87.60           |
> | PromptFL            | 72.3243            | 5546.20                           | 524,288                          | 5586                  | 87.40           |
> | PE_FL               | 72.3243            | 5710.59                           | 524,288                          | 5263                  | 87.25           |
> | SLoRA               | 72.7557            | 6199.18                           | 4,194,304                        | 5622                  | 84.10           |
> | HetLoRA             | 72.7557            | 5019.94                           | 4,194,304                        | 5229                  | 84.23           |
> | FedLoRA             | 72.7557            | 5100.28                           | 6,742,618,112                    | 5308                  | 87.74           |
> | FFA-LoRA            | 72.7557            | 5034.17                           | 4,194,304                        | 5358                  | 80.00           |
> | FLoRA               | 72.7557            | 4728.99                           | 4,194,304                        | 5293                  | 79.19           |
> | **RAFFT**           | **72.6401**        | **6229.19**                       | **3,146,112**                    | **5198**              | **90.80**       |
> | RAFFT-RGD           | 72.8020            | 5525.99                           | 4,194,816                        | 5151                  | 86.21           |
> | RAFFT-MR            | 72.7557            | 6392.10                           | 3,146,112                        | 5177                  | 86.73           |

---

> ### Author Response · Authors · 2024-11-24
> **Point-by-point response to the comments made by Reviewer aRkt (Continued)**
>
> **Table 2: Performance comparison of methods on ViT + CIFAR-10**
> | **Method**          | **FLOPS (GFLOPS)** | **Throughput (images/sec on 2080Ti)** | **Communication Volume (Number of parameters)** | **Training Time (s)** | **Accuracy (%)** |
> |----------------------|--------------------|---------------------------------------|-----------------------------------|-----------------------|------------------|
> | Centralized LoRA    | 41.9654            | 5730.05                              | 98,304                           | 3578                  | 60.48           |
> | AdaLoRA             | 41.9654            | 4722.30                              | 98,352                           | 3260                  | 60.92           |
> | SLoRA               | 41.9654            | 5739.46                              | 98,304                           | 3402                  | 58.89           |
> | HetLoRA             | 41.9654            | 5781.50                              | 98,304                           | 3387                  | 59.63           |
> | FedLoRA             | 41.9654            | 5663.19                              | 5,322,240                        | 3310                  | 59.11           |
> | FFA-LoRA            | 41.9654            | 5625.11                              | 98,304                           | 3412                  | 60.21           |
> | FLoRA               | 41.9654            | 5646.20                              | 98,304                           | 3249                  | 59.31           |
> | **RAFFT**           | **41.6335**        | **5721.55**                          | **78,962**                       | **3222**              | **61.22**       |
> | RAFFT-RGD           | 41.9927            | 5315.95                              | 98,304                           | 3288                  | 60.62           |
> | RAFFT-MR            | 41.9654            | 5681.28                              | 78,962                           | 3265                  | 59.91           |
>
> **Weaknesses 2:** Key parameters, like **rank thresholds** and **learning rates**, are not thoroughly analyzed. This could afect the method’s adaptability to various tasks and models.
>
> **Answer:** Thank you for pointing this out. In the submission, we have already conducted a detailed analysis of rank thresholds and learning rates, as shown in Tables 26-29 in Appendix 7.6.
>
> Tables 26-27 in Appendix 7.6 evaluate the performance effects by varying the rank selection threshold $\Theta$ from 0.6 to 1. It is observed that the performance score initially increases quickly and then become stable or even drop when $\Theta$ continuously increases. Initially, a large $\Theta$ can help generate a large rank and utilize the strength of more model parameters for ensuring better learning results. Later on, when $\Theta$ continues to increase and goes beyond some thresholds, the performance score becomes stable or drop. A rational guess is that excessive model parameters may bring redundant information or even noise, leading to performance degradation. It is important to find the optimal rank selection threshold to guarantee better  model performance.
>
> Tables 28-29 in Appendix 7.6 evaluate the performance effects by changing the learning rate ($LR$) from 0.0001 to 0.01. We have observed that for Roberta-based tasks, higher learning rates generally lead to better performance. SST-2, MRPC, and MPQA achieve the highest accuracy at $LR$=0.005 or 0.01. For LLaMA-based tasks, moderate learning rates, such as 0.001 or 0.0005, can produce the optimal results, while lower learning rates (e.g., 0.0001 or 0.0005) lead to better learning results for ViT-based tasks. This demonstrates that it is crucial to choose the optimal learning rates for the training to achieve the competitive performance.

---

> > ### Author Response · Authors · 2024-11-24
> > **Point-by-point response to the comments made by Reviewer aRkt (Continued)**
> >
> > **Question1:** What are the specific computational trade-offs of Riemannian operations compared to simpler alternatives like FedAvg.
> >
> > **Answer:** We have conducted the ablation study in the submission to compare three versions of the RAFFT model: RAFFT employs our Riemannian parameter matching technique for aggregation and RAFFT-RGD uses FedAvg as the aggregation method. All experiment results have been reported in all tables 1-2 and 4-13 in the submission. It has been observed that RAFFT achieves significant accuracy improvements, with a maximum increase of 5.75\% and an average improvement of 2.48\% compared to RAFFT-RGD. Despite these substantial accuracy gains, the running time of RAFFT only increases slightly, with an average time increase of just 1.15\%, demonstrating its efficiency and practicality in federated learning scenarios.
> > A reasonable explanation is that the Riemannian parameter matching introduce additional computational cost due to gradient computations on the manifold, but can effectively address the client-drift issue, resulting in the improvement of model performance, particularly in non-IID environments. These trade-offs make RAFFT highly efficient and practical for federated learning scenarios involving large-scale models and heterogeneous data distributions.
> >
> > **Question 2:** How does the proposed method perform under extreme non-IID data environments? Are there additional experiments or theoretical analyses that could demonstrate the stability and effectiveness of the RAFFT method across diverse data distribution scenarios?
> >
> > **Answer:** The following tables report how the test Accuracy changes under extreme non-IID data environment, by changing Dirichlet alpha $\alpha$ to 0.1 and 0.5. We use the Dirichlet distribution with $\alpha$ to partition the data into non-IID splits. Smaller values of $\alpha$ result in increasingly imbalanced data splits, leading to greater heterogeneity in the distribution of data across clients. In Appendix 7.6, Figure 3 shows the class distribution across clients on the MPQA dataset for $\alpha = 0.5$, $1$, $3$, and $5$. It clearly demonstrates that the class distribution becomes extremely imbalanced when $\alpha = 0.5$.
> > As shown in the table 3-4, our RAFFT method can still achieve superior performance and stability, under extreme non-IID data environment. For instance, in the LLaMA 7B + MPQA task with $\alpha=0.5$, RAFFT achieves the highest accuracy of 90.80\%, outperforming all baseline methods. Similarly, in the ViT + CIFAR-10 task with $\alpha=0.1$, RAFFT achieves the best accuracy of 61.22\%, surpassing all other methods. These results verify the effectiveness of our RAFFT method in handling high data heterogeneity, supported by its Riemannian parameter matching and Riemannian gradient descent techniques, which mitigates both client-drift and rank-drift issues and ensures stability across diverse data distributions.
> >
> > **Table 3: Performance comparison of methods on LLaMA 7B + MPQA with $\alpha = 0.5$.**
> > | **Method**       | **Accuracy (%)** | **Loss** | **Time (s)** |
> > |-------------------|------------------|----------|--------------|
> > | Centralized LoRA | 82.9000          | 0.2834   | 5432         |
> > | AdaLoRA          | 32.7500          | 10.5591  | 5503         |
> > | P-tuning v2      | 85.2000          | 0.2912   | 5664         |
> > | FedPrompt        | 87.3500          | 0.2639   | 5393         |
> > | FedPepTAO        | 87.6000          | 0.2584   | 5475         |
> > | PromptFL         | 87.4000          | 0.2574   | 5586         |
> > | PE_FL            | 87.2500          | 0.2661   | 5263         |
> > | SLoRA            | 84.1000          | 0.3372   | 5622         |
> > | HetLoRA          | 84.2300          | 0.3035   | 5229         |
> > | FedLoRA          | 87.7400          | 0.1664   | 5308         |
> > | FFA-LoRA         | 80.0000          | 0.3496   | 5358         |
> > | Fedkseed         | 82.4500          | 0.2918   | 5452         |
> > | FLoRA            | 79.1900          | 0.3016   | 5293         |
> > | **RAFFT**            | **90.8000**          | **0.2766**   | **5198**         |
> > | RAFFT-RGD        | 86.2100          | 0.2947   | 5151         |
> > | RAFFT-MR         | 86.7300          | 0.3004   | 5177         |

---

> > > ### Author Response · Authors · 2024-11-24
> > > **Point-by-point response to the comments made by Reviewer aRkt (Continued)**
> > >
> > > **Table 4: Performance comparison of methods on ViT + CIFAR-10 with $\alpha = 0.1$.**
> > > | **Method**       | **Accuracy (%)** | **Loss** | **Time (s)** |
> > > |-------------------|------------------|----------|--------------|
> > > | Centralized LoRA | 60.4800          | 1.0437   | 3578         |
> > > | AdaLoRA          | 60.9200          | 1.1428   | 3260         |
> > > | SLoRA            | 58.8900          | 0.8824   | 3402         |
> > > | HetLoRA          | 59.6300          | 1.0270   | 3387         |
> > > | FedLoRA          | 59.1100          | 0.9824   | 3310         |
> > > | FFA-LoRA         | 60.2100          | 0.9651   | 3412         |
> > > | FLoRA            | 59.3100          | 1.0490   | 3249         |
> > > | **RAFFT**            | **61.2200**          | **0.8544**   | **3222**         |
> > > | RAFFT-RGD        | 60.6200          | 0.9478   | 3288         |
> > > | RAFFT-MR         | 59.9100          | 1.0032   | 3265         |

---

> > > > ### Author Response · Authors · 2024-11-25
> > > > **Thanks to Reviewer aRkt**
> > > >
> > > > We would like to thank the reviewer for taking the time to review our paper and for the valuable comments. We hope our response has adequately addressed your concerns raised in the review.
> > > >
> > > > Please kindly let us know if anything is unclear. We truly appreciate this opportunity to improve our work and shall be most grateful for any feedback you could give us.

---

> > > > > ### Author Response · Authors · 2024-11-29
> > > > > **Thanks to Reviewer aRkt**
> > > > >
> > > > > Dear Reviewer aRkt
> > > > >
> > > > > We would like to thank you again for your valuable time and thoughtful and constructive comments. We have
> > > > > tried our best to clarify and address the concerns and comments in the initial response.
> > > > >
> > > > > We realize that you may have a busy schedule, but given the short discussion period, we decided to reach out
> > > > > again to see if you could review our responses. We would be happy to provide further clarifications if that helps.

---

> > > > > > ### Comment · Reviewer_aRkt · 2024-12-02
> > > > > >
> > > > > > Thank you for your response with the detailed explanations and experiments. Since no other new significant info is revealed, I would like to keep my score as it is.

---

### Official Review · Reviewer_51dy · 2024-11-02

**Soundness:** 3
**Presentation:** 1
**Contribution:** 3
**Rating:** 3
**Confidence:** 2

**Summary:**

This paper introduces RAFFT, a Riemannian LoRA algorithm with adaptive rank for federated fine-tuning of foundation models (FFT-FM). RAFFT effectively addresses the client-drift and rank-drift issues while significantly improving computational efficiency. To resolve the client-drift issue, the authors propose a Riemannian parameter matching method based on Riemannian Procrustes analysis to align local parameter matrices. To mitigate the rank-drift issue, they extend stochastic gradient descent (SGD) algorithms onto a Riemannian manifold.

**Strengths:**

1. The problem addressed is significant and appealing, focusing on a crucial aspect of LLM federated learning.
2. The paper applies a novel math technique to address the client-drift and rank-drift issues.

**Weaknesses:**

1. The writing quality is poor, with too many long sentences that are difficult to follow. Additionally, the paper includes many unexplained abbreviations, such as FedAvg, RAFFT, and RGD, which appear in the introduction without clarification.
2. The authors need to provide clear definitions of the client-drift and rank-drift issues. Referring readers to external works, as done with "Please refer to Sun et al. (2024a); Wang et al. (2024) for the definition of the specific client-drift issue in the FFT-FM," is inadequate. Readers need direct definitions within the paper to understand the problems being addressed.
3. The main text requires rigorous definitions of notations, as it is difficult to follow the proofs of the main results without them.
4. The paper lacks necessary lemmas that would aid in understanding the proof sketches.

**Questions:**

See weakness.

---

> ### Author Response · Authors · 2024-11-24
> **Point-by-point response to the comments made by Reviewer 51dy**
>
> We thank this reviewer for the commnets.
>
> **Weaknesses 1:** The writing quality is poor, with too many long sentences that are difcult to follow. Additionally, the paper includes many unexplained abbreviations, such as FedAvg, RAFFT, and RGD, which appear in the introduction without clarification.
>
> **Answer:** Thank you for your valuable feedback on our work. The following table 1 presents the definitions of all the abbreviations used in the submission.
>
> **Table 1: List of abbreviations used in the paper**
>
> | **Abbreviation** | **Definition** |
> |-------------------|----------------|
> | FFT              | Federated Fine-Tuning |
> | FT               | Fine-Tuning |
> | FFT-FM           | Federated Fine-Tuning of Foundation Models |
> | FT-FM            | Fine-Tuning of Foundation Models |
> | CFT              | Centralized Fine-Tuning |
> | RAFFT-RGD        | Variants of the RAFFT model, using RGD for updates and FedAvg for aggregation |
> | RAFFT-MR         | Using conventional optimization methods such as SGD, applying our aggregation method |
> | SVD              | Singular Value Decomposition |
> | FedAvg           | Federated Averaging Algorithm |
> | RAFFT            | Our proposed algorithm, a Riemannian LoRA algorithm with adaptive rank for federated fine-tuning of foundation models. |
> | RGD              | Riemannian Gradient Descent |
> | LoRA             | Rank-Adaptive Low-Rank Adaptation |
> | PEFT             | Parameter-Efficient Fine-Tuning |
> | FM               | Foundation Models |
>
> **Weaknesses 2:** The authors need to provide clear definitions of the client-drift and rank-drift issues.
>
> **Answer**: **Client-drift issue:** The client-drift issue is due to the aggregation of low-rank matrices in the FFT-FM. Here, we introduce an illustrative example with two clients to better understand the client-drift issue.
>
> Concretely, if the clients locally fine-tune on full parameters and the server aggregates with FedAvg, the new global model parameters can be expressed as follows.
>
> \begin{equation}
>     \mathbf{W} = \mathbf{W}^0 +\frac{1}{2} (\Delta \mathbf{W}^1+\Delta \mathbf{W}^2)
> \end{equation}
>
> In our work, the clients locally fine-tune on both low-rank parameter matrices $\mathbf{U}^k$ and $\mathbf{V}^k$ and diagonal matrix $\mathbf{\Sigma}^k$ to determine the rank. The server uses FedAvg to aggregate the low-rank matrices and diagonal matrix. The local model updates $\Delta \mathbf{W}^k$ are represented as:
>
> \begin{equation}
>     \Delta \mathbf{W}^k = \mathbf{U}^k \mathbf{\Sigma}^k \mathbf{V}^k,  \quad k ∈ \\{1, 2\\}
> \end{equation}
>
> The expected global model parameter updates $\Delta \mathbf{W}$ can be expressed as follows.
>
> \begin{equation}
>     \Delta \mathbf{W} = \frac{1}{2} (\Delta \mathbf{W}^1+\Delta \mathbf{W}^2) = \frac{1}{2} (\mathbf{U}^1 \mathbf{\Sigma}^1 \mathbf{V}^1 + \mathbf{U}^2 \mathbf{\Sigma}^2 \mathbf{V}^2)
> \end{equation}
>
> After using FedAvg to aggregate the local matrices $\mathbf{U}^k, \mathbf{\Sigma}^k, \mathbf{V}^k$, the server produces
>
> $$
> \begin{align\*}
>     \Delta \tilde{\mathbf{W}} & = \frac{1}{2}\left(\mathbf{U}^{1}+\mathbf{U}^{2}\right) \times \frac{1}{2}\left(\mathbf{\Sigma}^{1}+\mathbf{\Sigma}^{2}\right) \times \frac{1}{2}\left(\mathbf{V}^{1}+\mathbf{V}^{2}\right) \\\\\\ &= \frac{1}{2} (\mathbf{U}^1 \mathbf{\Sigma}^1 \mathbf{V}^1 + \mathbf{U}^2 \mathbf{\Sigma}^2 \mathbf{V}^2)\underline{-\frac{3}{8} \mathbf{U}^1 \mathbf{\Sigma}^1 \mathbf{V}^1 - \frac{3}{8} \mathbf{U}^2 \mathbf{\Sigma}^2 \mathbf{V}^2 + \frac{1}{8} \mathbf{U}^1 \mathbf{\Sigma}^1 \mathbf{V}^2 + \frac{1}{8} \mathbf{U}^1 \mathbf{\Sigma}^2 \mathbf{V}^1} \\\\\\ &+ \underline{\frac{1}{8} \mathbf{U}^1 \mathbf{\Sigma}^2\mathbf{V}^2 + \frac{1}{8} \mathbf{U}^2 \mathbf{\Sigma}^1\mathbf{V}^1 + \frac{1}{8} \mathbf{U}^2 \mathbf{\Sigma}^1\mathbf{V}^2 + \frac{1}{8} \mathbf{U}^2 \mathbf{\Sigma}^2\mathbf{V}^1} \\\\\\ &= \Delta \mathbf{W} \underline{-\frac{3}{8} \mathbf{U}^1 \mathbf{\Sigma}^1 \mathbf{V}^1 - \frac{3}{8} \mathbf{U}^2 \mathbf{\Sigma}^2 \mathbf{V}^2 + \frac{1}{8} \mathbf{U}^1 \mathbf{\Sigma}^1 \mathbf{V}^2 + \frac{1}{8} \mathbf{U}^1 \mathbf{\Sigma}^2 \mathbf{V}^1} \\\\\\ &+ \underline{\frac{1}{8} \mathbf{U}^1 \mathbf{\Sigma}^2\mathbf{V}^2 + \frac{1}{8} \mathbf{U}^2 \mathbf{\Sigma}^1\mathbf{V}^1 + \frac{1}{8} \mathbf{U}^2 \mathbf{\Sigma}^1\mathbf{V}^2 + \frac{1}{8} \mathbf{U}^2 \mathbf{\Sigma}^2\mathbf{V}^1}
> \end{align\*}
> $$
>
> where the underlined terms denote the difference between $\Delta \tilde{\mathbf{W}}$ and $\Delta \mathbf{W}$.

---

> ### Author Response · Authors · 2024-11-24
> **Point-by-point response to the comments made by Reviewer 51dy (Continued)**
>
> It is obvious that $\Delta \tilde{\mathbf{W}} \neq \Delta \mathbf{W}$ when $\mathbf{U}^{1} \neq \mathbf{U}^{2}$, $\mathbf{\Sigma}^{1} \neq \mathbf{\Sigma}^{2}$, and $\mathbf{V}^{1} \neq \mathbf{V}^{2}$ due to the data heterogeneity property of federated learning. Therefore, the **client-drift issue** is raised.
>
> $$
> \underbrace{\tilde{\mathbf{W}} = \mathbf{W}^0 + \Delta \tilde{\mathbf{W}}}_{\text{Parameter aggregation with SVD-based rank-adaptive LoRA + FedAvg}} \neq \underbrace{\mathbf{W}^0 + \Delta \mathbf{W} = \mathbf{W}}\_{\text{Ideal parameter aggregation with FedAvg on full parameter matrices}}
> $$
>
> The difference between $\Delta \tilde{\mathbf{W}}$ and $\Delta \mathbf{W}$ is mainly due to the noise introduced by the cross-products of LoRA modules from different clients. This difference may become more significant when (1) the number of local update steps between aggregations is large and (2) the local datasets are different across clients.
>
> We propose a Riemannian parameter matching method to match the local parameter matrices $\mathbf{U}^k$ and $\mathbf{V}^k$ on other clients with pivots $\mathbf{U}^1$ and $\mathbf{V}^1$ on client 1, in terms of their lengths and directions, i.e. $\tilde{\mathbf{U}}^k = \mathbf{U}^k \mathbf{R}^k \approx \mathbf{U}^1$ and $\tilde{\mathbf{V}}^k = \mathbf{S}^k \mathbf{V}^k \approx \mathbf{V}^1$.
> To maintain the consistency between low-rank parameter metrics before and after the Riemannian parameter matching, i.e., $ \mathbf{U}^2 \mathbf{\Sigma}^2 \mathbf{V}^2 = \tilde{\mathbf{U}}^2 \tilde{\mathbf{\Sigma}}^2 \tilde{\mathbf{V}}^2 \approx \mathbf{U}^1 \tilde{\mathbf{\Sigma}}^2 \mathbf{V}^1$, for ensuring the effectiveness of FFT-FM with rank-adaptive LoRA, we derive a modified diagonal matrix $\tilde{\mathbf{\Sigma}}^k $ for the other clients by performing the SVD on low-dimensional $r \times r$ matrices $(\mathbf{U}^1)^T \mathbf{U}^k$ and $(\mathbf{V}^1)^T \mathbf{V}^k$. Thus, the client-drift issue is resolved.
> \begin{equation}
> \begin{split}
> \Delta \tilde{\mathbf{W}} &=\frac{1}{2}\left(\mathbf{U}^{1}+ \tilde{\mathbf{U}^2}\right) \times \frac{1}{2}\left(\mathbf{\Sigma}^{1}+ \tilde{\mathbf{\Sigma}}^{2}\right) \times \frac{1}{2}\left(\mathbf{V}^{1}+ \tilde{\mathbf{V}^{K}}\right) \\
> &\approx\mathbf{U}^{1} \times \frac{1}{2}\left(\mathbf{\Sigma}^{1}+ \tilde{\mathbf{\Sigma}}^{2}\right) \times \mathbf{V}^{1}
> = \frac{1}{2}\left(\mathbf{U}^1 \mathbf{\Sigma}^1 \mathbf{V}^1 + \mathbf{U}^2 \mathbf{\Sigma}^2 \mathbf{V}^2\right) = \Delta \mathbf{W}
> \end{split}
> \end{equation}
> Based on the global diagonal matrix $\frac{1}{2}\left(\mathbf{\Sigma}^{1}+ \tilde{\mathbf{\Sigma}}^{2}\right)$, it is easy to find the optimal rank of the global parameter matrix, with aggregation on only the local low-rank matrices $\mathbf{U}^k, \tilde{\mathbf{\Sigma}}^{k}$, and $\mathbf{V}^k$.
>
> **Rank-drift issue:** The data heterogeneity is a common challenging problem in the federated learning. The collected data on different clients are Non-Independent Identically Distributed (Non-IID). Thus, the parameters of local models trained on the non-IID data are quite different from each other as well as the parameters of global model. Each local model may oscillate back and forth in different training rounds, leading to unstable and slow convergence and causing suboptimal model performance.
>
> At the same time, due to the heterogeneity of local models, the optimal ranks of local models are quite different from each other as well as the one of global model, raising the **rank-drift issue**. Similarly, the oscillation of the optimal rank of each local model can slow down the model convergence and degrade the model performance.
> For example, given two clients 1 and 2, in round 1, the local model on client 1/2 has the optimal rank 5/20. The server aggregates local models from clients to generate a global model based on FedAvg. The global model by model aggregation has the optimal rank 12. In round 2, after the global model are sent back to the clients, the clients begin the training with the global model with the optimal rank 12. Thus, the optimal rank of local models oscillate between 5/20 and 12. The rank oscillation may repeat in each training round, slowing down the model convergence.
>
> The following table presents a case study regarding the rank-drift in federated learning. In round 1, the optimal rank of the global model is 16. In round 2, the optimal rank is changed to 11. In subsequent rounds, the optimal rank continue to oscillate back and forth. The ranks of local models fluctuate around values such as 9.75, 10.17, and 10.53, while the rank of global modelgradually decreases and stabilizes at 8. These results confirm that rank-drift introduces instability into the training process, slowing convergence and degrading performance, which emphasizes the importance of our Riemannian gradient descent optimization approach to stabilize ranks and ensure efficient training.

---

> ### Author Response · Authors · 2024-11-24
> **Point-by-point response to the comments made by Reviewer 51dy (Continued)**
>
> **Table 1: Client and Server Ranks Over 25 Rounds on LLaMA 7B+MPQA**
> | Round (1-25) | Client Rank   | Server Rank   |
> |--------------|---------------|---------------|
> | 1            | 16            | 16            |
> | 2            | 9.20625       | 11            |
> | 3            | 9.153125      | 10            |
> | 4            | 9.75          | 10            |
> | 5            | 10.165625     | 9             |
> | 6            | 10.0046875    | 9             |
> | 7            | 9.88125       | 9             |
> | 8            | 9.9515625     | 9             |
> | 9            | 10.16875      | 9             |
> | 10           | 10.3921875    | 8             |
> | 11           | 10.4671875    | 8             |
> | 12           | 10.5265625    | 8             |
> | 13           | 10.50625      | 8             |
> | 14           | 10.51875      | 8             |
> | 15           | 10.5078125    | 8             |
> | 16           | 10.4546875    | 8             |
> | 17           | 10.39375      | 8             |
> | 18           | 10.2953125    | 8             |
> | 19           | 10.2125       | 8             |
> | 20           | 10.175        | 8             |
> | 21           | 10.1109375    | 8             |
> | 22           | 10.0296875    | 8             |
> | 23           | 9.98125       | 8             |
> | 24           | 9.9328125     | 8             |
> | 25           | 9.903125      | 8             |
> | 26           | 9.8640625     | 8             |

---

> ### Author Response · Authors · 2024-11-24
> **Point-by-point response to the comments made by Reviewer 51dy (Continued)**
>
> **Weaknesses 3:** The main text requires rigorous definitions of notations, as it is difficult to follow the proofs of the main results without them.
>
> **Answer**: The following table presents the definitions of all the notations used in the submission.
>
> **Table 2: Definitions of Notations**
>
> | **Notation** | **Explanation** |
> |--------------|------------------|
> | $\mathbf{W} \in \mathbb{R}^{m \times n}$ | Full parameter matrix of the foundation model |
> | $\mathbf{W}^k \in \mathbb{R}^{m \times n}$ | Local model parameter matrix for client $K$ |
> | $\mathbf{W}_0 \in \mathbb{R}^{m \times n}$ | Pre-trained foundation model parameters |
> | $\Delta \mathbf{W} \in \mathbb{R}^{m \times n}$ | Adapter parameters, used for fine-tuning |
> | $\Delta \mathbf{W}^k \in \mathbb{R}^{m \times n}$ | Adapter parameter matrix for client $K$ |
> | $\mathbf{U}^k \in \mathbb{R}^{m \times r}$, $\mathbf{V}^k \in \mathbb{R}^{r \times n}$ | Matrices representing the left and right singular vectors of $\Delta \mathbf{W}^k$ |
> | $\mathbf{\Sigma}^k \in \mathbb{R}^{r \times r}$ | Diagonal matrix containing the singular values of $\Delta \mathbf{W}^k$ |
> | $\mathbf{B} \in \mathbb{R}^{r \times n}$, $\mathbf{A} \in \mathbb{R}^{m \times r}$ | Low-rank matrices representing adapter parameters $\Delta \mathbf{W} = \mathbf{B} \mathbf{A}$ |
> | $r \in \mathbb{R}^{r \times r}$ | Rank of decomposition, much smaller than $m$ and $n$ |
> | $D = \\{D_1, \cdots, D_K\\}$ | The set of local training data from $K$ clients |
> | $K$ | The total number of clients involved in federated learning |
> | $n^{k}$ | The size of the local dataset \(D_k\), i.e., $n^{k} = \|D_k\|$ |
> | $n$ | The size of the total training data \(\mathbf{D}\), i.e., $n = n_1 + \dots + n_K$ |
> | $\mathcal{L}(\mathbf{W})$ | The global loss function to be minimized, aggregated over all clients |
> | $L^{k}(\mathbf{W})$ | The loss function for client $K$ |
> | $l_{i}(\mathbf{W})$ | The loss function for a single data sample $\{x_{i}, y_{i}\}$ |
> | $S$ | The set of orthogonal matrices on Riemann manifold |
> | $S_p$ | The Procrustes representation space on manifold $S$ |
> | $\mathbf{R}^k \in \mathbb{R}^{r \times r}$, $\mathbf{S}^k \in \mathbb{R}^{r \times r}$ | The symmetric positive matrix for Riemannian parameter matching |
> | $\mathbf{P}^k \in \mathbb{R}^{m \times r}$, $\mathbf{\Lambda}^k \in \mathbb{R}^{r \times r}$, $\mathbf{Q}^k \in \mathbb{R}^{r \times n}$ | Matrices obtained from the SVD of $(\mathbf{U}^1)^T \mathbf{U}^k$ |
> | $\mathbf{Y}^k$, $\mathbf{Z}^k$ | Auxiliary variables introduced to reformulate the Riemann problem |
> | $\tilde{\mathbf{U}}^k \in \mathbb{R}^{m \times r}$, $\tilde{\mathbf{V}}^k \in \mathbb{R}^{r \times n}$ | The aligned parameter matrices for client $k$ |
> | $\tilde{\mathbf{\Sigma}}^k$ | Modified version of diagonal matrix $\mathbf{\Sigma}^k$ for ensuring consistency |
> | $\lambda_{R_{\min}}, \lambda_{R_{\max}}$ | Minimum and maximum eigenvalues of the matrix $\mathbf{R}^k$ |
> | $\lambda_{S_{\min}}, \lambda_{S_{\max}}$ | Minimum and maximum eigenvalues of the matrix $\mathbf{S}^k$ |
> | $H,N$ | Constants bounding the Frobenius norms of matrices $\lambda_{R_{\min}}$ and $\lambda_{S_{\min}}$ |
> | $\Theta(r)$ | Singular value contribution rate |
> | $\varphi$ | Threshold value for determining rank |
> | $R(\mathbf{U}^k, \mathbf{V}^k)$ | Regularization term to enforce orthogonality of $\mathbf{U}^k$ and $\mathbf{V}^k$ |
> | $M$ | The set of fixed rank matrices on Riemann manifold |
> | $T_{M}$ | Tangent space of the Riemannian manifold $M$ at the point $\Delta \mathbf{W}$ |
> | $\mathbf{G} \in \mathbb{R}^{r \times r}$, $\mathbf{U}^k_p \in \mathbb{R}^{m \times r}$, $\mathbf{V}^k_p \in \mathbb{R}^{n \times r}$ | The tangent vector at $\Delta \mathbf{W}$ |
> | $\mathbf{U}^k_\perp$, $\mathbf{V}^k_\perp$ | The orthogonal complements of $\mathbf{U}^k$ and $\mathbf{V}^k$ |
> | $\nabla L^{k}(\Delta \mathbf{W}^k)$ | Gradient of the local objective function $L^k$ in Euclidean space |
> | $\Gamma L^{k}(\Delta \mathbf{W}^k)$ | Riemannian gradient of the local objective function $L^k$ |
> | $P$ | Orthogonal projection |
> | $\Gamma_{\mathbf{U}^k} L^{k}$, $\Gamma_{\mathbf{\Sigma}^k} L^{k}$, $\Gamma_{\mathbf{V}^k} L^{k}$ | The components of the Riemannian gradient at $\Delta \mathbf{W}^k$ |
> | $\eta_t$ | Learning rate |
> | $f()$ | Retraction function |
> | $t$ $(0 \leq t \leq C)$ | Federated learning round |
> | $c_g$ | Geodesic smoothness constant for the function $L^{k}$ |
> | $\zeta$ | Key geometric constant capturing the impact of manifold's curvature |

---

> ### Author Response · Authors · 2024-11-24
> **Point-by-point response to the comments made by Reviewer 51dy (Continued)**
>
> **Weaknesses 4:** The paper lacks necessary lemmas that would aid in understanding the proof sketches.
>
> **Answer**: Thanks for the kind suggestion. We will include the following lemmas in the submission for better understanding of the theorem proof.  Here, we will introduce and explain the lemmas one by one.
>
> **Lemmas 1-3 are the preliminary steps of proof of Theorem 1**. Lemmas 1-3 and Theorem 1 together derive the upper bound on the Frobenius norm error for off-diagonal elements after an orthogonal transformation in Eq.(14) in the submission. We have included all of the proof of these lemmas and theorem in Appendix 7.2 ``Approximation Matrix Upper Bound".
>
> Lemma 1 demonstrates the Frobenius norm of a matrix orthogonally invariant. It is used to conduct the approximation error analysis in Eq.(30).
>
> **Lemma 1 (Orthogonal Matrix and Frobenius Norm Preservation)**:
> Let $U \in \mathbb{R}^{n \times n}$ be an orthogonal matrix, i.e., $U^\top U = I$. For any $A \in \mathbb{R}^{n \times m}$, the Frobenius norm $\|A\|_F$ satisfies:
> $\|U^\top A\|_F = \|A\|_F.$
>
> Please refer to [1] for the detailed proof.
>
> Lemma 2 is the extension of the Cauchy-Schwarz inequality on matrices. It is utilized to derive an upper bound of Eq.(31) and further the bounded error of matrix approximation.
>
> **Lemma 2 (Cauchy-Schwarz Inequality for Frobenius Norm)**:
> For any $A, B \in \mathbb{R}^{n \times m}$, the Frobenius norms $\|A\|_F$ and  $\|B\|_F$ satisfy:
> $
> |\text{tr}(A^\top B)| \leq \|A\|_F \|B\|_F.
> $
> where $\text{tr}(\cdot)$ is the trace of a matrix.
>
> Please refer to [2] for the detailed proof.
>
> Lemma 3 validates how the condition number of  $S^k$ governs the amplification of off-diagonal errors during inversion, thereby quantifying the impact on the approximation error. It is employed to obtain the total approximation error bound in Eq. (35) in Theorem 1.
>
> **Lemma 3 (Condition Number and Off-Diagonal Error Amplification)**:
> Let $S \in \mathbb{R}^{n \times n}$ be a symmetric positive definite matrix with condition number $\kappa(S) = \lambda_{\max}(S) / \lambda_{\min}(S)$. Then the Frobenius norm of the off-diagonal part of the inverse matrix is bounded by:
> $\|S^{-1}\_{\text{off-diagonal}}\|_F \leq \kappa(S) \|S\_{\text{off-diagonal}}\|_F.$
> where $S^{-1}$ is the inverse of the matrix $S$.
>
> Please refer to [3] for the detailed proof.
>
> Theorem 1 obtains the upper bound on the Frobenius norm error for off-diagonal elements after an orthogonal transformation in Eq.(14) in the submission.
> Eq.(14) consists of three terms: the aligned parameter matrices $\mathbf{U}^k \mathbf{R}^k$ and $\mathbf{V}^k \mathbf{S}^k$ for client $k$ and the modified version $\tilde{\mathbf{\Sigma}}^k$ of diagonal matrix $\mathbf{\Sigma}^k$.
> The approximation error in Eq.(15) in Theorem 1 is raised by all the above three terms together.
> Eq.(29) leverages Lemma 1 to separate three components from the left side in Eq.(15).
> Eqs.(30)-(33) utilize Lemma 2 to estimate the error bounds led by $\mathbf{U}^k \mathbf{R}^k$ and $\mathbf{V}^k \mathbf{S}^k$.
> Eq.(35) employs Lemma 3 to derive the amplification of off-diagonal errors by $\|\tilde{\mathbf{\Sigma}}^k - \mathbf{\Sigma}^k \|_F$.
>
> **Lemmas 4-8 are the preliminary steps of proof of Theorem 2**. Lemmas 4-8 and Theorem 2 together demonstrates that the Riemannian gradient descent (RGD) optimization on the Riemannian manifold ensures the rank invariance during the local update process. We have included all of the proof of these lemmas and theorem in Appendix 7.3 ``Riemann Gradient and Retraction".
>
> Lemma 4 analyzes the correlation between a submatrix  $\mathbf{w}_{11}$ of a matrix $\mathbf{W}$ and other submatrices of $\mathbf{W}$, which is essential for constructing the local defining function for the smooth manifold $M_r$. Therefore, it underpins the theoretical foundation of the matrix decomposition in Eq.(46).
>
> **Lemma 4 (Partitioning and Submatrix Inversion)**:
> Let $W \in \mathbb{R}^{m \times n}$ with rank  $r$. If $w_{11} \in \mathbb{R}^{r \times r}$ is an invertible submatrix of $W$, then $W$ can be partitioned as:
>
> \begin{equation}
> W = \begin{bmatrix} w_{11} & w_{12} \\\\\\ w_{21} & w_{22} \end{bmatrix},
> \end{equation}
> where $w_{12}$, $w_{21}$, and $w_{22}$ are submatrices. The submatrix $X \in \mathbb{R}^{r \times (n - r)}$ satisfies:
> \begin{equation}
> X = w_{11}^{-1} w_{12}, \quad w_{22} = w_{21} X = w_{21} w_{11}^{-1} w_{12}.
> \end{equation}
> Please refer to [4] for the detailed proof.

---

> ### Author Response · Authors · 2024-11-24
> **Point-by-point response to the comments made by Reviewer 51dy (Continued)**
>
> Lemma 5 ensures that the tangent space $T_M$ is well-defined and $M_r$ is an embedded submanifold of $\mathbb{R}^{m \times n}$, and thus introduces Definition 1 in the submission.
>
> **Lemma 5 (Differentiability of Local Defining Function):**
> Let $h: \mathbb{R}^{(m-r) \times (n-r)} \to \mathbb{R}^{m \times n}$ be defined as:
> $h(Y) = Y_{22} - Y_{21} Y_{11}^{-1} Y_{12}$
> where $Y_{ij}$ are submatrices of $Y$. Then $h$ is smooth and has an inverse mapping $h^{-1}$ such that $\text{ker}(Dh(Y))$ is a linear subspace that spans $\mathbb{R}^{m \times n}$.
>
> Please refer to [4] for detailed proof.
>
> Lemma 6 characterizes the dynamics of the corresponding smooth curves $\mathbf{U}(t)$ and $\mathbf{V}(t)$ of any vectors within $T_\mathbf{U}St(m, r)$ and  $T_\mathbf{V}St(n, r)$ respectively, when they evolve smoothly on their respective manifolds.
>
> **Lemma 6 (Tangent Space on the Stiefel Manifold):**
> Let $\mathbf{U}^k \in \mathbb{R}^{m \times r}$ and $\mathbf{V}^k \in \mathbb{R}^{n \times r}$ lie on the Stiefel manifolds $\text{St}(m, r)$ and $\text{St}(n, r)$, respectively. For any $\Omega \in \text{Skew}(r)$ and $B \in \mathbb{R}^{(m-n) \times r}$, the tangent space velocities $\mathbf{U}^{k\prime}(0)$ and $\mathbf{V}^{k\prime}(0)$ are given by:
> \begin{equation}
> \mathbf{U}^{k\prime}(0) = \mathbf{U}^k \Omega + \mathbf{U}^k_\perp B, \quad \mathbf{V}^{k\prime}(0) = \mathbf{V}^k \Omega' + \mathbf{V}^k_\perp C,
> \end{equation}
> where $\mathbf{U}^k_\perp$ and $\mathbf{V}^k_\perp$ are orthogonal complements.
>
> Please refer to [5] for the detailed proof.
>
> Lemma 7 decomposes $Z^k$ into components within the tangent space, ensuring the projection results in a matrix that respects the manifold structure of a manifold $M_r$. It is utilized to compute the gradients in our proposed Riemannian gradient descent algorithm.
>
> **Lemma 7 (Orthogonal Projection onto Tangent Space):**
> Let $\Delta W^k = U^k \Sigma^k (V^k)^\top$ represent a point on $M_r$. The orthogonal projection of a matrix $Z^k \in \mathbb{R}^{m \times n}$ onto the tangent space $T_{M_r}$ at $\Delta W^k$ is given by:
> \begin{equation}
> \text{Proj}\_{\Delta W^k}(Z^k) = U^k G (V^k)^\top + U^k_\perp (V^k_\perp)^\top + U^k (V^k_\perp)^\top,
> \end{equation}
> where $G$ is a general matrix relaed to the variation of $U^k$ and $V^k$ along their tangent directions, $U^k_\perp$ and $V^k_\perp$ denote the components of $Z^k$ orthogonal to $U^k$ and $V^k$, respectively.
>
> Please refer to [5] for detailed proof.
>
> Lemma 8 provides the theoretical foundation for projecting the updated matrix back onto the manifold after applying a gradient step in Eq.(69). The Eckart-Young-Mirsky theorem ensures that the retraction operation preserves the rank structure.
>
> **Lemma 8 (Retraction for Fixed-Rank Matrices):**
> Let $H \in \mathbb{R}^{m \times n}$ represent the perturbation in the ambient space. The retraction function $f$ maps the tangent space $T_M$ to the manifold $M$ by minimizing the Frobenius norm distance:
> \begin{equation}
> f(H) = \arg \min_{\Delta W \in M} \| \Delta W^k + H - \Delta W \|_F^2.
> \end{equation}
> Under the constraint of fixed rank $r$, the retraction is given by the truncated singular value decomposition of $\Delta W^k + H$.
>
> Please refer to [1] for the detailed proof.
>
> Theorem 2 illustrates that the RGD optimization on the Riemannian manifold ensures the rank invariance during the local update process.
>
> First, Eq.(43) defines a manifold $M_r$ as an embedded submanifold of $\mathbb{R}^{m \times n}$, where the rank constraint $r$ is preserved through a matrix partitioning approach. Lemma 4 is leveraged to decompose the parameter matrices \( W \) with rank $r$ into four submatrices $w_{11}, w_{12}, w_{21}$ and $w_{22}$, while these submatrices still maintain the invariance of $r$, as shown in Eq.(46).
>
> Second, Eq.(47) defines a local parametrization function $h$ to formalize the manifold structure and ensure smooth updates. Its differentiability and inverse properties, supported by Lemma 5, guarantee that the manifold $M_r$ is smooth and that local updates remain within the manifold.
>
> Third, during the RGD  optimization process, tangent vectors are characterized through the evolution of $U^k$ and $V^k$ on the Stiefel manifolds $\text{St}(m, r)$ and $\text{St}(n, r)$. Lemma 6 characterizes the smooth dynamics of $\mathbf{U}^{k\prime}$ and $\mathbf{U}^{k\prime}$ in Eq.(53), ensuring orthogonality and the rank-preserving properties of these updates.

---

> ### Author Response · Authors · 2024-11-24
> **Point-by-point response to the comments made by Reviewer 51dy (Continued)**
>
> Fourth, Eq.(59) gives the orthogonal projection of any update direction $Z^k$ onto the tangent space $T_{M_r}$ at $\Delta W^k$. Lemma 7 guarantees that the projection respects the manifold structure and maintains the rank $r$. The resulting Riemannian gradient in Eq.(65) incorporates these projections to guide updates within the manifold.
>
> Finally, to construct the retraction function that maps the tangent space updates back onto the manifold, Eq.(68) minimizes the distance between the perturbed point and the manifold. During this process, Lemma 8 uses the truncated singular value decomposition (SVD) to ensure that the rank is preserved throughout the projection. The QR factorizations of the augmented matrices $[U^k, U_p]$ and $[V^k, V_p]$ in Eq.(70), yield the orthogonal matrices $Q_U$, $Q_V$ and upper triangular matrices $R_U$, $R_V$, which facilitate the computation of the perturbed matrix $\Delta W^k + H$ in Eq.(71). Applying SVD to this perturbed matrix produces the retracted point $f(H)$, as described in Eq.(72). Eq.(73) provides the updated components $(U^{k+1}, \Sigma^{k+1}, V^{k+1})$ in the RGD optimization.
>
> In summary, Theorem 2 rigorously proves the RGD optimization process on the Riemannian manifold $M_r$, ensuring rank invariance throughout the local update process while seamlessly integrating gradient computation, projection, and retraction operations.
>
> Lemma 9 establishes a key inequality that relates the geodesic distance between successive points on a Riemannian manifold $\mathcal{M}$ to the reduction in the objective function during gradient-based optimization. It captures the relationship between the gradient $g^{(t)}$, the geodesic distance $\text{dist}(x, x^{(t)})$, and the step size $\alpha$, while accounting for the manifold's curvature through the factor $\zeta(\kappa_{\min}, \text{dist}(x, x^{(t)}))$.
>
> **Lemma 9 (Geodesic Distance Properties):**
> For any Riemannian manifold $\mathcal{M}$ where the sectional curvature is lower bounded by $\kappa_{\min}$ and any point $x, x^{(t)} \in \mathcal{M}$, the update $x^{(t+1)} = \text{Exp}\_{x^{(t)}}(-\alpha g^{(t)})$ with $g^{(t)} \in T_{x^{(t)}} \mathcal{M}$ satisfies
>
> \begin{equation}
> \left\langle -g^{(t)}, \text{Exp}_{x^{(t)}}^{-1}(x) \right\rangle \leq \frac{1}{2 \alpha} \left(\text{dist}^2(x, x^{(t)}) - \text{dist}^2(x, x^{(t+1)}) \right) + \frac{\zeta(\kappa\_{\min}, \text{dist}(x, x^{(t)})) \alpha}{2} \|g^{(t)}\|^2
> \end{equation}
>
> Where $\text{Exp}\_{x^{(t)}}(x)$ is the exponentail map that projects a tangent vctor x onto the manifold, $\text{Exp}_{x^{(t)}}^{-1}(x)$ is the inverse exponential map that maps a point x back to the tangent space, $\alpha$ is the learning rate controlling the magnitude of the update, $\text{dist}(x, x^{(t+1)})$ denotes the geodesic distance between points x and $x^{(t+1)}$ on the manifold, and $\zeta(\kappa, c) = \frac{\sqrt{|\kappa|c}}{\tanh\left( \sqrt{|\kappa|c} \right)}$ is the curvature adjustment for distance.
>
> Please refer to [6] for the detailed proof.
>
> Theorems 3 and 4 conduct the convergence analysis of our RAFFT algorithm based on the RGD optimization on the Riemannian manifold in the settings of both non-convex and geodesically convex.
>
> For the non-convex case, Definition 2 introduces the smoothness property of the loss function $L$ on a manifold, ensuring bounded gradients in Eq.(74). Theorem 3 establishes the convergence of the optimization process, demonstrating that the expected squared norm of the Riemannian gradient decreases to a bounded region (Equation 79). This result is achieved by projecting updated points back to the manifold using the retraction operator and measuring distances via the geodesic inverse in Eqs.(80)-(82). By leveraging the Lipschitz smoothness property (Equation 75), the inequality in Equation 83 bounds the reduction in the loss function $L$ over iterations. Summing these inequalities and selecting appropriate step sizes ensures convergence, as shown in Eq.(85).
>
> For the convex scenario, Definition 3 ensures the function $L$ satisfies a convex-like property along geodesics, enabling convergence guarantees. For geodesically convex functions, Theorem 4 extends the analysis by proving the convergence rate for the difference between the current and optimal loss values. Lemma 9 is used to bound the geodesic distance between successive points and relates it to the reduction in loss in Eqs.(86)-(97). The result concludes with a convergence guarantee dependent on the curvature constant $\zeta$ and the geodesic distance to the optimal solution.

---

> > ### Author Response · Authors · 2024-11-24
> > **Point-by-point response to the comments made by Reviewer 51dy (Continued)**
> >
> > REFERENCES:
> >
> > [1] Gene H. Golub and Charles F. Van Loan. Matrix Computations - 4th Edition. Johns Hopkins
> > University Press, Philadelphia, PA, 2013.
> >
> > [2] Roger A. Horn and Charles R. Johnson. Matrix Analysis. Cambridge University Press, second
> > edition, 2012.
> >
> > [3] Lloyd N. Trefethen and David Bau. Numerical Linear Algebra. SIAM, 1997.
> >
> > [4] Nicolas Boumal. An Introduction to Optimization on Smooth Manifolds. ´Ecole Polytechnique
> > F´ed´erale de Lausanne, March 2023.
> >
> > [5] P.-A. Absil, R. Mahony, and R. Sepulchre. Optimization Algorithms on Matrix Manifolds. Princeton
> > University Press, Princeton, NJ, 2008.
> >
> > [6] Hongyi Zhang and Suvrit Sra. First-order methods for geodesically convex optimization, 2016.

---

> > > ### Author Response · Authors · 2024-11-25
> > > **Thanks to Reviewer 51dy**
> > >
> > > We would like to thank the reviewer for taking the time to review our paper and for the valuable comments. We hope our response has adequately addressed your concerns raised in the review.
> > >
> > > Please kindly let us know if anything is unclear. We truly appreciate this opportunity to improve our work and shall be most grateful for any feedback you could give us.

---

> > > > ### Author Response · Authors · 2024-11-29
> > > > **Thanks to Reviewer 51dy**
> > > >
> > > > Dear Reviewer 51dy,
> > > >
> > > > We would like to thank you again for your valuable time and thoughtful and constructive comments. We have
> > > > tried our best to clarify and address the concerns and comments in the initial response.
> > > >
> > > > We realize that you may have a busy schedule, but given the short discussion period, we decided to reach out
> > > > again to see if you could review our responses. We would be happy to provide further clarifications if that helps.

---

> > > > > ### Comment · Reviewer_51dy · 2024-11-29
> > > > >
> > > > > Thank you for your detailed reply. While I appreciate the effort put into addressing my concerns, I still have several reservations that need to be addressed for me to consider raising my score:
> > > > >
> > > > > Revised Paper Submission: It is essential to provide a revised version of your paper that includes all necessary explanations and definitions directly within the text. These should not be limited to the response document. The paper must meet the formatting and writing standards expected at ICLR.
> > > > >
> > > > > Abbreviations: For unexplained abbreviations, I strongly recommend avoiding a separate list in the paper. To ensure clarity and avoid ambiguity, please provide the full form of each abbreviation when it is first introduced.
> > > > >
> > > > > Definitions of Client-Drift and Rank-Drift: If client-drift and rank-drift are central problems your work aims to address, these concepts should be clearly defined and prominently highlighted in the introduction or background sections. Currently, it is challenging to locate relevant parts due to the paper’s lack of clarity in its writing.
> > > > >
> > > > > Explanation of Lemmas: All lemmas and their explanations should be integrated into the paper for better understanding. Additionally, it seems that many of the lemmas are cited from other works. To emphasize the novelty of your paper, I suggest explicitly highlighting any new techniques or results that set your contributions apart.
> > > > >
> > > > > Structure and Coherence: To improve the paper’s readability, I highly recommend using appropriate subtitles or structured sections. Currently, Sections 3 and 4 feel like overly long, unbroken blocks of text, which significantly hampers comprehension. Dividing these sections into smaller, coherent blocks will enhance both clarity and readability.
> > > > >
> > > > > Thank you for considering these suggestions. I look forward to reviewing the revised version of your paper.

---

> > > > > > ### Author Response · Authors · 2024-11-29
> > > > > > **Point-by-point response to the comments made by Reviewer 51dy**
> > > > > >
> > > > > > Thank you for your follow-up feedback. We have addressed the majority of the concerns you raised. Please refer to the latest revised version of our paper for detailed updates.
> > > > > >
> > > > > > **Abbreviations:** In our submission, we have ensured that the full form of each abbreviation is provided when it is first introduced in the text. For example, please refer to the following lines in our paper: LoRA (line 11), PEFT (line 11), FM (line 12), FFT-FM (line 16), RAFFT (line 16), SVD (line 20), RGD (line 25), FT-FM (line 36), CFT (line 53), RAFFT-RGD (line 433), RAFFT-MR (line 436).
> > > > > >
> > > > > > **Definitions of Client-Drift and Rank-Drift:** For the definitions of Client-Drift and Rank-Drift, we have provided clear and explicit definitions in our revised submission. Specifically: **Client-Drift is defined in line 75. Rank-Drift is defined in line 91.**
> > > > > >
> > > > > > **Explanation of Lemmas:**  We have included all the lemmas and their detailed explanations in Appendix 7.2, 7.3, and 7.4 of the revised submission. Specifically:
> > > > > >
> > > > > > Lemmas 1-3 are essential preliminary steps for the proof of Theorem 1. Together, these lemmas and the theorem derive the upper bound on the Frobenius norm error of our Riemannian parameter matching method in Eq. (14).
> > > > > >
> > > > > > Lemmas 4-8 serve as foundational steps for the proof of Theorem 2. These lemmas, combined with Theorem 2, demonstrate that Riemannian gradient descent (RGD) optimization on the Riemannian manifold ensures rank invariance during the local update process.
> > > > > >
> > > > > > Lemma 9, Theorems 3 and 4 conduct the convergence analysis of our RAFFT algorithm based on the RGD op-
> > > > > > timization on the Riemannian manifold in the settings of both non-convex and geodesically convex.
> > > > > >
> > > > > > **Structure and Coherence:**  We have addressed your concern by restructuring Sections 3 and 4 into smaller, coherent subsections to improve clarity and readability. In the next version of our paper, these sections will be organized as follows:
> > > > > >
> > > > > > Section 3:
> > > > > >
> > > > > > Lines 196-269: Riemannian Parameter Matching Objective and Efficient Optimization
> > > > > >
> > > > > > Lines 270-280: Approximation Error Analysis
> > > > > >
> > > > > > Section 4:
> > > > > >
> > > > > > Lines 321-360: Riemannian Gradient Optimization
> > > > > >
> > > > > > Lines 361-383: Convergence Analysis

---

> > > > > > > ### Author Response · Authors · 2024-12-01
> > > > > > > **Thanks to Reviewer 51dy**
> > > > > > >
> > > > > > > We would like to thank the reviewer for taking the time to review our paper and for the valuable comments. We have carefully addressed all your concerns in the revised version of our paper. We invite you to review the revised PDF. If you have any other concerns, please do not hesitate to let us know.

---

### Official Review · Reviewer_WxWw · 2024-11-03

**Soundness:** 3
**Presentation:** 3
**Contribution:** 2
**Rating:** 5
**Confidence:** 3

**Summary:**

This paper proposes a Riemannian low-rank adaptation (LoRA) method with adaptive rank, tailored to federated fine-tuning of foundation models. The key ida is to utilize Riemannian Procustes analysis and Riemannian manifold theory to tackle the client drift issue and the rank drift issue, respectively. The authors demonstrate that the Riemannian gradient descent optimization on the Riemannian manifold ensures convergence. Experimental results on 3 different benchmark datasets show the advantage of the proposed algorithm.

**Strengths:**

1. The motivation of the paper is clear, and is well written.

2. Federated fine-tuning of foundation models using LoRA is an important topic.

3. The algorithm is developed with some theoretical insights, including Riemannian Procrustes analysis and Riemannian manifold theory. The paper also provides convergence analysis of the proposed RGD algorithm.

**Weaknesses:**

1. Many of the presentations of the paper are similar with the ones in the (Sun et al., 2024) paper. As an example, equation 1 in the authors's paper is also presented in equation 3 of (Sun et al., 2024). It is not clear which part is new to this work. I feel that the authors need to add more details regarding the difference with the previous work.

(Sun et al., 2024) "IMPROVING LORA IN PRIVACY-PRESERVING FEDERATED LEARNING", ICLR 2024.

2. Also, Reiemannian LoRA has been studied in a centralized setting in the following paper:

Riemannian Preconditioned LoRA for Fine-Tuning Foundation Models, ICML 2024.

The authors should also cite this paper, and describe the technical contributions compared to this work. For example, how is the authors' work different from combining the above two papers? What are the new challenges and technical contributions compared with the combination of these works?

**Questions:**

Please refer to the weakness above.

---

> ### Author Response · Authors · 2024-11-24
> **Point-by-point response to the comments made by Reviewer WxWw**
>
> We thank this reviewer for the helpful comments.
>
> **Weakness 1:** Many of the presentations of the paper are similar with the ones in the (Sun et al., 2024) paper. As an example, equation 1 in the authors's paper is also presented in equation 3 of (Sun et al., 2024). It is not clear which part is new to this work. I feel that the authors need to add more details regarding the difference with the previous work.
>
> **Answer:** The problem, challenges, and techniques in our paper significantly differ from the above ICLR 2024 (Sun et al., 2024), although both papers have the client-drift issue and the reason of the issue in both papers are the same, due to the aggregation of low-rank matrices in the FFT-FM.
>
> The ICLR 2024 paper conducts this issue raised by only low-rank parameter matrices $\mathbf{A}^k$ and $\mathbf{B}^k$, but our work addresses a more challenging problem by considering both low-rank parameter matrices $\mathbf{U}^k$ and $\mathbf{V}^k$ and diagonal matrix $\mathbf{\Sigma}^k$ to determine the rank. Here, we introduce an illustrative example with two clients to better understand the difference of the client-drift issues and the corresponding techniques between the ICLR 2024 paper and our paper.
>
> Concretely, if the clients locally fine-tune on full parameters and the server aggregates with FedAvg, the new global model parameters can be expressed as follows.
>
> \begin{equation}
>     \mathbf{W} = \mathbf{W}^0 +\frac{1}{2} (\Delta \mathbf{W}^1+\Delta \mathbf{W}^2)
> \end{equation}
>
> In the ICLR 2024 paper, when the clients locally fine-tune on low-rank parameters based on LoRA and the server uses FedAvg to aggregate the low-rank matrices, the global model parameters can be expressed as follows.
>
> \begin{equation}
> \underbrace{\tilde{\mathbf{W}} = \mathbf{W}^0 + \frac{1}{2}(\mathbf{B}^1 + \mathbf{B}^2) \times \frac{1}{2}(\mathbf{A}^1 + \mathbf{A}^2)}_\{\text{Parameter aggregation with LoRA + FedAvg}} \neq \underbrace{\mathbf{W}^0 + \frac{1}{2}(\mathbf{B}^1 \mathbf{A}^1 + \mathbf{B}^2 \mathbf{A}^2) = \mathbf{W}_0 +\frac{1}{2} (\Delta \mathbf{W}^1+\Delta \mathbf{W}^2) = \mathbf{W}}\_{\text{Ideal parameter aggregation with FedAvg on full parameter matrices}}
> \end{equation}
>
> When the clients use LoRA locally, we have $\Delta \mathbf{W}^k = \mathbf{B}^k \mathbf{A}^k$ on client $k$. The data heterogeneity is a common challenging problem in the federated learning. The collected data on different clients are Non-Independent Identically Distributed (NonIID). Thus, the parameters of local models trained on the non-IID data are quite different from each other, i.e., $\mathbf{B}^1 \neq \mathbf{B}^2$ and $\mathbf{A}^1 \neq \mathbf{A}^2$. This leads to the **client-drift issue** in the ICLR 2024 paper, i.e., the difference between $\mathbf{W}$ and $\tilde{\mathbf{W}}$ is not equal to 0. In order to address the client-drift issue, the ICLR 2024 paper keeps both $\mathbf{W}^0$ and $\mathbf{A}^0$ frozen and makes only $\mathbf{B}$ trainable.
>
> \begin{equation}
> \underbrace{\tilde{\mathbf{W}} = \mathbf{W}^0 + \frac{1}{2}(\mathbf{B}^1 + \mathbf{B}^2) \times \mathbf{A}^0}_\{\text {Parameter aggregation with LoRA + FedAvg}} = \underbrace{\mathbf{W}^0 + \frac{1}{2}(\mathbf{B}^1 \mathbf{A}^0 + \mathbf{B}^2 \mathbf{A}^0) = \mathbf{W}_0 +\frac{1}{2} (\Delta \mathbf{W}^1+\Delta \mathbf{W}^2) = \mathbf{W}}\_{\text {Ideal parameter aggregation with FedAvg on full parameter matrices}}
> \end{equation}
>
>
> In our work, the clients locally fine-tune on both low-rank parameter matrices $\mathbf{U}^k$ and $\mathbf{V}^k$ and diagonal matrix $\mathbf{\Sigma}^k$ to determine the rank. The server uses FedAvg to aggregate the low-rank matrices and diagonal matrix. The local model $\Delta \mathbf{W}^k$updates are represented as:
>
> \begin{equation}
>     \Delta \mathbf{W}^k = \mathbf{U}^k \mathbf{\Sigma}^k \mathbf{V}^k, \quad k \in \\{1, 2\\}
> \end{equation}
>
> The expected global model parameter updates $\Delta \mathbf{W}$ can be expressed as follows.
>
> \begin{equation}
>     \Delta \mathbf{W} = \frac{1}{2} (\Delta \mathbf{W}^1+\Delta \mathbf{W}^2) = \frac{1}{2} (\mathbf{U}^1 \mathbf{\Sigma}^1 \mathbf{V}^1 + \mathbf{U}^2 \mathbf{\Sigma}^2 \mathbf{V}^2)
> \end{equation}

---

> > ### Author Response · Authors · 2024-11-24
> > **Point-by-point response to the comments made by Reviewer WxWw (Continued)**
> >
> > After using FedAvg to aggregate the local matrices $\mathbf{U}^k, \mathbf{\Sigma}^k, \mathbf{V}^k$, the server produces
> >
> > $$
> > \begin{align\*}
> >     \Delta \tilde{\mathbf{W}} & = \frac{1}{2}\left(\mathbf{U}^{1}+\mathbf{U}^{2}\right) \times \frac{1}{2}\left(\mathbf{\Sigma}^{1}+\mathbf{\Sigma}^{2}\right) \times \frac{1}{2}\left(\mathbf{V}^{1}+\mathbf{V}^{2}\right) \\\\\\ &= \frac{1}{2} (\mathbf{U}^1 \mathbf{\Sigma}^1 \mathbf{V}^1 + \mathbf{U}^2 \mathbf{\Sigma}^2 \mathbf{V}^2)\underline{-\frac{3}{8} \mathbf{U}^1 \mathbf{\Sigma}^1 \mathbf{V}^1 - \frac{3}{8} \mathbf{U}^2 \mathbf{\Sigma}^2 \mathbf{V}^2 + \frac{1}{8} \mathbf{U}^1 \mathbf{\Sigma}^1 \mathbf{V}^2 + \frac{1}{8} \mathbf{U}^1 \mathbf{\Sigma}^2 \mathbf{V}^1} \\\\\\ &+ \underline{\frac{1}{8} \mathbf{U}^1 \mathbf{\Sigma}^2\mathbf{V}^2 + \frac{1}{8} \mathbf{U}^2 \mathbf{\Sigma}^1\mathbf{V}^1 + \frac{1}{8} \mathbf{U}^2 \mathbf{\Sigma}^1\mathbf{V}^2 + \frac{1}{8} \mathbf{U}^2 \mathbf{\Sigma}^2\mathbf{V}^1} \\\\\\ &= \Delta \mathbf{W} \underline{-\frac{3}{8} \mathbf{U}^1 \mathbf{\Sigma}^1 \mathbf{V}^1 - \frac{3}{8} \mathbf{U}^2 \mathbf{\Sigma}^2 \mathbf{V}^2 + \frac{1}{8} \mathbf{U}^1 \mathbf{\Sigma}^1 \mathbf{V}^2 + \frac{1}{8} \mathbf{U}^1 \mathbf{\Sigma}^2 \mathbf{V}^1} \\\\\\ &+ \underline{\frac{1}{8} \mathbf{U}^1 \mathbf{\Sigma}^2\mathbf{V}^2 + \frac{1}{8} \mathbf{U}^2 \mathbf{\Sigma}^1\mathbf{V}^1 + \frac{1}{8} \mathbf{U}^2 \mathbf{\Sigma}^1\mathbf{V}^2 + \frac{1}{8} \mathbf{U}^2 \mathbf{\Sigma}^2\mathbf{V}^1}
> > \end{align\*}
> > $$
> >
> > where the underlined terms denote the difference between $\Delta \tilde{\mathbf{W}}$ and $\Delta \mathbf{W}$.
> >
> > It is obvious that $\Delta \tilde{\mathbf{W}} \neq \Delta \mathbf{W}$ when $\mathbf{U}^{1} \neq \mathbf{U}^{2}$, $\mathbf{\Sigma}^{1} \neq \mathbf{\Sigma}^{2}$, and $\mathbf{V}^{1} \neq \mathbf{V}^{2}$ due to the data heterogeneity property of federated learning. Therefore, the **client-drift issue** is raised.
> >
> > $$
> > \underbrace{\tilde{\mathbf{W}} = \mathbf{W}^0 + \Delta \tilde{\mathbf{W}}}_{\text{Parameter aggregation with SVD-based rank-adaptive LoRA + FedAvg}} \neq \underbrace{\mathbf{W}^0 + \Delta \mathbf{W} = \mathbf{W}}\_{\text{Ideal parameter aggregation with FedAvg on full parameter matrices}}
> > $$
> >
> > The difference between $\Delta \tilde{\mathbf{W}}$ and $\Delta \mathbf{W}$ is mainly due to the noise introduced by the cross-products of LoRA modules from different clients. This difference may become more significant when (1) the number of local update steps between aggregations is large and (2) the local datasets are different across clients.
> >
> > The magic of the ICLR 2024 paper to handle the client-drift issue is to have only one parameter matrix ($\mathbf{B}$) trainable while freezing other parameter matrices. However, this approach cannot be directly utilized to conduct the client-drift issue in our work. Our work has three trainable parameter matrices $\mathbf{U}^k$, $\mathbf{V}^k$, and $\mathbf{\Sigma}^k$. If updating only the diagonal matrix $\mathbf{\Sigma}^k$ and freezing the parameter matrices $\mathbf{U}^k$ and $\mathbf{V}^k$, then the algorithm fails to update the parameter matrices, resulting in poor model performance. If updating either $\mathbf{U}^k$ or $\mathbf{V}^k$ and freezing $\mathbf{\Sigma}^k$, then the rank cannot be optimized, leading to high computational cost (by higher rank) or poor model performance (by lower rank).

---

> ### Author Response · Authors · 2024-11-24
> **Point-by-point response to the comments made by Reviewer WxWw (Continued)**
>
> Thus, we propose a Riemannian parameter matching method to match the local parameter matrices $\mathbf{U}^k$ and $\mathbf{V}^k$ on other clients with pivots $\mathbf{U}^1$ and $\mathbf{V}^1$ on client 1, in terms of their lengths and directions, i.e. $\tilde{\mathbf{U}}^k = \mathbf{U}^k \mathbf{R}^k \approx \mathbf{U}^1$ and $\tilde{\mathbf{V}}^k = \mathbf{S}^k \mathbf{V}^k \approx \mathbf{V}^1$.
> To maintain the consistency between low-rank parameter metrics before and after the Riemannian parameter matching, i.e., $ \mathbf{U}^2 \mathbf{\Sigma}^2 \mathbf{V}^2 = \tilde{\mathbf{U}}^2 \tilde{\mathbf{\Sigma}}^2 \tilde{\mathbf{V}}^2 \approx \mathbf{U}^1 \tilde{\mathbf{\Sigma}}^2 \mathbf{V}^1$, for ensuring the effectiveness of FFT-FM with rank-adaptive LoRA, we derive a modified diagonal matrix $\tilde{\mathbf{\Sigma}}^k $ for the other clients by performing the SVD on low-dimensional $r \times r$ matrices $(\mathbf{U}^1)^T \mathbf{U}^k$ and $(\mathbf{V}^1)^T \mathbf{V}^k$. Thus, the client-drift issue is resolved.
>
> \begin{equation}
> \begin{split}
> \Delta \tilde{\mathbf{W}} &=\frac{1}{2}\left(\mathbf{U}^{1}+ \tilde{\mathbf{U}^2}\right) \times \frac{1}{2}\left(\mathbf{\Sigma}^{1}+ \tilde{\mathbf{\Sigma}}^{2}\right) \times \frac{1}{2}\left(\mathbf{V}^{1}+ \tilde{\mathbf{V}^{K}}\right) \\
> &\approx\mathbf{U}^{1} \times \frac{1}{2}\left(\mathbf{\Sigma}^{1}+ \tilde{\mathbf{\Sigma}}^{2}\right) \times \mathbf{V}^{1}
> = \frac{1}{2}\left(\mathbf{U}^1 \mathbf{\Sigma}^1 \mathbf{V}^1 + \mathbf{U}^2 \mathbf{\Sigma}^2 \mathbf{V}^2\right) = \Delta \mathbf{W}
> \end{split}
> \end{equation}
>
> Based on the global diagonal matrix $\frac{1}{2}\left(\mathbf{\Sigma}^{1}+ \tilde{\mathbf{\Sigma}}^{2}\right)$, it is easy to find the optimal rank of the global parameter matrix, with aggregation on only the local low-rank matrices $\mathbf{U}^k, \tilde{\mathbf{\Sigma}}^{k}$, and $\mathbf{V}^k$.
>
> **Weakness 2:** Also, Reiemannian LoRA has been studied in a centralized setting in the following paper: Riemannian Preconditioned LoRA for Fine-Tuning Foundation Models, ICML 2024.
>
> **Answer:** The research objective of this ICML 2024 paper is completely different from ours. It aims to use different learning rates on the low-rank parameter matrices $A$ and $B$, introducing an $r \times r$ preconditioner to stabilize the LoRA learning by scaling gradients. This gradient scaling method can be derived from a new Riemannian metric. This paper does not consider the rank-drift issue and address the optimization on a Riemannian manifold and maintaining a fixed rank during updates. However, one of main contributions in this paper is to address the rank-drift issue, which is unique in the FFT-FM but not in centralized learning. Therefore, the centralized learning method in the ICML 2024 paper cannot be directly applied to our scenario.
>
> **Rank-drift issue:** The data heterogeneity is a common challenging problem in the federated learning. The collected data on different clients are Non-Independent Identically Distributed (Non-IID). Thus, the parameters of local models trained on the non-IID data are quite different from each other as well as the parameters of global model. Each local model may oscillate back and forth in different training rounds, leading to unstable and slow convergence and causing suboptimal model performance.
>
> At the same time, due to the heterogeneity of local models, the optimal ranks of local models are quite different from each other as well as the one of global model, raising the **rank-drift issue**. Similarly, the oscillation of the optimal rank of each local model can slow down the model convergence and degrade the model performance.
> For example, given two clients 1 and 2, in round 1, the local model on client 1/2 has the optimal rank 5/20. The server aggregates local models from clients to generate a global model based on FedAvg. The global model by model aggregation has the optimal rank 12. In round 2, after the global model are sent back to the clients, the clients begin the training with the global model with the optimal rank 12. Thus, the optimal rank of local models oscillate between 5/20 and 12. The rank oscillation may repeat in each training round, slowing down the model convergence.

---

> > ### Author Response · Authors · 2024-11-24
> > **Point-by-point response to the comments made by Reviewer WxWw (Continued)**
> >
> > **Weakness 3:** What are the new challenges and technical contributions compared with the combination of these works?
> >
> > **Answer:** The problem, challenges, and techniques in our paper significantly differ from the above ICLR 2024 (Sun et al., 2024) and ICML 2024 papers. Our paper addresses two unique challenges in the FFT-FM: client drift and rank drift.
> >
> > As for the **client-drift issue**, the ICLR 2024 paper conducts this issue raised by only low-rank parameter matrices $\mathbf{A}^k$ and $\mathbf{B}^k$, but our work addresses a more challenging problem by considering both low-rank parameter matrices $\mathbf{U}^k$ and $\mathbf{V}^k$ and diagonal matrix $\mathbf{\Sigma}^k$ to determine the rank. In the answer to Weakness 1, we have provided a detailed comparison and discussion about the difference of the client-drift issues and the corresponding techniques between the ICLR 2024 paper and our paper. By utilizing Riemannian Procrustes analysis, we propose a Riemannian parameter matching method to avoid the client-drift issue for ensuring the effectiveness of FFT-FM with rank-adaptive LoRA, and to reduce the cost of matrix decomposition by transforming the singular value decomposition (SVD) of high-dimensional full parameter matrices into the SVD of low-dimensional $r \times r$ matrices, where $r$ is the rank parameter in the LoRA.
> >
> > Regarding the **rank-drift issue**, both the above ICLR 2024 and ICML 2024 papers do not tackle this issue. By leveraging Riemannian manifold theory, we develop a Riemannian gradient descent (RGD) method to guarantee the local full parameter matrices on clients in the form of low-rank ones with fixed rank optimized by the server in each FFT-FM round, for alleviating the rank-drift issue to speed up the convergence of RAFFT.
> >
> > Therefore, the techniques in the above ICLR 2024 (Sun et al., 2024) and ICML 2024 papers cannot be applied to our work.

---

> > > ### Author Response · Authors · 2024-11-25
> > > **Thanks to Reviewer WxWw**
> > >
> > > We would like to thank the reviewer for taking the time to review our paper and for the valuable comments. We hope our response has adequately addressed your concerns raised in the review.
> > >
> > > Please kindly let us know if anything is unclear. We truly appreciate this opportunity to improve our work and shall be most grateful for any feedback you could give us.

---

> > > > ### Author Response · Authors · 2024-11-29
> > > > **Thanks to Reviewer WxWw**
> > > >
> > > > Dear Reviewer WxWw,
> > > >
> > > > We would like to thank you again for your valuable time and thoughtful and constructive comments. We have tried our best to clarify and address the concerns and comments in the initial response.
> > > >
> > > > We realize that you may have a busy schedule, but given the short discussion period, we decided to reach out again to see if you could review our responses. We would be happy to provide further clarifications if that helps.

---

> > > > > ### Comment · Reviewer_WxWw · 2024-11-29
> > > > >
> > > > > I appreciate the authors for making some clarifications. It would be great if all these clarifications are included in the updated manuscript.
> > > > >
> > > > > However, after taking a look at other reviewers' comments as well, I'm sorry to say that I have decided to keep my original score. I have a feeling that the current version of the paper is below the ICLR acceptance bar, and the paper can get improved a lot by making more extensive updates according to the reviewers' comments.

---

> > > > > > ### Author Response · Authors · 2024-11-29
> > > > > > **Point-by-point response to the comments made by Reviewer WxWw**
> > > > > >
> > > > > > Thank you for your feedback. We want to assure you that we have incorporated all requested clarifications into the updated manuscript:
> > > > > >
> > > > > > For **Weakness 1**, we have addressed this in detail in Lines 63-67 of the revised paper, with additional elaboration in Appendix 7.8 (Lines 2456-2552).
> > > > > >
> > > > > > For **Weakness 2**, our response is included in Lines 91-98, with further details provided in Appendix 7.8 (Lines 2554-2576).
> > > > > >
> > > > > > For **Weakness 3**, the response can be found in Lines 100-124.
> > > > > >
> > > > > > We kindly invite you to review the latest revised version of our paper, which includes these updates. We hope these changes demonstrate our commitment to addressing all reviewer concerns and significantly improving the clarity and quality of the work.

---

> > > > > > > ### Author Response · Authors · 2024-12-01
> > > > > > > **Thanks to Reviewer WxWw**
> > > > > > >
> > > > > > > We would like to thank the reviewer for taking the time to review our paper and for the valuable comments. We have carefully addressed all your concerns in the revised version of our paper. We invite you to review the revised PDF. If you have any other concerns, please do not hesitate to let us know.

---

### Official Review · Reviewer_W9cF · 2024-11-03

**Soundness:** 3
**Presentation:** 2
**Contribution:** 2
**Rating:** 3
**Confidence:** 3

**Summary:**

This paper proposed Riemannian aggregation and update based on SVD parameterization of LoRA to address the client drift and rank drift issues of LoRA fine-tuning in federated learning.

**Strengths:**

I appreciate the extensive experiments on comparing the proposed methods to 13 baselines, and 30+ tables. The baselines include prompt tuning methods and LoRA methods. The Remannian aggregation and updates have certain theorems to show that they can work as intended.

**Weaknesses:**

However, I also find it a bit hard to interpret the results on why the proposed RAFFT methods are better, and the difference of the metrics (accuracy, loss and time). I hope the following comments can help improve the draft, and I am happy to discuss during the rebuttal period.

**Questions:**

I kindly ask the authors to clarify “the client-drift issue” for LoRA. Feel free to use the definition in Sun et al. 2024a and Wang et al. 2024, but I would make my decision based on a self-contained explanation. Eq (1) is a fact, but it is not clear to me if it is an issue. In the [open review](https://openreview.net/forum?id=NLPzL6HWNl) of Sun et al. ICLR’24, reviewers also raised concerns about “The explanations and the analysis of LoRA are not convincing. The discussion … heavily focus on the nonlinear nature of LoRA, but deep neural networks suffer from more severe nonlinearity.”. In [Cho et al. 2024](https://arxiv.org/abs/2401.06432), the authors observed that reconstruction \Delta W before aggregation achieves worse performance in their experiments.

Similarly, could the authors provide a self-contained definition of “rank-drift issue”?

The presentation of the paper can be improved. Specifically
- In section 3, the content seems to be a mix of derivation and algorithms. It is unclear what exactly needs to be done on both clients and the server.
- It is unclear to me what the advantage of Riemann aggregation in section 3 is compared to aggregating the reconstructed \Delta W from every client.
-  It is unclear how LoRA (SVD) parameters are determined. It is unclear whether adaptive ranks (what kind of adaptive?), or fixed rank are used.
- How are the projected gradients in section 4 computed? Could the authors clarify the advantages compared to Zhang et al. 2023, conventional LoRA, and full model fine-tuning? This may be related to the definition of rank-drift issue, but I am not sure.

Minor issue:
Some references are duplicated, e.g.,
Qingru Zhang, Minshuo Chen, Alexander Bukharin, Pengcheng He, Yu Cheng, Weizhu Chen, and Tuo Zhao. Adaptive budget allocation for parameter-efficient fine-tuning. ICLR’23
Youbang Sun, Zitao Li, Yaliang Li, and Bolin Ding. Improving lora in privacy-preserving federated
Learning. ICLR’24

---

> ### Author Response · Authors · 2024-11-24
> **Point-by-point response to the comments made by Reviewer W9cF**
>
> We thank this reviewer for the constructive comments.
>
> **Weakness 1:**  I also find it a bit hard to interpret the results on why the proposed RAFFT methods are better, and the difference of the metrics (accuracy, loss and time)
>
> **Answer:** Our paper addresses two unique challenges in the FFT-FM: client drift and rank drift.
>
> First, by leverages Riemannian parameter matching to dynamically adjust the rank based on the singular value matrix $\mathbf{\Sigma}$ after aggregating global parameters on the server side. This approach eliminates the need for additional operations such as full matrix SVD decomposition. Furthermore, it resolves the client-drift issue, which arises from independently aggregating the client parameter matrices $\mathbf{U}^k$, $\mathbf{\Sigma}^k$, and $\mathbf{V}^k$.
>
> Second, by leveraging Riemannian manifold theory, we develop a Riemannian gradient descent (RGD) method to guarantee the local full parameter matrices on clients in the form of low-rank ones with fixed rank optimized by the server in each FFT-FM round, for alleviating the rank-drift issue to speed up the convergence of RAFFT.
>
> In line with established protocols from prior studies on federated learning for large language models [1, 2, 3], we evaluate the efficacy of parameter-efficient fine-tuning for classification tasks using three key metrics: Loss, Accuracy, and Time. Loss quantifies the model's prediction error, providing insights into the effectiveness of the learning process. Accuracy assesses the model's performance in classifying unseen data, reflecting the practical applicability of the federated learning model. Time measures the speed of convergence and the overall computational demand, both of which are critical in federated settings where computational resources and time are often constrained.
>
> In our submission, we have conducted a comprehensive analysis of our proposed method, RAFFT, across six datasets and against 13 baseline methods. The experimental results, detailed in Tables 1-2 and 4-13 of the submission, demonstrate the significant advantages of RAFFT. Specifically, RAFFT achieves notable accuracy improvements, with a maximum increase of 30.22% over other baseline methods. Moreover, RAFFT excels in extreme non-IID data environments, achieving an accuracy of up to 58.05%, highlighting its robustness under challenging data distributions. In terms of efficiency, RAFFT reduces training time by up to 41.86%, underscoring its practicality and effectiveness in federated learning scenarios where both accuracy and computational efficiency are critical.
>
> **REFERENCES:**
>
> [1] Sara Babakniya, Ahmed Elkordy, Yahya Ezzeldin, Qingfeng Liu, Kee-Bong Song, MOSTAFA EL-
> Khamy, and Salman Avestimehr. SLoRA: Federated parameter efficient fine-tuning of language
> models. In International Workshop on Federated Learning in the Age of Foundation Models
> in Conjunction with NeurIPS 2023, 2023.
>
> [2] Yae Jee Cho, Luyang Liu, Zheng Xu, Aldi Fahrezi, Matt Barnes, and Gauri Joshi. Heterogeneous
> loRA for federated fine-tuning of on-device foundation models. In International Workshop on
> Federated Learning in the Age of Foundation Models in Conjunction with NeurIPS 2023, 2023.
>
> [3] Youbang Sun, Zitao Li, Yaliang Li, and Bolin Ding. Improving loRA in privacy-preserving federated
> learning. In The Twelfth International Conference on Learning Representations, 2024b.
>
> **Question 1:** I kindly ask the authors to clarify "the client-drift issue" for LoRA. Feel free to use the definition in Sun et al. 2024a and Wang et al. 2024, but I would make my decision based on a self-contained explanation. Eq (1) is a fact, but it is not clear to me if it is an issue.
>
> **Answer**: **Client-drift issue:** The client-drift issue is due to the aggregation of low-rank matrices in the FFT-FM. Here, we introduce an illustrative example with two clients to better understand the client-drift issue.
>
> Concretely, if the clients locally fine-tune on full parameters and the server aggregates with FedAvg, the new global model parameters can be expressed as follows.
>
> \begin{equation}
>     \mathbf{W} = \mathbf{W}^0 +\frac{1}{2} (\Delta \mathbf{W}^1+\Delta \mathbf{W}^2)
> \end{equation}
>
> In our work, the clients locally fine-tune on both low-rank parameter matrices $\mathbf{U}^k$ and $\mathbf{V}^k$ and diagonal matrix $\mathbf{\Sigma}^k$ to determine the rank. The server uses FedAvg to aggregate the low-rank matrices and diagonal matrix. The local model updates $\Delta \mathbf{W}^k$ are represented as:
>
> \begin{equation}
>     \Delta \mathbf{W}^k = \mathbf{U}^k \mathbf{\Sigma}^k \mathbf{V}^k,  \quad k ∈ \\{1, 2\\}
> \end{equation}
>
> The expected global model parameter updates $\Delta \mathbf{W}$ can be expressed as follows.
>
> \begin{equation}
>     \Delta \mathbf{W} = \frac{1}{2} (\Delta \mathbf{W}^1+\Delta \mathbf{W}^2) = \frac{1}{2} (\mathbf{U}^1 \mathbf{\Sigma}^1 \mathbf{V}^1 + \mathbf{U}^2 \mathbf{\Sigma}^2 \mathbf{V}^2)
> \end{equation}

---

> ### Author Response · Authors · 2024-11-24
> **Point-by-point response to the comments made by Reviewer W9cF (Continued)**
>
> After using FedAvg to aggregate the local matrices $\mathbf{U}^k, \mathbf{\Sigma}^k, \mathbf{V}^k$, the server produces
>
> $$
> \begin{align\*}
>     \Delta \tilde{\mathbf{W}} & = \frac{1}{2}\left(\mathbf{U}^{1}+\mathbf{U}^{2}\right) \times \frac{1}{2}\left(\mathbf{\Sigma}^{1}+\mathbf{\Sigma}^{2}\right) \times \frac{1}{2}\left(\mathbf{V}^{1}+\mathbf{V}^{2}\right) \\\\\\ &= \frac{1}{2} (\mathbf{U}^1 \mathbf{\Sigma}^1 \mathbf{V}^1 + \mathbf{U}^2 \mathbf{\Sigma}^2 \mathbf{V}^2)\underline{-\frac{3}{8} \mathbf{U}^1 \mathbf{\Sigma}^1 \mathbf{V}^1 - \frac{3}{8} \mathbf{U}^2 \mathbf{\Sigma}^2 \mathbf{V}^2 + \frac{1}{8} \mathbf{U}^1 \mathbf{\Sigma}^1 \mathbf{V}^2 + \frac{1}{8} \mathbf{U}^1 \mathbf{\Sigma}^2 \mathbf{V}^1} \\\\\\ &+ \underline{\frac{1}{8} \mathbf{U}^1 \mathbf{\Sigma}^2\mathbf{V}^2 + \frac{1}{8} \mathbf{U}^2 \mathbf{\Sigma}^1\mathbf{V}^1 + \frac{1}{8} \mathbf{U}^2 \mathbf{\Sigma}^1\mathbf{V}^2 + \frac{1}{8} \mathbf{U}^2 \mathbf{\Sigma}^2\mathbf{V}^1} \\\\\\ &= \Delta \mathbf{W} \underline{-\frac{3}{8} \mathbf{U}^1 \mathbf{\Sigma}^1 \mathbf{V}^1 - \frac{3}{8} \mathbf{U}^2 \mathbf{\Sigma}^2 \mathbf{V}^2 + \frac{1}{8} \mathbf{U}^1 \mathbf{\Sigma}^1 \mathbf{V}^2 + \frac{1}{8} \mathbf{U}^1 \mathbf{\Sigma}^2 \mathbf{V}^1} \\\\\\ &+ \underline{\frac{1}{8} \mathbf{U}^1 \mathbf{\Sigma}^2\mathbf{V}^2 + \frac{1}{8} \mathbf{U}^2 \mathbf{\Sigma}^1\mathbf{V}^1 + \frac{1}{8} \mathbf{U}^2 \mathbf{\Sigma}^1\mathbf{V}^2 + \frac{1}{8} \mathbf{U}^2 \mathbf{\Sigma}^2\mathbf{V}^1}
> \end{align\*}
> $$
>
> where the underlined terms denote the difference between $\Delta \tilde{\mathbf{W}}$ and $\Delta \mathbf{W}$.
>
> It is obvious that $\Delta \tilde{\mathbf{W}} \neq \Delta \mathbf{W}$ when $\mathbf{U}^{1} \neq \mathbf{U}^{2}$, $\mathbf{\Sigma}^{1} \neq \mathbf{\Sigma}^{2}$, and $\mathbf{V}^{1} \neq \mathbf{V}^{2}$ due to the data heterogeneity property of federated learning. Therefore, the **client-drift issue** is raised.
>
> $$
> \underbrace{\tilde{\mathbf{W}} = \mathbf{W}^0 + \Delta \tilde{\mathbf{W}}}_{\text{Parameter aggregation with SVD-based rank-adaptive LoRA + FedAvg}} \neq \underbrace{\mathbf{W}^0 + \Delta \mathbf{W} = \mathbf{W}}\_{\text{Ideal parameter aggregation with FedAvg on full parameter matrices}}
> $$
>
> The difference between $\Delta \tilde{\mathbf{W}}$ and $\Delta \mathbf{W}$ is mainly due to the noise introduced by the cross-products of LoRA modules from different clients. This difference may become more significant when (1) the number of local update steps between aggregations is large and (2) the local datasets are different across clients.
>
> We propose a Riemannian parameter matching method to match the local parameter matrices $\mathbf{U}^k$ and $\mathbf{V}^k$ on other clients with pivots $\mathbf{U}^1$ and $\mathbf{V}^1$ on client 1, in terms of their lengths and directions, i.e. $\tilde{\mathbf{U}}^k = \mathbf{U}^k \mathbf{R}^k \approx \mathbf{U}^1$ and $\tilde{\mathbf{V}}^k = \mathbf{S}^k \mathbf{V}^k \approx \mathbf{V}^1$.
> To maintain the consistency between low-rank parameter metrics before and after the Riemannian parameter matching, i.e., $ \mathbf{U}^2 \mathbf{\Sigma}^2 \mathbf{V}^2 = \tilde{\mathbf{U}}^2 \tilde{\mathbf{\Sigma}}^2 \tilde{\mathbf{V}}^2 \approx \mathbf{U}^1 \tilde{\mathbf{\Sigma}}^2 \mathbf{V}^1$, for ensuring the effectiveness of FFT-FM with rank-adaptive LoRA, we derive a modified diagonal matrix $\tilde{\mathbf{\Sigma}}^k $ for the other clients by performing the SVD on low-dimensional $r \times r$ matrices $(\mathbf{U}^1)^T \mathbf{U}^k$ and $(\mathbf{V}^1)^T \mathbf{V}^k$. Thus, the client-drift issue is resolved.
> \begin{equation}
> \begin{split}
> \Delta \tilde{\mathbf{W}} &=\frac{1}{2}\left(\mathbf{U}^{1}+ \tilde{\mathbf{U}^2}\right) \times \frac{1}{2}\left(\mathbf{\Sigma}^{1}+ \tilde{\mathbf{\Sigma}}^{2}\right) \times \frac{1}{2}\left(\mathbf{V}^{1}+ \tilde{\mathbf{V}^{K}}\right) \\
> &\approx\mathbf{U}^{1} \times \frac{1}{2}\left(\mathbf{\Sigma}^{1}+ \tilde{\mathbf{\Sigma}}^{2}\right) \times \mathbf{V}^{1}
> = \frac{1}{2}\left(\mathbf{U}^1 \mathbf{\Sigma}^1 \mathbf{V}^1 + \mathbf{U}^2 \mathbf{\Sigma}^2 \mathbf{V}^2\right) = \Delta \mathbf{W}
> \end{split}
> \end{equation}
> Based on the global diagonal matrix $\frac{1}{2}\left(\mathbf{\Sigma}^{1}+ \tilde{\mathbf{\Sigma}}^{2}\right)$, it is easy to find the optimal rank of the global parameter matrix, with aggregation on only the local low-rank matrices $\mathbf{U}^k, \tilde{\mathbf{\Sigma}}^{k}$, and $\mathbf{V}^k$.

---

> > ### Author Response · Authors · 2024-11-24
> > **Point-by-point response to the comments made by Reviewer W9cF (Continued)**
> >
> > **Question 2: In the open review of Sun et al. ICLR'24, reviewers also raised concerns about "The explanations and the analysis of LoRA are not convincing. The discussion heavily focus on the nonlinear nature of LoRA, but deep neural networks suffer from more severe nonlinearity.**
> >
> > **Answer:** The problem, challenges, and techniques in our paper significantly differ from the above ICLR 2024 (Sun et al., 2024), although both papers have the client-drift issue and the reason of the issue in both papers are the same, due to the aggregation of low-rank matrices in the FFT-FM.
> >
> > The ICLR 2024 paper conducts this issue raised by only low-rank parameter matrices $\mathbf{A}^k$ and $\mathbf{B}^k$, but our work addresses a more challenging problem by considering both low-rank parameter matrices $\mathbf{U}^k$ and $\mathbf{V}^k$ and diagonal matrix $\mathbf{\Sigma}^k$ to determine the rank. Here, we introduce an illustrative example with two clients to better understand the difference of the client-drift issues and the corresponding techniques between the ICLR 2024 paper and our paper.
> >
> > Concretely, if the clients locally fine-tune on full parameters and the server aggregates with FedAvg, the new global model parameters can be expressed as follows.
> >
> > \begin{equation}
> >     \mathbf{W} = \mathbf{W}^0 +\frac{1}{2} (\Delta \mathbf{W}^1+\Delta \mathbf{W}^2)
> > \end{equation}
> >
> > In the ICLR 2024 paper, when the clients locally fine-tune on low-rank parameters based on LoRA and the server uses FedAvg to aggregate the low-rank matrices, the global model parameters can be expressed as follows.
> >
> > \begin{equation}
> > \underbrace{\tilde{\mathbf{W}} = \mathbf{W}^0 + \frac{1}{2}(\mathbf{B}^1 + \mathbf{B}^2) \times \frac{1}{2}(\mathbf{A}^1 + \mathbf{A}^2)}_\{\text{Parameter aggregation with LoRA + FedAvg}} \neq \underbrace{\mathbf{W}^0 + \frac{1}{2}(\mathbf{B}^1 \mathbf{A}^1 + \mathbf{B}^2 \mathbf{A}^2) = \mathbf{W}_0 +\frac{1}{2} (\Delta \mathbf{W}^1+\Delta \mathbf{W}^2) = \mathbf{W}}\_{\text{Ideal parameter aggregation with FedAvg on full parameter matrices}}
> > \end{equation}
> >
> > When the clients use LoRA locally, we have $\Delta \mathbf{W}^k = \mathbf{B}^k \mathbf{A}^k$ on client $k$. The data heterogeneity is a common challenging problem in the federated learning. The collected data on different clients are Non-Independent Identically Distributed (NonIID). Thus, the parameters of local models trained on the non-IID data are quite different from each other, i.e., $\mathbf{B}^1 \neq \mathbf{B}^2$ and $\mathbf{A}^1 \neq \mathbf{A}^2$. This leads to the **client-drift issue** in the ICLR 2024 paper, i.e., the difference between $\mathbf{W}$ and $\tilde{\mathbf{W}}$ is not equal to 0. In order to address the client-drift issue, the ICLR 2024 paper keeps both $\mathbf{W}^0$ and $\mathbf{A}^0$ frozen and makes only $\mathbf{B}$ trainable.
> >
> > \begin{equation}
> > \underbrace{\tilde{\mathbf{W}} = \mathbf{W}^0 + \frac{1}{2}(\mathbf{B}^1 + \mathbf{B}^2) \times \mathbf{A}^0}_\{\text {Parameter aggregation with LoRA + FedAvg}} = \underbrace{\mathbf{W}^0 + \frac{1}{2}(\mathbf{B}^1 \mathbf{A}^0 + \mathbf{B}^2 \mathbf{A}^0) = \mathbf{W}_0 +\frac{1}{2} (\Delta \mathbf{W}^1+\Delta \mathbf{W}^2) = \mathbf{W}}\_{\text {Ideal parameter aggregation with FedAvg on full parameter matrices}}
> > \end{equation}
> >
> > The magic of the ICLR 2024 paper to handle the client-drift issue is to have only one parameter matrix ($\mathbf{B}$) trainable while freezing other parameter matrices. However, this approach cannot be directly utilized to conduct the client-drift issue in our work. Our work has three trainable parameter matrices $\mathbf{U}^k$, $\mathbf{V}^k$, and $\mathbf{\Sigma}^k$. If updating only the diagonal matrix $\mathbf{\Sigma}^k$ and freezing the parameter matrices $\mathbf{U}^k$ and $\mathbf{V}^k$, then the algorithm fails to update the parameter matrices, resulting in poor model performance. If updating either $\mathbf{U}^k$ or $\mathbf{V}^k$ and freezing $\mathbf{\Sigma}^k$, then the rank cannot be optimized, leading to high computational cost (by higher rank) or poor model performance (by lower rank).

---

> > > ### Author Response · Authors · 2024-11-25
> > > **Point-by-point response to the comments made by Reviewer W9cF (Continued)**
> > >
> > > **Question 3:** In Cho et al. 2024, the authors observed that reconstruction \Delta W before aggregation achieves worse performance in their experiments.
> > >
> > > **Answer:** In our method, we do not reconstruct the $\Delta \mathbf{W}$ matrix before aggregation. Instead, we directly aggregate the three independent parameter matrices $\mathbf{U}, \mathbf{\Sigma}$, and $\mathbf{V}$ using our Riemannian parameter matching method. The research focus of Cho et al. (2024) fundamentally differs from ours, as their study addresses the issue of rank heterogeneity. Rank heterogeneity occurs when clients' parameter matrices have differing ranks, resulting in inconsistent dimensions that hinder direct aggregation. To address this, the approach described in Cho et al. (2024) involves reconstructing the $\Delta \mathbf{W}$ matrix to ensure dimensional consistency, followed by performing truncated SVD after aggregation. Truncated SVD assigns specific rank values to each client, which results in the loss of critical information and features. In the answer to Question 1, we have provided a detailed discussion about our method. Therefore, Our method avoids the performance degradation associated with reconstructing the $\Delta \mathbf{W}$ matrix.
> > >
> > > **Question 4:** Similarly, could the authors provide a self-contained definition of “rank-drift issue”?
> > >
> > > **Answer**: **Rank-drift issue:** The data heterogeneity is a common challenging problem in the federated learning. The collected data on different clients are Non-Independent Identically Distributed (Non-IID). Thus, the parameters of local models trained on the non-IID data are quite different from each other as well as the parameters of global model. Each local model may oscillate back and forth in different training rounds, leading to unstable and slow convergence and causing suboptimal model performance.
> > >
> > > At the same time, due to the heterogeneity of local models, the optimal ranks of local models are quite different from each other as well as the one of global model, raising the **rank-drift issue**. Similarly, the oscillation of the optimal rank of each local model can degrade the model performance.
> > > For example, given two clients 1 and 2, in round 1, the local model on client 1/2 has the optimal rank 5/20. The server aggregates local models from clients to generate a global model based on FedAvg. The global model by model aggregation has the optimal rank 12. In round 2, after the global model are sent back to the clients, the clients begin the training with the global model with the optimal rank 12. Thus, the optimal rank of local models oscillate between 5/20 and 12. The rank oscillation may repeat in each training round, slowing down the model convergence.
> > >
> > > The following table 1 presents a case study regarding the rank-drift in federated learning. In round 1, the optimal rank of the global model is 16. In round 2, the optimal rank is changed to 11. In subsequent rounds, the optimal rank continue to oscillate back and forth. The ranks of local models fluctuate around values such as 9.75, 10.17, and 10.53, while the rank of global modelgradually decreases and stabilizes at 8. These results confirm that rank-drift introduces instability into the training process, slowing convergence and degrading performance, which emphasizes the importance of our Riemannian gradient descent optimization approach to stabilize ranks and ensure efficient training.
> > >
> > > **Table 1: Client and Server Ranks Over 25 Rounds on LLaMA 7B+MPQA**
> > > | Round (1-25) | Client Rank   | Server Rank   |
> > > |--------------|---------------|---------------|
> > > | 1            | 16            | 16            |
> > > | 2            | 9.20625       | 11            |
> > > | 3            | 9.153125      | 10            |
> > > | 4            | 9.75          | 10            |
> > > | 5            | 10.165625     | 9             |
> > > | 6            | 10.0046875    | 9             |
> > > | 7            | 9.88125       | 9             |
> > > | 8            | 9.9515625     | 9             |
> > > | 9            | 10.16875      | 9             |
> > > | 10           | 10.3921875    | 8             |
> > > | 11           | 10.4671875    | 8             |
> > > | 12           | 10.5265625    | 8             |
> > > | 13           | 10.50625      | 8             |
> > > | 14           | 10.51875      | 8             |
> > > | 15           | 10.5078125    | 8             |
> > > | 16           | 10.4546875    | 8             |
> > > | 17           | 10.39375      | 8             |
> > > | 18           | 10.2953125    | 8             |
> > > | 19           | 10.2125       | 8             |
> > > | 20           | 10.175        | 8             |
> > > | 21           | 10.1109375    | 8             |
> > > | 22           | 10.0296875    | 8             |
> > > | 23           | 9.98125       | 8             |
> > > | 24           | 9.9328125     | 8             |
> > > | 25           | 9.903125      | 8             |
> > > | 26           | 9.8640625     | 8             |

---

> ### Author Response · Authors · 2024-11-25
> **Point-by-point response to the comments made by Reviewer W9cF (Continued)**
>
> **Question 5:** In section 3, the content seems to be a mix of derivation and algorithms. It is unclear what exactly needs to be done on both clients and the server.
>
> **Answer:** In our paper's appendix (Section 7.9, Algorithm 1), we have provided a detailed description of the complete RAFFT algorithm. In each communication round, clients perform local updates using the Riemannian gradient and the retraction function, ensuring that the optimal rank selected by the server remains unchanged (lines 11-14). After the updates, each client sends the updated parameters back to the server (line 17). The server then performs Riemannian parameter matching between the $\mathbf{U}$ and $\mathbf{V}$ matrices of all clients and those of the first client to achieve alignment in both length and direction (line 24-28). The server adjusts the corresponding singular value matrix to ensure consistency before and after matching (line 29). The aggregated $\mathbf{U}$, $\mathbf{V}$, and $\mathbf{\Sigma}$ matrices from all clients are then combined (line 30-33). The rank for the next round of training is determined based on the values in the singular value matrix, and the model parameter matrix is updated accordingly (line 34-37). The server distributes the updated $\mathbf{U}$, $\mathbf{V}$, and $\mathbf{\Sigma}$ matrices back to the clients, and training continues until convergence.
>
> **Question 6:** It is unclear to me what the advantage of Riemann aggregation in section 3 is compared to aggregating the reconstructed \Delta W from every client.
>
> **Answer:** Thanks for your helpful comments. Reconstructing $\Delta \mathbf{W}$ requires every client to perform full-matrix SVD decomposition to select the optimal rank, which is computationally expensive. In contrast, our Riemann aggregation method eliminates this overhead by allowing the server to directly determine the rank during aggregation. Instead of reconstructing $\Delta \mathbf{W}$, our approach operates directly in the decomposed matrix space ($\mathbf{U}$, $\mathbf{\Sigma}$, and $\mathbf{V}$), where the singular values in $\mathbf{\Sigma}$ are leveraged to dynamically adapt the rank.
>
> **Question 7:** It is unclear how LoRA (SVD) parameters are determined. It is unclear whether adaptive ranks (what kind of adaptive?), or fixed rank are used.
>
> **Answer:** We initialize the parameters following the previous work, AdaLoRA. Specifically, we randomly initialize the matrices $\mathbf{U}$ and $\mathbf{V}$, and set $\mathbf{\Sigma}$ to 0, ensuring that $\Delta \mathbf{W}=0$ at the beginning of training. During subsequent parameter updates, all three matrices are updated.
>
> Our overall goal is to achieve adaptive rank LoRA fine-tuning in a federated learning scenario by combining dynamic rank selection on the server side with fixed-rank optimization on the client side. This two-part strategy ensures that the federated system remains stable and effectively adapts to the heterogeneity of client data distributions, improving both performance and convergence.
>
> Specifically:
>
> On the server side, adaptive rank selection dynamically determines the optimal rank after each training round. Our approach uses the singular value contribution rate, $\Theta(r)$, to select the rank $\tilde{\mathbf{\Sigma}}^{k'}$. The contribution rate $\Theta(r)$ is defined as the cumulative percentage of the $r$ largest singular values relative to the total sum of all singular values. The server retains the $r$ singular values such that $\Theta(r)$ exceeds a predefined threshold $\varphi$. For instance, if there are 16 singular values in total and the cumulative sum of the 5 largest singular values accounts for more than 90% of the total, the server adjusts the rank to 5. This method ensures that the server retains the most important information while reducing redundancy in the parameter matrices.
>
> On the client side, by leveraging Riemannian manifold theory, we develop a Riemannian gradient descent (RGD) method to guarantee the local full parameter matrices on clients in the form of low-rank ones with fixed rank optimized by the server in each FFT-FM round.

---

> ### Author Response · Authors · 2024-11-25
> **Point-by-point response to the comments made by Reviewer W9cF (Continued)**
>
> **Question 8:** How are the projected gradients in section 4 computed? Could the authors clarify the advantages compared to Zhang et al. 2023, conventional LoRA, and full model fine-tuning? This may be related to the definition of rank-drift issue, but I am not sure.
>
> **Answer:** Before explaining projected gradients, we first compare RAFFT with previous works. Methods such as AdaLoRA (Zhang et al., 2023), conventional LoRA, and full model fine-tuning are designed for single-machine settings. When extended to federated learning scenarios, these methods encounter the rank-drift issue due to data heterogeneity, as described in detail in the response to Question 4. However, these methods do not guarantee that the selected optimal rank remains consistent during updates, negatively impacting convergence speed and model performance. In contrast, our approach utilizes a Riemannian manifold encompassing all fixed-rank matrices. By optimizing directly on this manifold, we ensure that the rank remains invariant throughout client updates. This is because the Riemannian manifold inherently constrains the optimization process to matrices of the same fixed rank. In contrast, traditional gradient descent methods like SGD have no such constraints and may cause updates to deviate from the manifold, resulting in unintended changes in rank.
>
> In contrast to traditional optimization techniques, our approach ensures rank invariance during updates by leveraging the geometry of the Riemannian manifold. To achieve this, we project the Euclidean gradient onto the tangent space $T_{M}$ at the point $\Delta \mathbf{W}$ on the manifold $M$, thereby obtaining the Riemannian gradient. The projected gradient lies within the tangent space, ensuring that the update direction respects the manifold's constraints.
>
> After the gradient step, the updated point is retracted back onto the Riemannian manifold using a retraction operation, which guarantees that the updated model parameters remain within the space of fixed-rank matrices. By keeping updates confined to the Riemannian manifold, our approach preserves the rank consistency throughout local updates, addressing the limitations of conventional methods that may inadvertently alter the rank during optimization.
>
> Therefore, our method alleviates the rank-drift issue to speed up the convergence of RAFFT.
>
> For specific formulas and proofs, please refer to the appendix. The projection function is provided in Appendix 7.3 (Eq. 61). The Riemannian gradient is defined in Section 4 (Eq. 21), and the retraction function is described in Appendix 7.3 (Eq. 68). Furthermore, the process is detailed in Algorithm 1 (lines 11 to 14), Appendix 7.9.
>
> **Question 9:** Minor issue: Some references are duplicated, e.g., Qingru Zhang, Minshuo Chen, Alexander Bukharin, Pengcheng He, Yu Cheng, Weizhu Chen, and Tuo Zhao. Adaptive budget allocation for parameter-efficient fine-tuning. ICLR’23 Youbang Sun, Zitao Li, Yaliang Li, and Bolin Ding. Improving lora in privacy-preserving federated Learning. ICLR’24
>
> **Answer:** Thanks for your helpful comments. We have carefully proofread our manuscript and corrected the duplication of the mentioned references in the rebuttal revision to enhance the paper's readability.

---

> > ### Author Response · Authors · 2024-11-25
> > **Thanks to Reviewer W9cF**
> >
> > We would like to thank the reviewer for taking the time to review our paper and for the valuable comments. We
> > hope our response has adequately addressed your concerns raised in the review.
> >
> > Please kindly let us know if anything is unclear. We truly appreciate this opportunity to improve our work and
> > shall be most grateful for any feedback you could give us.

---

> > > ### Comment · Reviewer_W9cF · 2024-11-26
> > >
> > > I thank the authors for their response. However, I do not think my concerns are addressed.
> > >
> > > 1) while I appreciate the experiments and observation that ranks are different on server and clients, I believe both client-drift and rank-drift need clearer definition. The current description are "facts" rather than "issues", and I am not fully convinced.
> > >
> > > 2) Even if we acknowledge these "issues" and try to resolve them, I am not fully convinced the advantage of the proposed method. For example, why the proposed method is better than the following simple strategy: run LoRA on each client, reconstruct \Delta W and send back to server, server aggregate \Delta W from clients, and run SVD before distributing LoRA module with different rank to each client. The \Delta W reconstruction can happen on the server, though that potentially raises more privacy concerns.

---

> ### Author Response · Authors · 2024-11-28
> **Point-by-point response to the comments made by Reviewer W9cF**
>
> **Question 10**: while I appreciate the experiments and observation that ranks are different on server and clients, I believe both client-drift and rank-drift need clearer definition. The current description are "facts" rather than "issues", and I am not fully convinced.
>
> **Answer**: Thank you for your follow-up feedback. The **client-drift issue** is defined as the discrepancy between independently aggregating the local low-rank parameter matrices on clients and aggregating the full parameter matrices on clients. Specifically, when the server aggregates low-rank parameter matrices $\frac{1}{k}\Sigma_k\mathbf{U}^k \times \frac{1}{k}\Sigma_k\mathbf{\Sigma}^k \times \frac{1}{k}\Sigma_k\mathbf{V}^k$ independently, the resulting updates deviate from the exact centralized LoRA $\mathbf{U} \times \mathbf{\Sigma} \times \mathbf{V}$ as well as the aggregation $\frac{1}{k}\Sigma_k\mathbf{U}^k \times \mathbf{\Sigma}^k \times \mathbf{V}^k$ of the full parameter matrices on clients. We have provide the detailed analysis and discussion the discrepancy between two aggregation methods in the setting of federated learning in the answer to Question 1. The main reason is due to the noise introduced by the cross-products of the local low-rank parameter matrices from different clients. This discrepancy may become more significant when (1) the number of local update steps between aggregations is large and (2) the local datasets are different across clients.
>
> The client-drift issue has been clearly defined in previous studies including [Sun et al. (2024a) [1]; Wang et al. (2024) [2]; Singha et al. (2024) [3]; Guo et al. (2024) [4]; Koo et al. (2024) [5]; Nguyen et al. (2024)] [6], where they explicitly defined the client-drift problem caused by the aggregation of the local low-rank parameter matrices in federated fine-tuning of foundation models (FFT-FM), similar to the definition in our paper. All these previous works suggest that the client drift issue is not just a fact but a critical issue that needs to be addressed. Our definition of the client-drift problem is consistent with these works, as the reason for the issue in all cases stems from the aggregation of low-rank matrices in FFT-FM.
>
> In addition, our work addresses a more complex problem. While the aforementioned papers limit their focus to low-rank parameter matrices $\mathbf{A}_k$ and $\mathbf{B}_k$, our method simultaneously considers the aggregation of three components: low-rank parameter matrices $\mathbf{U}^k$ and $\mathbf{V}^k$, as well as the diagonal matrix $\mathbf{\Sigma}^k$, which determines the rank. By addressing this more challenging problem, our approach goes beyond the scenarios discussed in prior works, providing a more comprehensive solution to the client-drift issue in FFT-FM.
>
> **Rank-drift issue** refers to the discrepancy between the global optimal rank on the server and the locally optimal ranks on individual clients, caused by data heterogeneity, leading to oscillations in the optimal ranks during training and slowing down the convergence.
>
> Data heterogeneity is a common and challenging problem in federated learning systems. Let $r^\star$ denote the global optimal rank on the server and $r^k$ represent the optimal rank for each client. Due to the data heterogeneity, these local optimal ranks $r^k$ may differ significantly from each other and from the global optimal rank $r^\star$. Each client, influenced by its non-IID data, pushes the model toward its own optimal rank in the optimization space, moving away from the global optimal rank. Even if all clients start with the same rank, they will eventually converge toward their individual optimal ranks $r^k$, causing $r^k$ to deviate from $r^\star$.
>
> This discrepancy results in \textbf{rank-drift}, where the optimal rank for each client oscillates back and forth across training rounds. Consequently, this rank deviation also causes the local parameter matrices to deviate from the optimal global parameter matrices, leading to unstable and slow convergence, ultimately resulting in suboptimal model performance. The issue is particularly pronounced in scenarios where client models are trained on highly heterogeneous data, as the varying ranks exacerbate the divergence from the global optimum.
>
> For example, given two clients 1 and 2, in round 1, the local model on client 1/2 has the optimal rank 5/20. The server aggregates local models from clients to generate a global model based on FedAvg. The global model by model aggregation has the optimal rank 12. In round 2, after the global model are sent back to the clients, the clients begin the training with the global model with the optimal rank 12. Thus, the optimal rank of local models oscillate between 5/20 and 12. The rank oscillation may repeat in each training round, slowing down the model convergence.

---

> ### Author Response · Authors · 2024-11-28
> **Point-by-point response to the comments made by Reviewer W9cF (Continued)**
>
> **Question 11:** Even if we acknowledge these issues and try to resolve them, I am not fully convinced the advantage of the proposed method. For example, why the proposed method is better than the following simple strategy: run LoRA on each client, reconstruct $\Delta W$ and send back to server, server aggregate $\Delta W$ from clients, and run SVD before distributing LoRA module with different rank to each client. The $\Delta W$ reconstruction can happen on the server, though that potentially raises more privacy concerns.
>
> **Answer**: As mentioned in early discussion, Cho et al. (2024) [7] observed that reconstruction $\Delta W$ before aggregation achieves worse performance in their experiments. Another Mahla et al. [8] work also reported that reconstructing $\Delta \mathbf{W}$ before aggregation leads to performance degradation.
>
> This kind of method based on the reconstruction of $\Delta \mathbf{W}$ before aggregation will result in the following three drawbacks, in terms of effectiveness and efficiency:
>
> **1. Performance Degradation due to Matrix Reconstruction:**
>
> Reconstructing matrices before aggregation leads to performance degradation, as it causes the loss of critical information and features during the aggregation process. This results in incomplete updates and negatively impacts model performance, as observed in Cho et al. (2024) and Mahla et al. , where the reconstruction process contributed to a significant drop in accuracy (See their results in Tables 3 in [7] and Tables 1-2 in [8], [7] has an average of 46% accuracy degrade and [8] has an average of 73\% F1 score decrease).
>
> **2. Excessive Memory and Communication Overhead:**
>
> Reconstructing $\Delta \mathbf{W}$ creates substantial communication and memory costs, making it impractical for large-scale federated learning.
> The LLaMA-7B model is used in our experiments, reconstructing $\Delta \mathbf{W}$ will produce a high-dimensional full parameter matrix with 6,742,618,112 elements in total. In the setting of federated learning, each client needs to upload this high-dimensional full parameter matrix to the server, resulting in a high communication cost of approximately 6.75GB per client when using float32 precision (6,742,618,112 * 4bytes). In addition, the server needs huge memory consumption to maintain these local reconstructed $\Delta \mathbf{W}$. This expensive strategy is impractical and not scalable for high-dimensional foundation models. Current mainstream methods of federated learning of foundation models is to directly aggregate on low-rank parameter matrices without the reconstruction [1-8]. Our work follows the same approaches.
>
> **3. High Computational Complexity of SVD:**
>
> After reconstructing $\Delta \mathbf{W}$, performing singular value decomposition (SVD) on the global full parameter matrix introduces substantial computational overhead. For a model like LLaMA-7B, where the parameter matrix dimensions are $4096 \times 4096$, the computational complexity of SVD is $O(n^3)$, which means approximately $4096^3$ ($\sim 68.72$ billion floating-point operations) per round. This high computational burden makes the method inefficient and impractical for large-scale federated learning of foundation models.
>
> In contrast, our method avoids the reconstruction of $\Delta \mathbf{W}$ by design with improved effectiveness and efficiency:
>
> 1. **Avoiding Matrix Reconstruction**:  We do not perform the reconstruction of full parameter matrices. Consequently, our method does not suffer from the information loss associated with these approaches. By preserving the complete low-rank structure of the parameter matrices, our method ensures both efficiency and effectiveness in federated learning.
>
> 2. **Reduced Communication and Memory Overhead**:
>    Clients in our method directly upload independent low-rank matrices, significantly reducing communication and memory costs. For the LLaMA-7B model, our method only requires a  communication volume of $78,962 \times 4 \, \text{bytes}$ $(\sim 0.3 \, \text{MB})$ per client. This represents a significant reduction compared to the 6.75GB communication cost caused by reconstructing $\Delta W$.
>
> 3. **Dynamic Rank Selection without SVD on Full-parameter Matrix**:
>   In our method, the server selects the optimal rank based on the singular value matrix $\mathbf{\Sigma}$ without performing SVD on the full parameter matrix, resulting in reduced computational complexity. Specifically,we perform SVD on low-dimensional  $r \times r$ matrices, where $r$ is the rank parameter in LoRA. In the case of  $r = 8$ in our experiments, our computational complexity $8^3$ is significantly reduced compared to that of the full matrix SVD $4096^3$.
>
> In addition, our experimental results have validated the effectiveness and efficiency of our method for federated fine-tuning of foundation models (FFT-FM).

---

> > ### Author Response · Authors · 2024-11-28
> > **Point-by-point response to the comments made by Reviewer W9cF (Continued)**
> >
> > Reference:
> >
> > [1] Youbang Sun, Zitao Li, Yaliang Li, and Bolin Ding. Improving loRA in privacy-preserving federated
> > learning. In The Twelfth International Conference on Learning Representations, 2024.
> >
> > [2] Ziyao Wang, Zheyu Shen, Yexiao He, Guoheng Sun, Hongyi Wang, Lingjuan Lyu, and Ang Li.
> > FLoRA: Federated fine-tuning large language models with heterogeneous low-rank adaptations.
> > In The Thirty-eighth Annual Conference on Neural Information Processing Systems, 2024.
> >
> > [3] Raghav Singhal, Kaustubh Ponkshe, and Praneeth Vepakomma. Exact aggregation for federated and
> > efficient fine-tuning of foundation models, 2024.
> >
> > [4] Pengxin Guo, Shuang Zeng, Yanran Wang, Huijie Fan, Feifei Wang, and Liangqiong Qu. Selective
> > aggregation for low-rank adaptation in federated learning, 2024.
> >
> > [5] Jabin Koo, Minwoo Jang, and Jungseul Ok. Towards robust and efficient federated low-rank adap-
> > tation with heterogeneous clients, 2024.
> >
> > [6] Ngoc-Hieu Nguyen, Tuan-Anh Nguyen, Tuan Nguyen, Vu Tien Hoang, Dung D. Le, and Kok-Seng
> > Wong. Towards efficient communication and secure federated recommendation system via low-
> > rank training. In Proceedings of the ACM Web Conference 2024, WWW ’24, pp. 3940–3951,
> > New York, NY, USA, 2024. Association for Computing Machinery. ISBN 9798400701719. doi:
> > 10.1145/3589334.3645702.
> >
> > [7] Yae Jee Cho, Luyang Liu, Zheng Xu, Aldi Fahrezi, and Gauri Joshi. Heterogeneous lora for fed-
> > erated fine-tuning of on-device foundation models, 2024.
> >
> > [8] Navyansh Mahla and Ganesh Ramakrishnan. Why gradient subspace? identifying and mitigating
> > lora’s bottlenecks in federated fine-tuning of large language models, 2024.

---

> > > ### Author Response · Authors · 2024-11-29
> > > **Thanks to Reviewer W9cF**
> > >
> > > Dear Reviewer W9cF,
> > >
> > > We would like to thank you again for your valuable time and thoughtful and constructive comments. We have tried our best to clarify and address the concerns and comments in the initial response.
> > >
> > > We realize that you may have a busy schedule, but given the short discussion period, we decided to reach out again to see if you could review our responses. We would be happy to provide further clarifications if that helps.

---

> > > > ### Author Response · Authors · 2024-12-01
> > > > **Thanks to Reviewer W9cF**
> > > >
> > > > We thank Reviewer W9cF for taking the time to review our paper. We are confident that we have thoroughly addressed all your concerns. We would greatly appreciate your feedback at your earliest convenience.

---

> > > > > ### Author Response · Authors · 2024-12-02
> > > > > **Thanks to Reviewer W9cF**
> > > > >
> > > > > Thank you again for your review and feedback. We believe we have thoroughly addressed all your concerns. In your initial review, you mentioned that reconstructing $\Delta \mathbf{W}$ before aggregation could lead to performance degradation. However, your rebuttal response suggests using this approach. To clarify, we have analyzed the drawbacks of this method in detail under Question 11, highlighting the following drawbacks: (1) Performance Degradation due to Matrix Reconstruction, (2) Excessive Memory and Communication Overhead due to Matrix Reconstruction, and (3) High Computational Complexity of SVD. Our aim has been to design a high-performance method that effectively addresses these drawbacks.
> > > > >
> > > > > We hope this explanation helps clarify our approach. If you have any further questions, please feel free to let us know.

---

> > > > > > ### Author Response · Authors · 2024-12-02
> > > > > > **Thanks to Reviewer W9cF**
> > > > > >
> > > > > > Dear Reviewer W9cF,
> > > > > >
> > > > > > We thank Reviewer W9cF for taking the time to review our paper. We have made every effort to provide detailed clarifications and additional experimental results to comprehensively address the concerns and comments you raised. As the discussion period is ending, we appreciate knowing if our rebuttal adequately addresses your questions. We are ready to engage in further discussions if necessary. Thank you once again for your contribution.

---

> > > > > > > ### Author Response · Authors · 2024-12-03
> > > > > > > **Thanks to Reviewer W9cF**
> > > > > > >
> > > > > > > Dear Reviewer W9cF,
> > > > > > >
> > > > > > > We want to express our gratitude to you for reviewing our paper. As the discussion period of the rebuttal process is coming to an end, we would be grateful if you could let us know whether the rebuttal addresses your concerns. We are eager to engage in any further discussions if needed. Thank you again for your expertise and assistance.

---

### Author Response · Authors · 2024-11-25
**Common Comments to all Reviewers**

We would like to thank the four reviewers for the helpful and constructive comments. We have tried our best to clarify the concerns and comments by all four reviewers. We have presented the point-by-point response to the comments made by each of the four reviewers. We have included the discussions, analyses, explanations, and experiment results in this rebuttal into the rebuttal revision. We are glad to answer and clarify any further questions and advice from the reviewer for better readability.

Our RAFFT method made novel contributions to the federated fine-tuning of foundation models (FFT-FM) with two unique techniques: (1) Riemannian parameter matching to efficiently address the client-drift issue while reducing computational costs, and (2) Riemannian gradient descent to mitigate rank-drift issue and accelerate convergence. The combination of these two techniques ensures an effective balance between accuracy and efficiency for federated fine-tuning of foundation models.

---

### Note · Authors · 2025-01-16

I have read and agree with the venue's withdrawal policy on behalf of myself and my co-authors.